# Stable and Interpretable Unrolled Dictionary Learning

**Bahareh Tolooshams**                                          *btolooshams@seas.harvard.edu*
**Demba Ba**                                                    *demba@seas.harvard.edu*
*School of Engineering and Applied Sciences*
*Harvard University*

**Reviewed on OpenReview:** *https://openreview.net/forum?id=e3S0Bl2RO8*

## Abstract

The dictionary learning problem, representing data as a combination of a few atoms, has long stood as a popular method for learning representations in statistics and signal processing. The most popular dictionary learning algorithm alternates between sparse coding and dictionary update steps, and a rich literature has studied its theoretical convergence. The success of dictionary learning relies on access to a "good" initial estimate of the dictionary and the ability of the sparse coding step to provide an unbiased estimate of the code. The growing popularity of unrolled sparse coding networks has led to the empirical finding that backpropagation through such networks performs dictionary learning. We offer the theoretical analysis of these empirical results through PUDLE, a Provable Unrolled Dictionary LEarning method. We provide conditions on the network initialization and data distribution sufficient to recover and preserve the support of the latent code. Additionally, we address two challenges; first, the vanilla unrolled sparse coding computes a biased code estimate, and second, gradients during backpropagated learning can become unstable. We show approaches to reduce the bias of the code estimate in the forward pass, and that of the dictionary estimate in the backward pass. We propose strategies to resolve the learning instability by tuning network parameters and modifying the loss function. Overall, we highlight the impact of loss, unrolling, and backpropagation on convergence. We complement our findings through synthetic and image denoising experiments. Finally, we demonstrate PUDLE's interpretability, a driving factor in designing deep networks based on iterative optimizations, by building a mathematical relation between network weights, its output, and the training set.

## 1 Introduction

This paper[1] considers the dictionary learning problem, namely representing data $\boldsymbol{x} \in \mathcal{X} \subset \mathbb{R}^m$ as linear combinations of a few atoms from a dictionary $\boldsymbol{D} \in \mathcal{D} \subset \mathbb{R}^{m \times p}$. Given $\boldsymbol{x}$ and $\boldsymbol{D}$, the problem of recovering the sparse (few non-zero elements) coefficients $\boldsymbol{z} \in \mathbb{R}^p$ is referred to as sparse coding, and can be solved through the lasso (Tibshirani, 1996) (also known as basis pursuit (Chen et al., 2001)):

$$\ell_{\boldsymbol{x}}(\boldsymbol{D}) \coloneqq \min_{\boldsymbol{z} \in \mathbb{R}^p} \ \mathcal{L}_{\boldsymbol{x}}(\boldsymbol{z}, \boldsymbol{D}) + h(\boldsymbol{z}) \tag{1}$$

where $\mathcal{L}_{\boldsymbol{x}}(\boldsymbol{z}, \boldsymbol{D}) = \frac{1}{2}\|\boldsymbol{x} - \boldsymbol{D}\boldsymbol{z}\|_2^2$, and $h(\boldsymbol{z}) = \lambda\|\boldsymbol{z}\|_1$. Specifically, the problem aims to recover a dictionary $\boldsymbol{D}^*$ that generates the data, i.e.,

$$\boldsymbol{x} = \boldsymbol{D}^* \boldsymbol{z}^* \tag{2}$$

where $\boldsymbol{z}^*$ is sparse. Olshausen and Field (Olshausen & Field, 1997) introduced (2) in computational neuroscience as a model for how early layers of the visual cortex process natural images. Sparse coding has been widely studied and utilized in the statistics (Hastie et al., 2015) and signal processing communities (Elad, 2010). A few practical examples are denoising (Elad & Aharon, 2006), super-resolution (Yang et al., 2010), text processing (Jenatton et al., 2011), and classification (Mairal et al., 2009b), where it enables the extraction

---

[1]Source code is available at https://github.com/btolooshams/stable-interpretable-unrolled-dl

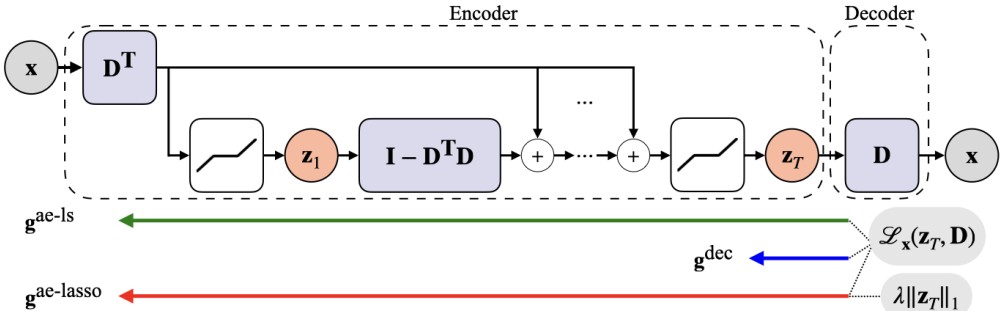

Figure 1: Provable unrolled dictionary learning (PUDLE): Unrolled network architecture with dictionary $\boldsymbol{D}$.

of sparse high-dimensional features representing data. Moreover, sparse modelling is ubiquitous in many other fields such as seismic signal processing (Nose-Filho et al., 2018), radar sensing for target detections (Bajwa et al., 2011), and astrophysics for image reconstruction from interferometric data (Akiyama et al., 2017). Furthermore, Cleary et al. (2017; 2021) use this model to learn a dictionary consisting of gene modules for efficient imaging transcriptomics.

Sparse coding has been utilized to construct neural architectures through approaches such as sparse energy-based models (Ranzato et al., 2007; 2008) or recurrent sparsifying encoders (Gregor & LeCun, 2010). The latter has initiated a growing literature on constructing interpretable deep networks based on an approach referred to as algorithm unrolling (Hershey et al., 2014; Monga et al., 2019). Deep unrolled neural networks have gained popularity as inference maps in recent years due to their computational efficiency and their performance in various domains such as image denoising (Simon & Elad, 2019; Tolooshams et al., 2021a; 2020), super-resolution (Wang et al., 2015), medical imaging (Solomon et al., 2020), deblurring (Schuler et al., 2016; Li et al., 2020), radar sensing (Tolooshams et al., 2021b), and speech processing (Hershey et al., 2014).

Prior to the advent of unrolled networks, gradient-based dictionary learning relied on analytic gradients computed from the lasso given the sparse code. With unrolled networks, automatic differentiation (Baydin et al., 2018), referred to as backpropagation (LeCun et al., 2012) in the reverse-mode, gained attention for parameter estimation (Tolooshams et al., 2018). The automatic gradient is obtained by backpropagation through the algorithm used to estimate the code. Automatic differentiation in reverse and forward-mode (Franceschi et al., 2017) is used in other areas, e.g., hyperparameter selection (Feurer & Hutter, 2019), and in a more relevant context, in the seminal work of LISTA (Gregor & LeCun, 2010). Other works demonstrated empirically the convergence of $\ell_1$-based dictionary learning by backpropagation through unrolled networks (Tolooshams et al., 2021a). Given finite computational power, Tolooshams et al. (2021a) convert sparse coding into an encoder by unrolling $T$ iterations of ISTA (Daubechies et al., 2004; Blumensath & Davies, 2008), and attach to it a linear decoder for reconstructing. Unrolled networks obtained in this manner suffer from two important limitations.

First, the sparse coding step in the forward pass computes a biased estimate of the code. This results, in turn, in a biased estimate of the backward gradient and, hence, a degradation of dictionary recovery performance. Second, as studied recently (Malézieux et al., 2022), inaccuracies in the early iterations of the unrolled network make backpropagation unstable. We address both of these shortcomings in this paper. Moreover, while Malézieux et al. (2022) analyze the gradient computed by backpropagation through unrolled sparse-coding networks, there is no known theoretical analysis of how weight updates using this gradient impact the recovery of a ground-truth code $\boldsymbol{z}^*$, nor of their convergence to a ground-truth dictionary $\boldsymbol{D}^*$.

This paper proposes a Provable Unrolled Dictionary LEarning (PUDLE) (Figure 1). We aim to recover $\boldsymbol{D}^*$ by training the network using backpropagation with a learning rate of $\eta$. Three different choices affect the gradient: the number of unrolled iterations, the loss, and whether one backpropagates through the decoder only or through both the encoder and decoder. We highlight the impact of such choices on the convergence of the training algorithm. Backpropagation through the decoder results in the analytic gradient $\boldsymbol{g}_t^{\text{dec}}$ using the code estimate $\boldsymbol{z}_t$. The gradients $\boldsymbol{g}_t^{\text{ae-lasso}}$ and $\boldsymbol{g}_t^{\text{ae-ls}}$ are computed by backpropagation through the autoencoder using the lasso and least-squares objectives, respectively (Algorithm 2). We compare the gradients with the classical gradient-based alternating-minimization algorithm for dictionary learning (Chatterji & Bartlett, 2017) (i.e., cycling between sparse coding and dictionary update steps using the analytic gradient $\hat{\boldsymbol{g}}$ (Algorithm 1)),

and provide a theoretical analysis of gradient-based recovery of the dictionary $\boldsymbol{D}^*$. We provide sufficient conditions under which the gradient computation, hence the learning, is stable. Additionally, we show how using the reconstruction loss with backpropagation not only does not suffer from backpropagated instability but also ameliorates the propagation of the forward pass bias into the gradient estimate from the backward pass. Finally, we demonstrate the interpretability of the unrolled network. Our contributions are:

- **Unrolled sparse coding**  Unlike prior work (Malézieux et al., 2022) that studies the ability of sparse coding to recover the *solution of the lasso* (1) given the current estimate of the dictionary (we call this local estimation), we study unrolled sparse coding for recovery of the *true generating code in* (2) (we call this global estimation). We provide sufficient conditions on the network and data distributions such that the forward pass recovers (Theorem 4.1) and preserves (Theorem 4.2) the correct code support. Assuming support identification, we show the linear convergence of the code estimated through the unrolled iterations to the solution of the lasso (Theorem 4.3). We provide an explicit code expression at unrolled layer $t$ and its error with respect the ground-truth code $\boldsymbol{z}^*$; we highlight the biased estimate of the code when the forward pass strictly solves lasso (Theorems 4.4 and 4.5). Moreover, in a more general scenario, we show that the error in the code estimate is upper bounded by two terms, i.e., one associated with the dictionary error and the other to the bias of the estimate of code amplitude, due to $\ell_1$-based optimization (Theorem 4.6). The latter highlights that vanilla lasso ($\ell_1$-based) sparse coding computes a biased estimate of codes, and below we discuss strategies to either alleviate this bias in the forward pass or mitigate its propagation into the backward pass for dictionary learning.
- **Mitigation of coding bias propagation into dictionary learning**  We study gradient estimation for dictionary learning in PUDLE. We decompose the upper bound on the gradient errors compared to the gradient direction to recover $\boldsymbol{D}^*$ into terms involving the current dictionary error, the bias of the code estimate, and the lasso loss used to compute the gradient. We show that using only the reconstruction loss while backpropagating (i.e., $\boldsymbol{g}_t^{\text{ae-ls}}$) results in the vanishing of the upper bound due to the usage of lasso loss. This means that given fixed $\lambda$, $\boldsymbol{g}_t^{\text{ae-ls}}$ ameliorates the propagation of the forward pass bias into the backward pass. Specifically, we show that $\boldsymbol{g}_t^{\text{ae-ls}}$ is a better estimator of the direction to recover $\boldsymbol{D}^*$ than $\boldsymbol{g}_t^{\text{dec}}$ and $\boldsymbol{g}_t^{\text{ae-lasso}}$. Hence, weight updates using $\boldsymbol{g}_t^{\text{ae-ls}}$ converges to a closer neighbourhood of $\boldsymbol{D}^*$ (Theorem 4.10). In a supervised image denoising task, we show that the advantage of $\boldsymbol{g}_t^{\text{ae-ls}}$ goes beyond dictionary learning; $\boldsymbol{g}_t^{\text{ae-ls}}$ results in better image denoising compared to $\boldsymbol{g}_t^{\text{dec}}$. Furthermore, our network outperforms the sparse coding scheme in NOODL, a state-of-the-art online dictionary learning algorithm (Rambhatla et al., 2018) (Table 1). Moreover, we show that the bias in the estimate of $\boldsymbol{D}^*$ vanishes as $\lambda_t = \lambda\nu^t$ (with $0 < \nu < 1$) decays within the forward unrolled layers (Figure 16). This strategy, supported by Theorem 4.4, results in an unbiased estimate of the code $\boldsymbol{z}^*$ and recovery of $\boldsymbol{D}^*$ (Theorem 4.11).
- **Stability of unrolled learning**  Our approach to resolve the instability issue of backpropagation in unrolled networks is two-fold. First, we show that under proper dictionary initialization, the instability of the gradient $\boldsymbol{g}_t^{\text{ae-lasso}}$ computation, studied by Malézieux et al. (2022), as $T$ increases is resolved. We give a condition under which the code support is identified and recovered after one iteration and, hence, gradient computation stays stable. Second, in the absence of support identification in early iterations, we propose to use the gradient $\boldsymbol{g}_t^{\text{ae-ls}}$ which resolves the stability issue introduced by lasso loss in the backward pass. We highlight this stability through image denoising training without gradient explosion (Figure 6).
- **Interpretable sparse codes and dictionary**  Prior work has discussed algorithm unrolling for designing interpretable deep architectures based on optimization models (Monga et al., 2019), or interpretability of sparse representations in dictionary learning models (Kim et al., 2010). However, there is no known work to mathematically characterize the interpretability of unrolled network architectures. In this regard, first, we construct a mathematical relation between learned weights (dictionary) at gradient convergence and the training data (Theorem 5.1). Second, we relate the inferred representation/reconstruction of test examples to the training data. We highlight several interpretable features of the unrolled dictionary learning network. Specifically, we perform analysis that provide insights into questions such as *why am I learning a particular feature in the dictionary?* or *from what part of the training set or an image I am learning that feature?* (Figure 8). Moreover, we provide an explanation of the relation between the new test image denoised/reconstructed through the network and the training dataset. The model provides insights on *how training images are used to reconstruct a new test image* (Figure 9) or *how the test image picks up training images that have a similar representation to itself to reconstruct* (Figure 10).

## 2 Related Works

There is vast literature on the theoretical convergence of dictionary learning. Spielman et al. (2012) proposed a factorization method to recover the dictionary in the undercomplete setting (i.e., $p \leq m$). Barak et al. (2015) proposed to solve dictionary learning via sum-of-squares semidefinite program. K-SVD (Aharon et al., 2006) and MOD (Engan et al., 1999) are popular greedy approaches. Alternating-minimization-based methods have been used extensively in theory and practice (Jain et al., 2013; Agarwal et al., 2014; Arora et al., 2014).

Recent work has incorporated gradient-based updates into alternating minimization (Chatterji & Bartlett, 2017; Arora et al., 2015; Rambhatla et al., 2018). Chatterji & Bartlett (2017) provided a finite sample analysis and convergence guarantees when updating the dictionary using the analytic gradient. Arora et al. (2015) proposed neurally plausible sparse coding approaches with analytic gradients. Another work focused on online dictionary learning (Mairal et al., 2009a) with an unbiased gradient updates (Rambhatla et al., 2018). Arora et al. (2015) discussed methods to reduce the bias of dictionary estimate, and Rambhatla et al. (2018) showed how to reduce bias in code and dictionary estimates. A common feature in the above-mentioned work is the use of analytic gradients, i.e., explicitly designing gradient updates independent of the sparse coding step and not utilizing automatic gradients with deep learning optimizers. A theoretical analysis of backpropagation for dictionary learning exists only for shallow autoencoders (Rangamani et al., 2018; Nguyen et al., 2019).

The theoretical analysis of unrolled neural networks has mainly analyzed the convergence speed of variants of LISTA (Gregor & LeCun, 2010), where the focus is on sparse coding (i.e., the encoder) not dictionary learning (Sprechmann et al., 2012; Xin et al., 2016; Moreau & Bruna, 2017; Giryes et al., 2018; Chen et al., 2018; Liu & Chen, 2019; Ablin et al., 2019). Moreau & Bruna (2017) showed that upon successful factorization of the Gram matrix of the dictionary within layers, the network achieves accelerated convergence. Giryes et al. (2018) examined the tradeoffs between reconstruction accuracy and convergence speed of LISTA. Moreover, Chen et al. (2018) studied the learning dynamics of the weights and biases of unrolled-ISTA and proved that it achieves linear convergence. Follow-up works investigated the dynamics of step size in a recursive sparse coding encoder (Liu & Chen, 2019; Ablin et al., 2019). Ablin et al. (2019) minimized the lasso through backpropagation but still assumed the knowledge of the dictionary at the decoder.

Ablin et al. (2020) compared analytic and automatic gradient estimators of min-min optimizations with smooth and differentiable functions. Moreover, Malézieux et al. (2022) studied the stability of gradient approximation in the early regime of unrolling for dictionary learning. Unlike our work, where we evaluate the gradients for model recovery, Ablin et al. (2020) and Malézieux et al. (2022) studied the asymptotic gradient errors locally in each step of an alternating minimization and did not provide errors concerning $\boldsymbol{z}^*$ or $\boldsymbol{D}^*$.

## 3 Preliminaries

Given $n$ independent samples, dictionary learning aims to minimize the empirical risk, i.e.,

$$\min_{\boldsymbol{D} \in \mathcal{D}} \ \mathcal{R}_n(\boldsymbol{D}) \quad \text{with} \quad \mathcal{R}_n(\boldsymbol{D}) \triangleq \tfrac{1}{n} \sum_{i=1}^{n} \ell_{\boldsymbol{x}^i}(\boldsymbol{D}) \tag{3}$$

where $\lim_{n \to \infty} \mathcal{R}_n(\boldsymbol{D}) = \mathbb{E}_{\boldsymbol{x} \in \mathcal{X}} [\ell_{\boldsymbol{x}}(\boldsymbol{D})]$ a.s. To prevent scaling ambiguity between the code $\boldsymbol{z}$ and dictionary $\boldsymbol{D}$, it is common to constrain the norm of the dictionary columns. Hence, we define the set of feasible solutions for the dictionary as $\mathcal{D} \triangleq \{\boldsymbol{D} \in \mathbb{R}^{m \times p} \text{ s.t. } \forall j \in \{1, 2, \dots, p\}, \ \|\boldsymbol{D}_j\|_2^2 \leq 1\}$. We can project estimates of $\boldsymbol{D}$ onto the feasible set by performing $\boldsymbol{D}_j \leftarrow {}^1/_{\max(\|\boldsymbol{D}_j\|_2, 1)} \boldsymbol{D}_j$, either at every update or at the end of training. We assume certain properties on the data, specifically its domain (Assumption 3.1), energy (Assumption 3.2), code distribution (Assumption 3.3), and generating dictionary (Assumption 3.4).

**Assumption 3.1** (Domain signals). *$\mathcal{X}$ and $\mathcal{D}$ are both compact convex sets.*

**Assumption 3.2** (Bounded signals). *$\exists M > 0$ s.t. $\|\boldsymbol{x}\|_2 < M \ \forall \boldsymbol{x} \in \mathcal{X}$.*

**Assumption 3.3** (Code distribution). *The code $\boldsymbol{z}^*$ is at most s-sparse with the support $S^* = supp(\boldsymbol{z}^*)$. Each element in $S^*$ is chosen from the set $[1, p]$, uniformly at random without replacement. $p_i = P(i \in S^*) = \Theta(s/p)$, and $p_{ij} = P(i, j \in S^*) = \Theta(s^2/p^2)$. Given the support, $\boldsymbol{z}_S^*$ is i.i.d, has symmetric probability distribution density function, $\mathbb{E}[\boldsymbol{z}_{(i)}^* \mid i \in S^*] = 0$ and $\mathbb{E}[\boldsymbol{z}_{(S)}^* \boldsymbol{z}_{(S^*)}^{*T} \mid S^*] = \boldsymbol{I}$. Moreover, the non-zero entries of the code are sub-Gaussian and lower bounded, i.e., for $i \in S^*$, $|\boldsymbol{z}_{(i)}^*| \geq C_{\min}$ where $0 < C_{\min} \leq 1$.*

**Assumption 3.4** (Generating dictionary)**.** $\boldsymbol{D}^*$ *is $\mu$-incoherent (see Definition A.1) where $\mu = \mathcal{O}(\log{(m)})$.* $\boldsymbol{D}^*$ *is unit-norm columns matrix ($\|\boldsymbol{D}_i^*\|_2 = 1$), $\|\boldsymbol{D}^*\|_2 = \mathcal{O}(\sqrt{p/m})$, and $p = \mathcal{O}(m)$.*

To achieve model recovery using gradient descent, we assume an appropriate dictionary initialization, i.e.,

**Assumption 3.5** (Dictionary closeness)**.** *The initial dictionary $\boldsymbol{D}^{(0)}$ is $(\delta_0, 2)$-close to $\boldsymbol{D}^*$ (see Definition A.2). The dictionary closeness at every update is denoted by $\|\boldsymbol{D}_j^{(l)} - \boldsymbol{D}_j^*\|_2 \leq \delta_l \ \forall j$. Furthermore, $\delta_l = \mathcal{O}^*(1/\log p)$.*

Arora et al. (2015) proposed a dictionary initialization method offering $(\delta, 2)$-close to $\boldsymbol{D}^*$ for $\delta = O^*(1/\log m)$. The method is based on pairwise reweighting of samples $\{\boldsymbol{x}^i\}_{i=1}^n$ from the generative model (2), and does not require access to $\boldsymbol{D}^*$. In addition, Rambhatla et al. (2018) utilize dictionary closeness assumptions and such dictionary initialization for their theoretical analysis. Moreover, Agarwal et al. (2017) proposed a clustering approach to find a close initial estimate of the dictionary.

Given the $\mu$-incoherence of $\boldsymbol{D}^*$ (Assumption 3.4) and $\delta_l$-closeness of the dictionary, $\boldsymbol{D}^{(l)}$ is $\mu_l$-incoherent, i.e.,

**Lemma 3.1** ($\mu_l$-incoherent)**.** $\boldsymbol{D}^{(l)}$ *is $\mu_l$-incoherent where $\mu_l = \mu + 2\sqrt{m}\delta_l$.*

The recurrent encoder and decoder, which perform the computations shown in Algorithm 2, use the loss $\mathcal{L}$ and proximal operator $\mathcal{P}_b(v) \triangleq \text{sign}(v)\max(|v| - b, 0)$ for the $\ell_1$ norm $h\colon \mathbb{R}^p \to \mathbb{R}$. The encoder implements ISTA (Daubechies et al., 2004; Blumensath & Davies, 2008) with step size $\alpha$, assumed to be less than $1/\sigma_{\max}^2(\boldsymbol{D})$. With infinite encoder unrolling, the encoder's output is the solution to the lasso (1), following the optimality condition (Lemma A.3) where we denote $f_{\boldsymbol{x}}(\boldsymbol{z}, \boldsymbol{D}) \triangleq \mathcal{L}_{\boldsymbol{x}}(\boldsymbol{z}, \boldsymbol{D}) + h(\boldsymbol{z})$. One immediate observation is that $\lambda \geq \|\boldsymbol{D}^{\mathrm{T}}\boldsymbol{x}\|_\infty \Leftrightarrow \{\boldsymbol{0}\} \in \arg\min f_{\boldsymbol{x}}(\boldsymbol{z}, \boldsymbol{D})$. We assume $\lambda < \|\boldsymbol{D}^{\mathrm{T}}\boldsymbol{x}\|_\infty$. We specify in Theorem 4.1 and Theorem 4.2 the conditions on $\lambda$ at every encoder iteration to ensure support recovery and its preservation through the encoder. In case of a constant $\lambda$ across encoder iterations while using $\boldsymbol{D}^*$ as the dictionary (i.e., sparse coding using $\ell_1$ norm), the network recovers a biased code $\hat{\boldsymbol{z}}^*$. We denote this amplitude error in the code by $\hat{\delta}^* \triangleq \|\hat{\boldsymbol{z}}^* - \boldsymbol{z}^*\|_2$ which is small and goes to zero with $\lambda$ decaying through the encoder.

In addition, we assume the solution to (1) is unique; sufficient conditions for uniqueness in the overcomplete case (i.e., $p > m$) are extensively studied in the literature (Wainwright, 2009; Candès & Plan, 2009; Tibshirani, 2013). Tibshirani (2013) discussed that the solution is unique with probability one if entries of $\boldsymbol{D}$ are drawn from a continuous probability distribution (Tibshirani, 2013) (Assumption 3.6). This assumption implies that $\boldsymbol{D}_S^{\mathrm{T}}\boldsymbol{D}_S$ is full-rank. We argue that as long as the data $\boldsymbol{x} \in \mathcal{X}$ are sampled from a continuous distribution, this assumption holds for the entire learning process. The preservation of this property is guaranteed at all iterations of the alternating minimization proposed in (Agarwal et al., 2014). Moreover, this assumption has been previously considered in analyses of unrolled sparse coding networks (Ablin et al., 2019; Malézieux et al., 2022) and can be extended to $\ell_1$-based optimization problems (Tibshirani, 2013; Rosset et al., 2004).

**Assumption 3.6** (Lasso uniqueness)**.** *The entries of the dictionary $\boldsymbol{D}$ are continuously distributed. Hence, the minimizer of (1) is unique, i.e., $\hat{\boldsymbol{z}} = \arg\min f_{\boldsymbol{x}}(\boldsymbol{z}, \boldsymbol{D})$ with probability one.*

Lemma 3.2 states the fixed-point property of the encoder recursion (Parikh & Boyd, 2014). Given the definitions for *Lipschitz* and *Lipschitz differentiable* functions, (Definitions A.3 and A.4), the loss $\mathcal{L}$ and function $h$ satisfy following *Lipschitz* properties.

**Lemma 3.2** (Fixed-point property of lasso)**.** *Given Assumption 3.6, we have $\boldsymbol{0} \in \nabla_1\mathcal{L}_{\boldsymbol{x}}(\hat{\boldsymbol{z}}, \boldsymbol{D}) + \partial h(\hat{\boldsymbol{z}})$. The minimizer is a fixed-point of the mapping, i.e., $\hat{\boldsymbol{z}} = \mathcal{P}_{\alpha\lambda}(\hat{\boldsymbol{z}} - \alpha\nabla_1\mathcal{L}_{\boldsymbol{x}}(\hat{\boldsymbol{z}}, \boldsymbol{D})) = \Phi(\hat{\boldsymbol{z}})$ (Parikh & Boyd, 2014).*

**Lemma 3.3** (Lipschitz differentiable least squares)**.** *Given $\mathcal{L}_{\boldsymbol{x}}(\boldsymbol{z}, \boldsymbol{D}) = \frac{1}{2}\|\boldsymbol{x} - \boldsymbol{D}\boldsymbol{z}\|_2^2$, $\mathcal{D}$, and Assumption 3.2, the loss is Lipschitz differentiable. Let $L_1$ and $L_2$ denote the Lipschitz constants of the first derivatives $\nabla_1\mathcal{L}_{\boldsymbol{x}}(\boldsymbol{z}, \boldsymbol{D})$ and $\nabla_2\mathcal{L}_{\boldsymbol{x}}(\boldsymbol{z}, \boldsymbol{D})$, $L_{11}$ and $L_{21}$ the Lipschitz constants of the second derivatives $\nabla_{11}^2\mathcal{L}_{\boldsymbol{x}}(\boldsymbol{z}, \boldsymbol{D})$ and $\nabla_{21}^2\mathcal{L}_{\boldsymbol{x}}(\boldsymbol{z}, \boldsymbol{D})$, all w.r.t $\boldsymbol{z}$. Let $\nabla_1\mathcal{L}_{\boldsymbol{x}}(\boldsymbol{z}, \boldsymbol{D})$ be $L_{1D}$-Lipschitz w.r.t $\boldsymbol{D}$, and we denote the Lipschitz constant of $\nabla_{11}\mathcal{L}_{\boldsymbol{x}}(\boldsymbol{z}, \boldsymbol{D})$ and $\nabla_{21}\mathcal{L}_{\boldsymbol{x}}(\boldsymbol{z}, \boldsymbol{D})$ w.r.t to $\boldsymbol{D}$ by $L_{11D}$ and $L_{21D}$, respectively.*

**Lemma 3.4** (Lipschitz proximal)**.** *Given $h(\boldsymbol{z}) = \lambda\|\boldsymbol{z}\|_1$, its proximal operator has bounded sub-derivative, i.e., $\|\partial\mathcal{P}_\lambda(\boldsymbol{z})\|_2 \leq c_{prox}$.*

---

**Algorithm 1:** Classical alternating-minimization-based dictionary learning using lasso (1).

---

**Initialize:** Samples $\{\boldsymbol{x}^i\}_{i=1}^n \in \mathcal{X}$, initial dictionary $\boldsymbol{D}^{(0)}$
**Repeat:** $l = 0, 1, \ldots,$ number of epochs
   **Sparse coding step**:   $\boldsymbol{z}^{i(l)} = \arg\min_{\boldsymbol{z}} \mathcal{L}_{\boldsymbol{x}^i}(\boldsymbol{z}, \boldsymbol{D}^{(l)}) + h(\boldsymbol{z}),$   (for $i \in [1, n]$)
   **Dictionary update**:   $\boldsymbol{D}^{(l+1)} = \boldsymbol{D}^{(l)} - \eta\hat{\boldsymbol{g}}^{(l)}$   where   $\hat{\boldsymbol{g}}^{(l)} \triangleq \frac{1}{n}\sum_{i=1}^n \nabla_2 \mathcal{L}_{\boldsymbol{x}^i}(\boldsymbol{z}^{i(l)}, \boldsymbol{D}^{(l)})$

---

**Algorithm 2:** PUDLE: Provable unrolled dictionary learning framework.

---

**Initialize:** Samples $\{\boldsymbol{x}^i\}_{i=1}^n \in \mathcal{X}$, initial dictionary $\boldsymbol{D}^{(0)}$, and $\boldsymbol{z}_0 = \boldsymbol{0}$.
**Repeat:** $l = 0, 1, \ldots,$ number of epochs
  **Forward pass**: (for $i \in [1, n]$)

$$\begin{aligned}
&\text{Encoder:} \quad \boldsymbol{z}_{t+1}^{i(l)} = \Phi(\boldsymbol{z}_t^{i(l)}, \boldsymbol{D}^{(l)}) = \mathcal{P}_{\alpha\lambda}(\boldsymbol{z}_t^{i(l)} - \alpha\nabla_1\mathcal{L}_{\boldsymbol{x}^i}(\boldsymbol{z}_t^{i(l)}, \boldsymbol{D}^{(l)})) \text{ (repeat for } T) \\
&\text{Decoder:} \qquad \hat{\boldsymbol{x}}^{i(l)} = \boldsymbol{D}^{(l)}\boldsymbol{z}_T^{i(l)}
\end{aligned} \quad (6)$$

  **Backward pass**: $\boldsymbol{D}^{(l+1)} = \boldsymbol{D}^{(l)} - \eta\boldsymbol{g}_T^{(l)}$   where $\boldsymbol{g}_T^{(l)}$ is either of

$$\begin{aligned}
\boldsymbol{g}_T^{(l)\ \text{dec}} &\triangleq \frac{1}{n}\sum_{i=1}^n \nabla_2\mathcal{L}_{\boldsymbol{x}^i}(\boldsymbol{z}^{i(l)}, \boldsymbol{D}^{(l)}) \\
\boldsymbol{g}_T^{(l)\ \text{ae-lasso}} &\triangleq \frac{1}{n}\sum_{i=1}^n \nabla_2\mathcal{L}_{\boldsymbol{x}^i}(\boldsymbol{z}_T^{i(l)}, \boldsymbol{D}^{(l)}) + \boldsymbol{J}_T^{i(l)+}\left(\nabla_1\mathcal{L}_{\boldsymbol{x}^i}(\boldsymbol{z}_T^{i(l)}, \boldsymbol{D}^{(l)}) + \partial h(\boldsymbol{z}_T^{i(l)})\right) \\
\boldsymbol{g}_T^{(l)\ \text{ae-ls}} &\triangleq \frac{1}{n}\sum_{i=1}^n \nabla_2\mathcal{L}_{\boldsymbol{x}^i}(\boldsymbol{z}_T^{i(l)}, \boldsymbol{D}^{(l)}) + \boldsymbol{J}_T^{i(l)+}\nabla_1\mathcal{L}_{\boldsymbol{x}^i}(\boldsymbol{z}_T^{i(l)}, \boldsymbol{D}^{(l)})
\end{aligned} \quad (7)$$

  See Definition 4.1 for $\boldsymbol{J}_T^+$.

---

## 4 Unrolled Dictionary Learning

The gradients defined in PUDLE (Algorithm 2) can be compared against the local direction at each update of classical alternating-minimization (Algorithm 1). Assuming there are infinite samples, i.e.,

$$\text{Best local direction}: \quad \hat{\boldsymbol{g}} \triangleq \lim_{n\to\infty}\frac{1}{n}\sum_{i=1}^n \nabla_2\mathcal{L}_{\boldsymbol{x}^i}(\hat{\boldsymbol{z}}^i, \boldsymbol{D}) = \mathbb{E}_{\boldsymbol{x}\in\mathcal{X}}\left[\nabla_2\mathcal{L}_{\boldsymbol{x}}(\hat{\boldsymbol{z}}, \boldsymbol{D})\right] \quad (4)$$

where $\hat{\boldsymbol{z}} = \arg\min_{\boldsymbol{z}\in\mathbb{R}^p} \mathcal{L}_{\boldsymbol{x}}(\boldsymbol{z}, \boldsymbol{D}) + h(\boldsymbol{z})$. Additionally, to assess the estimators for model recovery, hence dictionary learning, we compare them against gradient pointing towards $\boldsymbol{D}^*$, namely

$$\text{Desired global gradient for } \boldsymbol{D}^*: \quad \boldsymbol{g}^* \triangleq \lim_{n\to\infty}\frac{1}{n}\sum_{i=1}^n \nabla_2\mathcal{L}_{\boldsymbol{x}^i}(\boldsymbol{z}^{i*}, \boldsymbol{D}) = \mathbb{E}_{\boldsymbol{x}\in\mathcal{X}}\left[\nabla_2\mathcal{L}_{\boldsymbol{x}}(\boldsymbol{z}^*, \boldsymbol{D})\right]. \quad (5)$$

To see why the above is the desired direction, $(\boldsymbol{z}^*, \boldsymbol{D}^*)$ is a critical point of the loss $\mathcal{L}$ which reaches zero for data following the model (2). Hence, to reach $\boldsymbol{D}^* \in \arg\min_{\boldsymbol{D}\in\mathcal{D}} \mathbb{E}_{\boldsymbol{x}\in\mathcal{X}}[\mathcal{L}_{\boldsymbol{x}}(\boldsymbol{z}^*, \boldsymbol{D})]$, we move towards the direction minimizing the loss in expectation. Specifically, using the gradient $\nabla_2\mathcal{L}_{\boldsymbol{x}}(\boldsymbol{z}^*, \boldsymbol{D}) = -(\boldsymbol{x} - \boldsymbol{D}\boldsymbol{z}^*)\boldsymbol{z}^{*\text{T}} = (\boldsymbol{D} - \boldsymbol{D}^*)\boldsymbol{z}^*\boldsymbol{z}^{*\text{T}}$ as a descent direction, we move from $\boldsymbol{D}$ toward $\boldsymbol{D}^*$ modulo the code presence matrix $\boldsymbol{z}^*\boldsymbol{z}^{*\text{T}}$. Given these directions, we analyze the error of the gradients $\boldsymbol{g}_t^{\text{dec}}$, $\boldsymbol{g}_t^{\text{ae-lasso}}$, and $\boldsymbol{g}_t^{\text{ae-ls}}$ assuming infinite samples. In local analysis, we compare the code and gradient estimates to the lasso optimization in each update of the alternating minimization. In global analysis, we evaluate the performance in recovery of the ground-truth code $\boldsymbol{z}^*$ and the dictionary $\boldsymbol{D}^*$. In this regard, we first study the forward pass.

### 4.1 Forward pass

We show convergence results in the forward pass for $\boldsymbol{z}$ and the Jacobian, i.e.,

**Definition 4.1** (Code Jacobian)**.** *Given $\boldsymbol{D}$, the Jacobian of $\boldsymbol{z}_t$ is defined as $\boldsymbol{J}_t \triangleq \frac{\partial \boldsymbol{z}_t}{\partial \boldsymbol{D}}$ with adjoint $\boldsymbol{J}_t^+$.*

The forward pass analyses give upper bounds on the error between $\boldsymbol{z}_t$ and $\hat{\boldsymbol{z}}$ and the error between $\boldsymbol{J}_t$ and $\hat{\boldsymbol{J}}$ as a function of unrolled iterations $t$. We define $\hat{\boldsymbol{J}}$ as following: considering the function $\boldsymbol{z} \to \mathcal{L}_{\boldsymbol{x}}(\boldsymbol{z}, \boldsymbol{D}) + h(\boldsymbol{z})$, $\hat{\boldsymbol{z}}(\boldsymbol{D})$ is its minimizer and $\hat{\boldsymbol{J}} = \frac{\partial \hat{\boldsymbol{z}}(\boldsymbol{D})}{\partial \boldsymbol{D}}$. We will require these errors in Section 4.2, where we analyze the gradient estimation errors. Similar to (Chatterji & Bartlett, 2017), the error associated with $\boldsymbol{g}_t^{\text{dec}}$ depends on

the code convergence. Unlike $\boldsymbol{g}_t^{\text{dec}}$, the convergence of backpropagation with gradient estimates $\boldsymbol{g}_t^{\text{ae-lasso}}$ and $\boldsymbol{g}_t^{\text{ae-ls}}$ relies on the convergence properties of the code *and* the Jacobian (Ablin et al., 2020). Forward-pass theories are based on studies by Gilbert (1992) on the convergence of variables and their derivatives in an iterative process governed by a smooth operator (Gilbert, 1992). Moreover, Hale et al. (2007) studied the convergence analysis of fixed point iterations for $\ell_1$ regularized optimization problems (Hale et al., 2007).

**Support recovery and preservation**   We re-state a result from (Hale et al., 2007) on support selection.

**Proposition 4.1** (Finite-iteration support selection)**.** *Given  Assumption 3.6, let $\hat{\boldsymbol{z}} = \arg\min f_{\boldsymbol{x}}(\boldsymbol{z}, \boldsymbol{D})$ with $S \triangleq supp(\hat{\boldsymbol{z}})$. There exists a $B > 0$ such that $supp(\boldsymbol{z}_t) = S, \forall t > B$.*

This means the unrolled encoder identifies the support in finite iterations. Support recovery in finite iterations has been studied in the literature for LISTA (Chen et al., 2018), Step-LISTA (Ablin et al., 2019), and shallow autoencoders (Arora et al., 2015; Rangamani et al., 2018; Nguyen et al., 2019; Tolooshams et al., 2020). We show that under proper initialization of the dictionary, the encoder achieves linear convergence. Arora et al. (2015) discussed some appropriate initialization which is used by Rambhatla et al. (2018). Given initial closeness $\delta_0$, the encoder selects and recovers the correct signed support of the code with high probability in one iteration $B = 1$ (Theorem 4.1), and the iterations preserve the correct support (Theorem 4.2). In spite of slow convergence of ISTA Liang et al. (2014), support recovery after one iteration in unrolled networks is studied in the literature (Arora et al., 2015; Rambhatla et al., 2018; Chen et al., 2018; Nguyen et al., 2019).

**Theorem 4.1** (Forward pass support recovery)**.** *Given Assumptions 3.3 and 3.4, suppose $\boldsymbol{D}^{(l)}$ is $\delta_l = \mathcal{O}^*(1/\sqrt{\log p})$ close to $\boldsymbol{D}^*$. If $s = \mathcal{O}^*(\sqrt{m}/\mu \log m)$, and $\mu = \mathcal{O}(\log m)$, then with probability of at least $1 - \epsilon_{supp\text{-}rec}^{(l)}$, the choice of $\lambda_0 = C_{\min}/4$ recovers the support of the code $\boldsymbol{z}^*$ in one encoder iteration, i.e., $sign(\mathcal{S}_{\alpha\lambda_0}(\alpha\boldsymbol{D}^{(l)T}\boldsymbol{x}) = sign(\boldsymbol{z}^*)$, where $\epsilon_{supp\text{-}rec}^{(l)} = 2p\exp\left(-\frac{C_{\min}^2}{\mathcal{O}^*(\delta_l^2)}\right)$.*

**Theorem 4.2** (Forward pass support preservation)**.** *Given Assumptions 3.3 and 3.4, suppose $\boldsymbol{D}^{(l)}$ is $\delta_l = \mathcal{O}^*(1/\log p)$ close to $\boldsymbol{D}^*$. If $s = \mathcal{O}^*(\sqrt{m}/\mu \log m)$, $\mu = \mathcal{O}(\log m)$, and the regularizer and step size are chosen such that $\lambda_t^{(l)} = \frac{\mu_l}{\sqrt{m}}\|\boldsymbol{z}^* - \boldsymbol{z}_t\|_1 + a_\gamma = \boldsymbol{\Omega}\left(\frac{s\log m}{\sqrt{m}}\right)$ and $\alpha^{(l)} \leq 1 - \frac{2\lambda_t^{(l)} - (1 - \frac{\delta_l^2}{2})C_{\min}}{\lambda_{t-1}^{(l)}}$, then with probability of at least $1 - \epsilon_{supp\text{-}pres}^{(l)}$, the support, recovered at the first iteration, is preserved through the encoder iterations. We have $a_\gamma = \mathcal{O}(\sqrt{s\delta_l})$ and $\epsilon_{supp\text{-}pres}^{(l)} := \epsilon_{supp\text{-}rec}^{(l)} + \epsilon_\gamma^{(l)} = 2p\exp\left(\frac{-C_{\min}^2}{\mathcal{O}^*(\delta_l^2)}\right) + 2s\exp\left(\frac{-1}{\mathcal{O}(\delta_l)}\right)$.*

The support preservation conditions on $\lambda_t$ and $\alpha$ introduce two insights. First, with an increase of $t$, the code error decrease, hence the lower bound on $\lambda_t$. Second, the decay of $\lambda_t$ as the encoder unrolls increases the upper bound on $\alpha$. Hence, we suggest a decaying strategy in values of $\lambda_t$ as $t$ increases.

The utilization of knowledge of the code error, as we do in Theorem 4.2, to set the proper thresholding/bias/regularization parameters ($\lambda$) constitutes a fairly standard practice. Below we discuss similar results in the literature. For the preservation of correct signed-support in a sparse coding network, Rambhatla et al. (2018) provided a proper thresholding value at every iteration as a function of the $\ell_1$-norm of the code error with respect to a ground-truth code; they additionally demonstrated an upper bound on the estimate of the code coefficients as a function of dictionary closeness. Moreover, Nguyen et al. (2019) used information on the range of ground-truth code to choose proper biases in their neural network to guarantee support recovery. Chen et al. (2018) similarly provided an upper bound on the bias of their unrolled sparse coding network at every layer as a function of $\ell_2$-norm error between the code estimate at the layer and the ground-truth code. Overall, the error between a code estimate and the ground-truth code appearing in the lower bound on $\lambda_t^{(l)}$ can further simplified into terms related to terms such as the dictionary closeness $\delta_l$, code sparsity. For example, Chatterji & Bartlett (2017), for their particular sparse coding algorithm, provided $\ell_\infty$-norm upper bound as a function of terms such as code sparsity, data dimensionality, code range, and dictionary error.

**Code convergence and error**   Given the support recovery and its preservation, the encoder convergence studied in (Malézieux et al., 2022) can achieve linear convergences after its first iteration. We re-state this result on the rate of convergence of the encoder in Theorem 4.3. We drop the superscript $(l)$ to simplify the notation.

**Theorem 4.3** (Local forward pass code convergence). *Given the encoder $z_{t+1} = \Phi(z_t, D)$, Assumption 3.6, Lemmas 3.2, A.1 and A.2, then $\exists\ \rho < 1, B > 0$ s.t. $\|z_t - \hat{z}\|_2 \leq \mathcal{O}(\rho^t)\ \forall t > B$, where $\hat{z}$ is the unique minimizer of lasso (1). Furthermore, given Theorem 4.1 and Theorem 4.2, $B = 1$.*

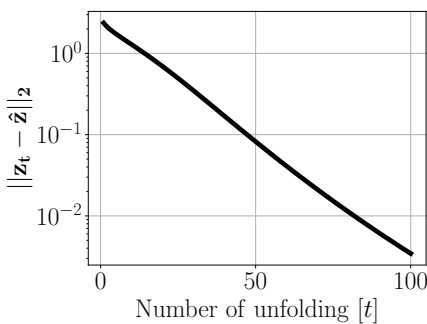

Theorem 4.3 shows that in PUDLE, $z_t$ converges to $\hat{z}$ at a linear rate eventually after a certain number of unrolling (Figure 2). The local linear convergence of ISTA and FISTA (Beck & Teboulle, 2009) (with global rates of $\mathcal{O}(1/t)$ and $\mathcal{O}(1/t^2)$) in the neighbourhood of a fixed-point is studied in (Tao et al., 2016). The speed of convergence depends on when support selection happens (Proposition 4.1) (Bredies & Lorenz, 2008; Zhang et al., 2017b; Liang et al., 2014). We showed in Theorem 4.1 and Theorem 4.2 that under mild assumptions, the

Figure 2: Code convergence (Theorem 4.3). As the network unrolls, $z_t$ converges to $\hat{z}$, the solution of lasso.

support is selected and recovered after one encoder iteration. In addition to local convergence, we focus on recovery of $z^*$ and show error on the unrolled code coefficients $z_{t,(j)}^{(l)}$ with respect to ground-truth $z_{(j)}^*$ as $t$ increases. In Theorem 4.4, we consider the case where $\lambda_t$ at layer $t$ is set to according to Theorem 4.2; the bias decreases as the code error decreases among the layers and dictionary updates. We provide an upper bound on the coefficients errors as a function of code sparsity, dictionary error, and an unrolling error $e_{t,j}^{(l)\text{unroll}}$. The unrolling error goes to zero for appropriately large $t$. Moreover, Theorem 4.5 studies the case where the bias is fixed across the layers. In this scenario, we observe an additional term of $\lambda^{\text{fixed}}$ in the upper bounds on the code coefficients error; this term shows that the code error when we strictly perform $\ell_1$-norm based sparse coding does not go to zero. We refer to this error as an amplitude bias estimate error.

**Theorem 4.4** (Global forward pass code error with variable $\lambda_t$). *Given Assumptions 3.3 and 3.4, suppose $D^{(l)}$ is $\mu_l$-incoherent and $\delta_l = \mathcal{O}^*(1/\log p)$ close to $D^*$. If $s = \mathcal{O}^*(\sqrt{m}/\mu \log m)$, $\mu = \mathcal{O}(\log m)$, and the regularizer and step size are chosen such that $\lambda_t^{(l)} = \frac{\mu_l}{\sqrt{m}}\|z^* - z_t\|_1 + a_\gamma = \Omega(\frac{s \log m}{\sqrt{m}})$ and $\alpha^{(l)} \leq 1 - \frac{2\lambda_t^{(l)} - (1 - \frac{\delta_l^2}{2})C_{\min}}{\lambda_{t-1}^{(l)}}$, then with probability of at least $1 - \epsilon_{supp\text{-}pres}^{(l)}$, for $j \in supp(z^*)$, the code coefficient error is*

$$|z_{t,(j)}^{(l)} - z_{(j)}^*| \leq \mathcal{O}(\sqrt{s\|D_j^{(l)} - D_j^*\|_2} + e_{t,j}^{(l)\text{unroll}}) \tag{8}$$

*and*

$$z_{T,(j)} = z_{(j)}^*(1 - \beta_j^{(l)}) + \zeta_{T,j}^{(l)} \tag{9}$$

*where $e_{t,j}^{(l)\text{unroll}} := 2(s-1)t\alpha\frac{\mu_l}{\sqrt{m}}\max_i |z_{0,(i)}^{(l)} - z_{(i)}^*|\delta_{\alpha,t-1} + |z_{0,(j)}^{(l)} - z_{(j)}^*|\delta_{\alpha,t}$, $\delta_{\alpha,t} := (1 - \alpha + 2\alpha\frac{\mu_l}{\sqrt{m}})^t$, $|\zeta_{T,j}^{(l)}| = \mathcal{O}(a_\gamma)$ with $a_\gamma = \mathcal{O}(\sqrt{s}\delta_l)$, $\beta_j^{(l)} = \langle D_j^* - D_j^{(l)}, D_j^* \rangle \leq \frac{\delta_l^2}{2}$ and $\epsilon_{supp\text{-}pres}^{(l)} := \epsilon_{supp\text{-}rec}^{(l)} + \epsilon_\gamma^{(l)} = 2p\exp\left(\frac{-C_{\min}^2}{\mathcal{O}^*(\delta_l^2)}\right) + 2s\exp\left(\frac{-1}{\mathcal{O}(\delta_l)}\right)$. With appropriately large $t$, $|z_{t,(j)}^{(l)} - z_{(j)}^*| = \mathcal{O}(\sqrt{s\|D_j^{(l)} - D_j^*\|_2})$.*

**Theorem 4.5** (Global forward pass code error with fixed $\lambda_t$). *Given Assumptions 3.3 and 3.4, suppose $D^{(l)}$ is $\mu_l$-incoherent and $\delta_l = \mathcal{O}^*(1/\log p)$ close to $D^*$. If $s = \mathcal{O}^*(\sqrt{m}/\mu \log m)$, $\mu = \mathcal{O}(\log m)$, and the regularizer and step size are chosen such that $\lambda_t^{(l)} = \lambda^{\text{fixed}} = \frac{\mu_l}{\sqrt{m}}\|z^* - z_0\|_1 + a_\gamma = \Omega(\frac{s \log m}{\sqrt{m}})$ and $\alpha^{(l)} \leq 1 - \frac{2\lambda_t^{(l)} - (1 - \frac{\delta_l^2}{2})C_{\min}}{\lambda_{t-1}^{(l)}}$, then with probability of at least $1 - \epsilon_{supp\text{-}pres}^{(l)}$, for $j \in supp(z^*)$, the code coefficient error is*

$$|z_{t,(j)}^{(l)} - z_{(j)}^*| \leq \mathcal{O}(\sqrt{s\|D_j^{(l)} - D_j^*\|_2} + e_{t,j}^{(l)\text{unroll,fixed}} + \lambda^{\text{fixed}}) \tag{10}$$

*and*

$$z_{T,(j)} = z_{(j)}^*(1 - \beta_j^{(l)}) + \zeta_{T,j}^{(l)} \tag{11}$$

*where $e_{t,j}^{(l)\text{unroll, fixed}} := (s-1)t\alpha\frac{\mu_l}{\sqrt{m}}\max_i |z_{0,(i)}^{(l)} - z_{(i)}^*|\delta_{\alpha,t-1}^{\text{fixed}} + |z_{0,(j)}^{(l)} - z_{(j)}^*|\delta_{\alpha,t}^{\text{fixed}}$, $\delta_{\alpha,t}^{\text{fixed}} := (1 - \alpha + \alpha\frac{\mu_l}{\sqrt{m}})^t$, $|\zeta_{T,j}^{(l)}| = \mathcal{O}(a_\gamma + \lambda^{\text{fixed}})$ with $a_\gamma = \mathcal{O}(\sqrt{s}\delta_l)$, $\beta_j^{(l)} = \langle D_j^* - D_j^{(l)}, D_j^* \rangle \leq \frac{\delta_l^2}{2}$, and $\epsilon_{supp\text{-}pres}^{(l)} := \epsilon_{supp\text{-}rec}^{(l)} + \epsilon_\gamma^{(l)} = 2p\exp\left(\frac{-C_{\min}^2}{\mathcal{O}^*(\delta_l^2)}\right) + 2s\exp\left(\frac{-1}{\mathcal{O}(\delta_l)}\right)$. With appropriately large $t$, $|z_{t,(j)}^{(l)} - z_{(j)}^*| = \mathcal{O}(\sqrt{s\|D_j^{(l)} - D_j^*\|_2} + \lambda^{\text{fixed}})$.*

Aside from code estimation where the network parameters (e.g., regularization and step size) are finely tuned according to support recovery and preservation conditions (Theorems 4.1 and 4.2), we provide a general upper bound on the error between the converged code and $z^*$; the bound can be decomposed into two terms of the dictionary error and the biased amplitude estimate of the code.

**Theorem 4.6** (Global forward pass code error). *Let $\hat{z}$ be the fixed-point of the encoder with iterations $z_{t+1} = \Phi(z_t, D)$. Given Assumption 3.6, Lemmas 3.2, A.1 and A.2, we have $\|\hat{z} - z^*\|_2 \leq \mathcal{O}(\|D - D^*\|_2 + \hat{\delta}^*)$, where $\hat{\delta}^* = \|\hat{z}^* - z^*\|_2$, $\hat{z}$ is the unique minimizer of lasso (1) given the dictionary $D$, $\hat{z}^*$ is the unique minimizer of lasso (1) given the dictionary $D^*$, and $z^*$ is the ground-truth code.*

This general decomposition is to emphasize that aside from the current estimate of the dictionary, the code error is a function of the forward pass algorithm used to solve the sparse coding problem. Specifically, the upper bound states that at the best scenario where there is access to the generating dictionary $D^*$, the forward pass solving lasso with fixed $\lambda$ still gives a biased amplitude estimate of $z^*$. Overall, the assumptions to get this bound are mild; the bound is valid independent of successful support recovery or data distribution. With incorporation of data distribution and conditions stated in Theorem 4.1 and Theorem 4.2, the upper bound $\hat{\delta}^*$ can be replaced with terms involving $\lambda$, and reaches at zero as $\lambda$ decays across forward iterations.

**Jacobian convergence and error**  Following properties similar to those used in Theorem 4.3, and assuming $J_t$ is bounded (Assumption 4.1), we show in Theorem 4.7 that, as the PUDLE unrolls, the code Jacobian $J_t$ converges to $\hat{J}$, the Jacobian of the solution of the lasso. The convergence of the Jacobian of proximal gradient descent is also studied in (Bertrand et al., 2021) for hyperparameter selection through implicit differentiation (Bengio, 2000), where the Jacobian is taken w.r.t to the hyperparameter $\lambda$ as opposed to $D$.

**Assumption 4.1** (Bounded Jacobian). *The Jacobian is bounded, i.e., $\exists M_J > 0$, s.t. $\|J_t\|_2 \leq M_J \ \forall t$.*

**Theorem 4.7** (Local forward pass Jacobian convergence). *Given the recursion $z_{t+1} = \Phi(z_t, D)$, and $\hat{z}$ the unique minimizer of lasso with Jacobian $\hat{J}$, then $\exists \rho < 1, B > 0$ s.t. $\|J_t - \hat{J}\|_2 \leq \mathcal{O}(t\rho^t) \ \forall t > B$. Furthermore, given Theorem 4.1 and Theorem 4.2, $B = 1$.*

The forward pass code and Jacobian convergences *after* support selection is similar to the results from (Malézieux et al., 2022). The highlights of our finding are that the order of upper bound convergences can be achieved from the first iteration of the encoder. In other words, we specify, in Theorem 4.1 and Theorem 4.2, the dictionary and data conditions such that the support can be recovered with $B = 1$. This resolves the instability issue discussed by Malézieux et al. (2022) in computation of the gradient $g_t^{\text{ae-lasso}}$ outside of the support. Finally, we show that the global Jacobian error is in the order of dictionary error.

**Theorem 4.8** (Global forward pass Jacobian error). *Let $\hat{z}$ be the fixed-point of the encoder with iterations $z_{t+1} = \Phi(z_t, D)$. Given Assumption 3.6, Lemmas 3.2, A.1 and A.2, we have $\|\hat{J} - J^*\|_2 \leq \mathcal{O}(\|D - D^*\|_2 + \hat{\delta}_J^*)$, where $\hat{\delta}_J^* := \|\hat{J}^* - J^*\|_2$, and $\hat{J}$, $\hat{J}^*$ and $J^*$ are Jacobians corresponding to $\hat{z}$, $\hat{z}^*$ and $z^*$.*

## 4.2  Backward pass

We show two results for local gradient $\hat{g}$ and global gradient $g^*$ convergence. The goal is not to provide a finite sample analysis but to emphasize the relative differences between the gradients in Algorithm 2. The impact of gradient error for parameter estimation in the convex setting has been studied by Devolder et al. (2013) indicating that the convergence to the parameter's neighbourhood is dictated by the gradient error (Devolder et al., 2013; 2014). As dictionary learning is a bi-convex problem, findings of Devolder et al. (2013) hold as well for better estimation of the local dictionary at every step of alternating minimization. Moreover, Arora et al. (2015), provided a detailed analysis of sparse coding and various gradient estimations for dictionary learning, showing that by computing a more accurate gradient at every step of the alternating minimization scheme, the dictionary estimates

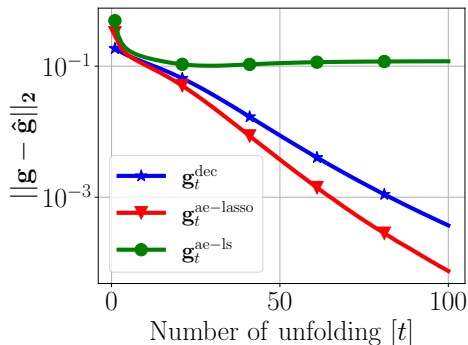

Figure 3: Convergence rate of gradients (Theorem 4.9).

converge to a closer neighbourhood of $\boldsymbol{D}^*$. Overall, the intuition is that the size of the gradient error dictates the size of the neighbourhood of the dictionary within which one can guarantee convergence. We argue that the method with lower gradient error recovers the dictionary better.

**Local gradient estimations** We highlight the effect of finite unrolling on the gradient for parameter estimation (Ablin et al., 2020). Theorem 4.9 shows the convergence rate of gradients to $\hat{\boldsymbol{g}}$, determining the similarity of PUDLE and Algorithm 1.

**Theorem 4.9** (Local convergence of gradients). *Given the forward pass convergence results (Theorems 4.3 and 4.7), $\exists\, \rho < 1, B > 0$ such that $\forall t > B$, the errors of gradients defined in Algorithm 2 w.r.t $\hat{\boldsymbol{g}}$ (4) satisfy*

$$
\begin{aligned}
\|\boldsymbol{g}_t^{dec} - \hat{\boldsymbol{g}}\|_2 &\leq \mathcal{O}(\rho^t) \\
\|\boldsymbol{g}_t^{ae\text{-}lasso} - \hat{\boldsymbol{g}}\|_2 &\leq \mathcal{O}(t\rho^{2t}) \\
\|\boldsymbol{g}_t^{ae\text{-}ls} - \hat{\boldsymbol{g}}\|_2 &\leq \mathcal{O}(t\rho^{2t} + M_J\lambda\sqrt{s}).
\end{aligned}
\tag{12}
$$

*Moreover, the order of upper bounds is tight (see Lemma A.4).*

First, upper bounds on the errors related to $\boldsymbol{g}_t^{\text{dec}}$ and $\boldsymbol{g}_t^{\text{ae-lasso}}$ go to zero as $t$ increases. Hence, both gradients converge to $\hat{\boldsymbol{g}}$. This means that asymptotically as $t$ increases, training PUDLE with $\boldsymbol{g}_t^{\text{dec}}$ and $\boldsymbol{g}_t^{\text{ae-lasso}}$ is equivalent to classical alternating-minimization (Algorithm 1). Second, as $t$ increases, $\boldsymbol{g}_t^{\text{ae-lasso}}$ has faster convergence than $\boldsymbol{g}_t^{\text{dec}}$. Lastly, $\boldsymbol{g}_t^{\text{ae-ls}}$ is a biased estimator of $\hat{\boldsymbol{g}}$ (Figure 3). The convergence results on the error $\|\boldsymbol{g}_t^{\text{ae-lasso}} - \hat{\boldsymbol{g}}\|_2$ is previously studied by Malézieux et al. (2022).

Given the above convergence results, one may conclude that $\boldsymbol{g}_t^{\text{ae-lasso}}$ should be used for dictionary recovery. However, we show next that for dictionary recovery, the gradient $\boldsymbol{g}_t^{\text{ae-lasso}}$, used by Malézieux et al. (2022), is indeed a biased estimator of the global gradient $\boldsymbol{g}^*$ for recovery of $\boldsymbol{D}^*$. We decrease this bias by replacing $\boldsymbol{g}_t^{\text{ae-lasso}}$ with $\boldsymbol{g}_t^{\text{ae-ls}}$ and show that $\boldsymbol{g}_t^{\text{ae-ls}}$ results in a better recovery of $\boldsymbol{D}^*$ than $\boldsymbol{g}_t^{\text{ae-lasso}}$.

**Global gradient estimations** Theorem 4.10 shows the global gradient errors w.r.t $\boldsymbol{g}^*$ from (5). We omit the gradient $\boldsymbol{g}_t^{\text{dec}}$, as it is asymptotically equivalent to $\boldsymbol{g}_t^{\text{ae-lasso}}$. We study the errors in the limit to unrolling, i.e., as $t \to \infty$. This determines which PUDLE gradients recover $\boldsymbol{D}^*$ better (Devolder et al., 2013; 2014).

**Theorem 4.10** (Global error of gradients). *Given the convergence results from the forward pass, (Theorems 4.6 and 4.8), the errors of gradients defined in Algorithm 2 w.r.t global direction $\boldsymbol{g}^*$ (defined in (5)) satisfy*

$$
\begin{aligned}
\|\boldsymbol{g}_\infty^{ae\text{-}lasso} - \boldsymbol{g}^*\|_2 &\leq \mathcal{O}(\|\boldsymbol{D} - \boldsymbol{D}^*\|_2^2 + \|\boldsymbol{D} - \boldsymbol{D}^*\|_2 + \hat{\delta}^* + \hat{\delta}_J^* + M_J\lambda\sqrt{s}) \\
\|\boldsymbol{g}_\infty^{ae\text{-}ls} - \boldsymbol{g}^*\|_2 &\leq \mathcal{O}(\|\boldsymbol{D} - \boldsymbol{D}^*\|_2^2 + \|\boldsymbol{D} - \boldsymbol{D}^*\|_2 + \hat{\delta}^* + \hat{\delta}_J^*).
\end{aligned}
\tag{13}
$$

Several factors affect the order of upper bounds: the current estimate of the dictionary, code amplitude-bias error due to $\ell_1$ norm, and the usage of $\ell_1$ norm in the loss used for backpropagation. To study the bias in the gradient computation, let consider the scenario where $\boldsymbol{D} = \boldsymbol{D}^*$. We denote those gradients by superscript $\boldsymbol{D}^*$. If the gradients are not biased, then the upper bounds should goes to zero. The gradient errors are

$$
\|\boldsymbol{g}_\infty^{\text{ae-lasso},\boldsymbol{D}^*} - \boldsymbol{g}^*\|_2 \leq \mathcal{O}(\hat{\delta}^* + \hat{\delta}_J^* + M_J\lambda\sqrt{s}) \qquad \text{and} \qquad \|\boldsymbol{g}_\infty^{\text{ae-ls},\boldsymbol{D}^*} - \boldsymbol{g}^*\|_2 \leq \mathcal{O}(\hat{\delta}^* + \hat{\delta}_J^*). \tag{14}
$$

For $\boldsymbol{g}_\infty^{\text{ae-ls},\boldsymbol{D}^*}$, the radius of the error ball is only a function of the amplitude error of the code estimated through lasso compare to the ground-truth code $\boldsymbol{z}^*$. However, the error ball for the gradient $\boldsymbol{g}_\infty^{\text{ae-lasso},\boldsymbol{D}^*}$ includes an additional term concerning the usage of lasso loss containing the regularization term $\lambda$. This implies that the $\boldsymbol{D}^*$ neighbourhood at which the gradient $\boldsymbol{g}_\infty^{\text{ae-ls},\boldsymbol{D}^*}$ is guaranteed to converge to is smaller than of the $\boldsymbol{g}_\infty^{\text{ae-lasso},\boldsymbol{D}^*}$ (Figure 4a). Implications of such gradient estimation are seen in dictionary learning where $\boldsymbol{g}_\infty^{\text{ae-ls}}$ recovers $\boldsymbol{D}^*$ better (Figures 4b and 4c). In Figure 4b, the encoder unrolls for $T = 25$, hence the phenomenon of implicit acceleration is seen in faster and better dictionary learning performance of $\boldsymbol{g}_\infty^{\text{ae-lasso}}$ than $\boldsymbol{g}_\infty^{\text{dec}}$. In Figure 4c where $T = 100$, similar performance of $\boldsymbol{g}_\infty^{\text{dec}}$ and $\boldsymbol{g}_\infty^{\text{ae-lasso}}$ illustrates their asymptotic equivalence as $t \to \infty$ (See Appendix for additional noisy dictionary learning experiments where the measurements $\boldsymbol{x}$ are corrupted with zero-mean Gaussian noise such that the Signal-to-Noise-Ratio is approximately 12 SNR; in this setting, the aforementioned comparative analysis still holds.)

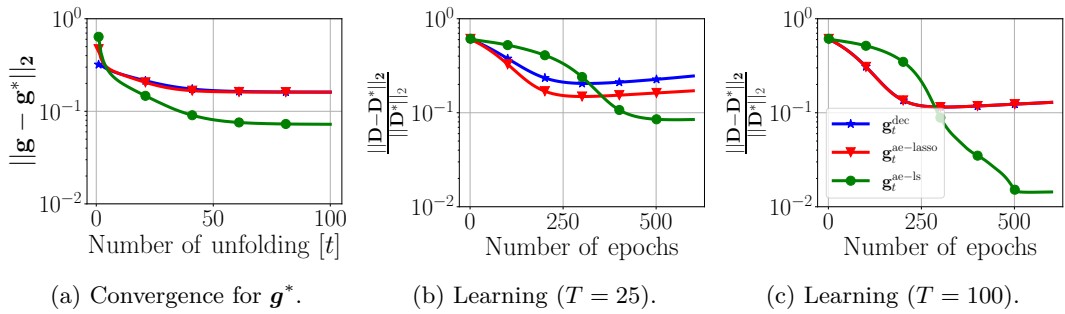

Figure 4: Results for PUDLE's global convergence (Theorem 4.10) and dictionary learning.

**Towards unbiased estimation** As long as $\lambda$ is fixed within PUDLE, all defined gradients remain biased estimators of $\boldsymbol{g}^*$, due to the biased estimate of the code $\boldsymbol{z}^*$ through $\ell_1$ norm. This bias exists while dictionary learning is performed strictly using lasso through Algorithm 1. Given the conditions on the regularizer in Theorem 4.2 which we discussed in Section 4.1 and the derived upper bounds in Theorem 4.10, we suggest the decaying of $\lambda$ across the encoder to reduce the gradient biases and improve dictionary learning. Next, we prove in Theorem 4.11 that PUDLE converges to $\boldsymbol{D}^*$ if $\lambda$ decays across the layers $t$ according to Theorem 4.4. Moreover, Theorem 4.12 proves that if $\lambda$ stays fixed according to Theorem 4.5, then PUDLE only guarantees to converge to a close neighbourhood of the dictionary. In these analyses, we focus on $\boldsymbol{g}_T^{\text{dec}}$. Furthermore, we show in Section 4.3 that by decaying $\lambda$ at each unrolled layer, the gradient bias vanishes, and we recover $\boldsymbol{D}^*$.

**Dictionary learning** Given the network parameters set by Theorem 4.4, Theorem 4.11 proves that using $\boldsymbol{g}_T^{\text{dec}}$, PUDLE recovers the dictionary; the dictionary error contracts at every update. Moreover, Theorem 4.12 proves that as long as $\lambda$ stays fixed across the unrolled layers, PUDLE guarantees to converge to only $\boldsymbol{D}^*$ neighbourhood characterized by the regularization parameter $\lambda$. These analyses requires for $\boldsymbol{D}^{(l)}$ to maintain a closeness to $\boldsymbol{D}^*$ which we provide a proof for in Lemma A.7. Hence, the dictionary closeness assumption (Assumption 3.5) stays valid.

**Theorem 4.11** (Dictionary learning with variable $\lambda_t$). *Given Assumptions 3.3 and 3.4, suppose $\boldsymbol{D}^{(l)}$ is $\mu_l$-incoherent and $(\delta_l, 2)$-close to $\boldsymbol{D}^*$ with $\delta_l = \mathcal{O}^*(1/\log p)$. If $s = \mathcal{O}(\sqrt{m})$, $\mu = \mathcal{O}(\log m)$, learning rate is $\eta = \mathcal{O}(\frac{p}{s(1-\delta_l^2/2)})$, and the regularizer and step size are set according to Theorem 4.4, then for any dictionary update $l$ using $\boldsymbol{g}_T^{dec}$, with probability of at least $1 - \epsilon_{supp\text{-}pres}^{(l)}$,*

$$\|\boldsymbol{D}_j^{(l+1)} - \boldsymbol{D}_j^*\|_2^2 \leq (1-\psi)\|\boldsymbol{D}_j^{(l)} - \boldsymbol{D}_j^*\|_2^2 \tag{15}$$

*where $\epsilon_{supp\text{-}pres}^{(l)} := \epsilon_{supp\text{-}rec}^{(l)} + \epsilon_\gamma^{(l)} = 2p \exp\left(\frac{-C_{\min}^2}{\mathcal{O}^*(\delta_l^2)}\right) + 2s \exp\left(\frac{-1}{\mathcal{O}(\delta_l)}\right)$.*

**Theorem 4.12** (Dictionary learning with fixed $\lambda_t$). *Given Assumptions 3.3 and 3.4, suppose $\boldsymbol{D}^{(l)}$ is $\mu_l$-incoherent and $(\delta_l, 2)$-close to $\boldsymbol{D}^*$ with $\delta_l = \mathcal{O}^*(1/\log p)$. If $s = \mathcal{O}(\sqrt{m})$, $\mu = \mathcal{O}(\log m)$, learning rate is $\eta = \mathcal{O}(\frac{p}{s(1-\delta_l^2/2)})$, and the regularizer $\lambda^{fixed}$ and step size are set according to Theorem 4.5, then for any dictionary update $l$ using $\boldsymbol{g}_T^{dec}$, with probability of at least $1 - \epsilon_{supp\text{-}pres}^{(l)}$,*

$$\|\boldsymbol{D}_j^{(l+1)} - \boldsymbol{D}_j^*\|_2^2 \leq (1-\psi)\|\boldsymbol{D}_j^{(l)} - \boldsymbol{D}_j^*\|_2^2 + \epsilon_\lambda^{(l)} \tag{16}$$

*where $\epsilon_\lambda^{(l)} := \eta \frac{2p}{s(1-\beta_j^{(l)})}\lambda^{fixed2}$, $\epsilon_{supp\text{-}pres}^{(l)} := \epsilon_{supp\text{-}rec}^{(l)} + \epsilon_\gamma^{(l)} = 2p \exp\left(\frac{-C_{\min}^2}{\mathcal{O}^*(\delta_l^2)}\right) + 2s \exp\left(\frac{-1}{\mathcal{O}(\delta_l)}\right)$.*

### 4.3 Experiments

**Dictionary learning** We focus on the performance of the best-performing gradient estimator $\boldsymbol{g}_t^{\text{ae-ls}}$, and compare it with NOODL (Rambhatla et al., 2018), a state-of-the-art online dictionary learning algorithm, and SPORCO (Wohlberg, 2017), an alternating-minimization dictionary learning algorithm that uses lasso. NOODL, which uses iterative hard-thresholding (HT) for sparse coding and a gradient update

employing the code's sign, has linear convergence upon proper initialization (Rambhatla et al., 2018). We note that the results from $\boldsymbol{g}_t^{\text{ae-lasso}}$ are not shown, as the gradient computation was unstable (Malézieux et al., 2022). We emphasize that our proposed gradient $\boldsymbol{g}_t^{\text{ae-ls}}$ does not suffer such instability. We train:

- $\boldsymbol{g}_t^{\text{ae-ls}}$: $\lambda$ is fixed across iterations.
- $\boldsymbol{g}_t^{\text{ae-ls, decay}}$ : $\lambda$ decays (i.e., $\lambda_t = \lambda \nu^t$, with $0 < \nu < 1$) where $\nu$ decreases as training progresses.
- $\boldsymbol{g}_t^{\text{ae-ls, HT}}$ : $\mathcal{P}_{\alpha\lambda}(v)$ is replaced with $\text{HT}_b(v) \triangleq v\mathbf{1}_{|v| \geq b}$.

With HT, the sparse coding step reduces to that from NOODL. In this case, we highlight the difference between the gradient update of our method (backpropagation) with NOODL. We focus on convergence, as $\eta$ across methods is not comparable.

Figure 5 shows the convergence of $\boldsymbol{D} \in \mathbb{R}^{1000 \times 1500}$ to $\boldsymbol{D}^*$ when the code is 20-sparse (for other sparsity levels and details see Appendix C). A biased estimate of the code amplitudes results in convergence only to a neighbourhood of the dictionary (Rambhatla et al., 2018). This is observed in the

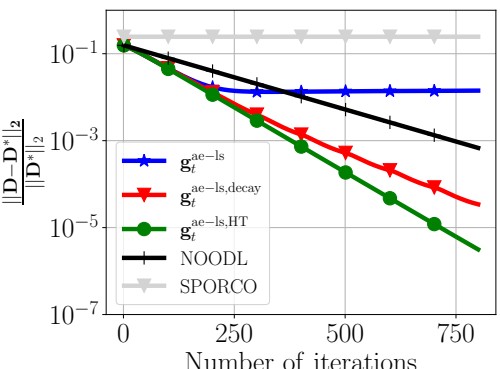

Figure 5: Dictionary convergences.

convergence of $\boldsymbol{g}_t^{\text{ae-ls}}$ and SPORCO (final error is shown). The convergence of $\boldsymbol{g}_t^{\text{ae-ls}}$ to a closer neighbourhood than SPORCO supports Theorem 4.10. Moreover, with decaying $\lambda$, the code bias vanishes, hence $\boldsymbol{g}_t^{\text{ae-ls, decay}}$ and $\boldsymbol{g}_t^{\text{ae-ls, HT}}$ converges to $\boldsymbol{D}^*$ similar to NOODL.

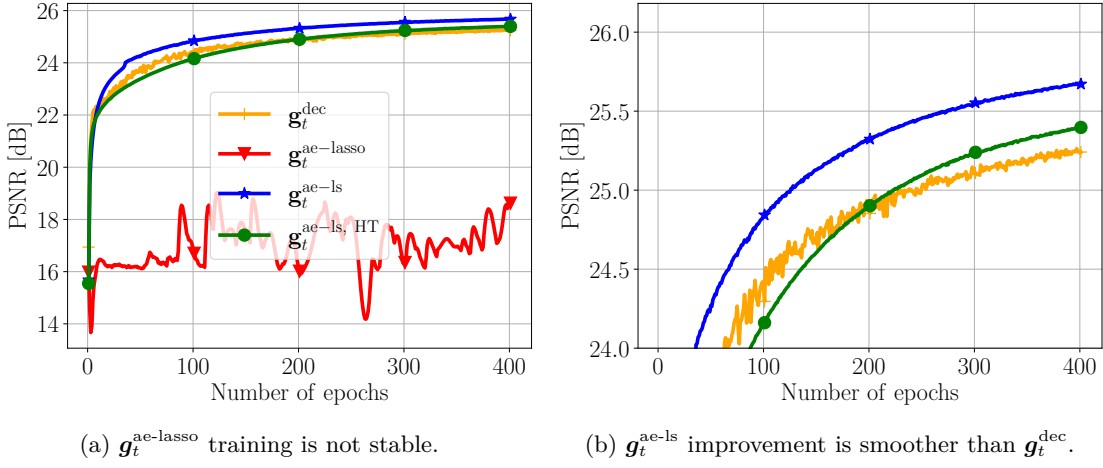

(a) $\boldsymbol{g}_t^{\text{ae-lasso}}$ training is not stable.

(b) $\boldsymbol{g}_t^{\text{ae-ls}}$ improvement is smoother than $\boldsymbol{g}_t^{\text{dec}}$.

Figure 6: Networks behaviour (test PSNR) during training as a function of epochs.

**Image denoising** To further highlight the advantage of $\boldsymbol{g}_t^{\text{ae-ls}}$ over the other gradients, we compare them in a supervised task of image denoising. In addition to $\boldsymbol{g}_t^{\text{ae-ls}}$, $\boldsymbol{g}_t^{\text{ae-lasso}}$, and $\boldsymbol{g}_t^{\text{dec}}$, we consider $\boldsymbol{g}_t^{\text{ae-ls, HT}}$ where the proximal operator is replaced with HT. This is to compare with sparse coding scheme of NOODL. We do not compare against NOODL's dictionary update, as this computation for two-dimensional convolutions is not straightforward. Prior works have shown that variants of PUDLE either rival or outperform state-of-the-art architectures (Simon & Elad, 2019; Tolooshams et al., 2020). Thus, we focus on a comparative analysis of the gradients. We trained on 432 and tested on 68 images from BSD (Martin et al., 2001). BSD dataset is a popular training dataset for denoising (Zhang et al., 2017a; Simon & Elad, 2019; Mohan et al., 2019). We used a convolutional dictionary and corrupted images with zero-mean Gaussian noise of standard deviation of 25 (see Appendix C for details). We initialized the dictionary filters by standard Normal distribution; this is to follow the norm in the deep learning literature and to demonstrate the practicality and usefulness of PUDLE in the absence of an initialization method. We evaluate the denoising performance of soft-thresholding using $\lambda$ and HT with $b$ in peak signal-to-noise-ratio (PSNR).

First, we highlight the stability of $\boldsymbol{g}_t^{\text{ae-ls}}$ against $\boldsymbol{g}_t^{\text{ae-lasso}}$; Figure 6a shows the network dynamics in terms of test PSNR as a function of epochs when $\lambda = 0.16$ for $\boldsymbol{g}_t^{\text{ae-ls}}$, $\boldsymbol{g}_t^{\text{ae-lasso}}$, $\boldsymbol{g}_t^{\text{dec}}$ and $b = 0.05$ for $\boldsymbol{g}_t^{\text{ae-ls, HT}}$. We observed that $\boldsymbol{g}_t^{\text{ae-ls}}$ uses full backpropagation and stays stable. However, the training with $\boldsymbol{g}_t^{\text{ae-lasso}}$ is not stable and unstable to perform denoising where the noisy PSNR is approximately 20 dB (Malézieux et al., 2022). Second, Figure 6b shows that compared to $\boldsymbol{g}_t^{\text{dec}}$, the backpropagated gradients result in a smoother improvement during training. Moreover, Table 1 shows that the advantage of $\boldsymbol{g}_t^{\text{ae-ls}}$ over $\boldsymbol{g}_t^{\text{dec}}$ is not limited to dictionary learning and is seen in denoising. We have excluded the results for $\boldsymbol{g}_t^{\text{ae-lasso}}$ from Table 1 as the network failed to denoise (see Figure 6a). Additionally, the superior performance of $\boldsymbol{g}_t^{\text{ae-ls}}$ compared to $\boldsymbol{g}_t^{\text{ae-ls, HT}}$ highlights the benefits of PUDLE (i.e., $\ell_1$-based unrolling) against HT used in NOODL.

Table 1: Denoising of BSD68. Reported numbers are mean (std) PSNR given three independent trials.

| METHOD | | PSNR [dB] | | | |
|---|---|---|---|---|---|
| | $\lambda$ | 0.08 | 0.12 | 0.16 | 0.2 |
| $\boldsymbol{g}_t^{\text{dec}}$ | | 24.21 (0.12) | 24.93 (0.14) | 25.25 (0.06) | 24.88 (0.00) |
| $\boldsymbol{g}_t^{\text{ae-ls}}$ | | 24.79 (0.03) | 25.43 (0.03) | **25.63** (0.04) | 25.46 (0.05) |
| | $b$ | 0.02 | 0.05 | 0.08 | 0.1 |
| $\boldsymbol{g}_t^{\text{ae-ls, HT}}$ | | 22.92 (0.07) | 25.26 (0.1) | 24.76 (0.06) | 23.94 (0.13) |

## 5 Interpretable Sparse Codes and Dictionary

One motivation behind using algorithm unrolling to design deep architectures is interpretability (Monga et al., 2019); they argue that the designed networks are interpretable as they capture domain knowledge via an optimization model. For example, Tolooshams et al. (2021a) takes advantage of the interpretability of learned weights in an unrolled dictionary learning network to solve spike sorting, an unsupervised source separation problem in computational neuroscience. Moreover, Kim et al. (2010) uses sparse coding to learn interpretable representations of human motions. However, none of the existing methods in the literature provide interpretability results that open the black-box network through building a mathematical relation between the learned dictionary, training data, and test representation/reconstruction. This section analyzes the interpretability of the unrolled sparse coding method in this context. We note that such mathematical relation and interpretability results also hold for dictionary learning. However, it is missing in the literature, irrespective of whether one uses an unrolling network. We provide the following theorem.

**Theorem 5.1** (Interpretable unrolled network). *Consider the dictionary learning optimization of the form $\min_{\boldsymbol{Z},\boldsymbol{D}} \frac{1}{2}\|\boldsymbol{X} - \boldsymbol{D}\boldsymbol{Z}\|_F^2 + \lambda\|\boldsymbol{Z}\|_1 + \omega/2\|\boldsymbol{D}\|_F^2$, where $\boldsymbol{X} = [\boldsymbol{x}^1, \boldsymbol{x}^2, \ldots, \boldsymbol{x}^n] \in \mathbb{R}^{m \times n}$ and $\boldsymbol{Z} = [\boldsymbol{z}^1, \boldsymbol{z}^2, \ldots, \boldsymbol{z}^n] \in \mathbb{R}^{p \times n}$. Let $\tilde{\boldsymbol{Z}}$ be the given converged sparse codes, then stationary points of the problem w.r.t the network weights (dictionary) follows $\tilde{\boldsymbol{D}} = \boldsymbol{X}\boldsymbol{G}^{-1}\tilde{\boldsymbol{Z}}^T$, where we denote $\boldsymbol{G} := (\tilde{\boldsymbol{Z}}^T\tilde{\boldsymbol{Z}} + \omega\boldsymbol{I})la$.*

**The dictionary interpolates the training data** Given Theorem 5.1, each learned atom interpolates the training data, i.e.,

$$\tilde{\boldsymbol{D}}_j = \boldsymbol{X}(\boldsymbol{G}^{-1}\boldsymbol{w}_j) = \sum_{k=1}^{n}(\boldsymbol{G}^{-1}\boldsymbol{w}_j)_k\boldsymbol{x}^k \tag{17}$$

where $\boldsymbol{w}_j = [\tilde{\boldsymbol{z}}_j^1, \tilde{\boldsymbol{z}}_j^2, \ldots, \tilde{\boldsymbol{z}}_j^n]^{\text{T}} \in \mathbb{R}^n$ is a vector containing the training code activity for dictionary atom $j$. Specifically, the importance of training image $\boldsymbol{x}^k$ in learning dictionary atom $j$ is captured by the term $(\boldsymbol{G}^{-1}\boldsymbol{w}_j)_k$. This proves the dictionary lives in the spans of the

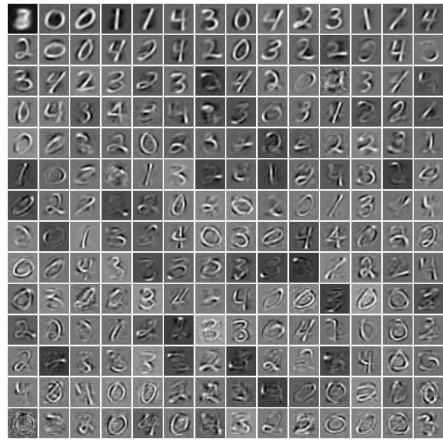

Figure 7: Fraction of dictionary atoms learned from $\{0, 1, 2, 3, 4\}$ MNIST.

training set. Given the small number of atoms compared to the training size, (17) shows that the dictionary summarizes the training examples. We trained the network on digits of $\{0, 1, 2, 3, 4\}$ MNIST (Figure 7 shows a fraction of the most used learned atoms). Figure 8 visualizes dictionary atoms along with training images with the highest contribution (green) and the lowest contribution (red). In addition, we used (17) on the partial training data to reconstruct learned atoms (shown as Estimate). Next, we interpret the relation between a new data to the training data using representer point selection, similar to (Yeh et al., 2018).

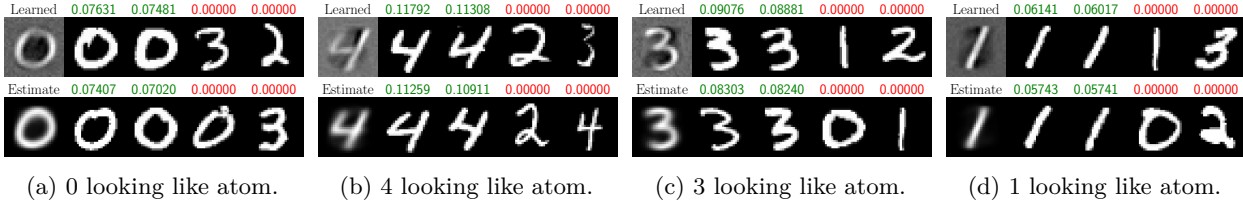

(a) 0 looking like atom.  (b) 4 looking like atom.  (c) 3 looking like atom.  (d) 1 looking like atom.

Figure 8: Training image contributions to learning the dictionary.

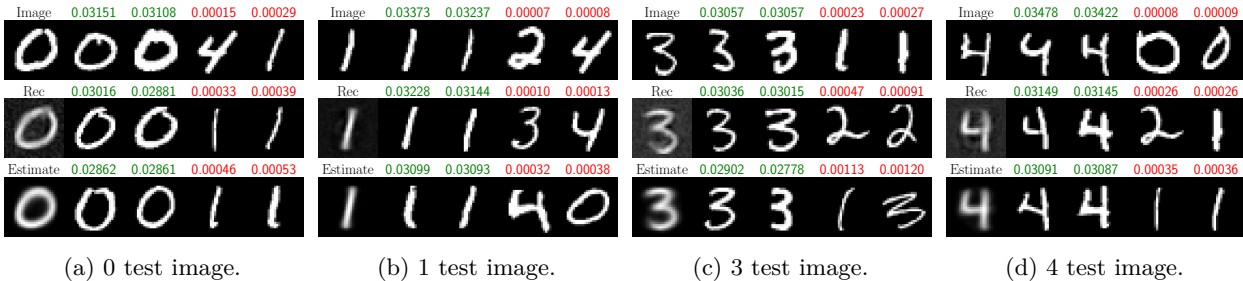

(a) 0 test image.  (b) 1 test image.  (c) 3 test image.  (d) 4 test image.

Figure 9: Interpolation of training data to reconstruct a new image. Contribution of training images are shown in green (high contribution) and red (low contribution). $\boldsymbol{\beta}^j$ is normalized over the used examples.

**Relation between new test image and training data**  For representation of a new data, we observe that the reconstruction of a new example $\boldsymbol{x}^j$ is a linear combination of all the training examples, i.e.,

$$\hat{\boldsymbol{x}}^j = \tilde{\boldsymbol{D}}\hat{\boldsymbol{z}}^j = \boldsymbol{X}\boldsymbol{\beta}^j = \sum_{k=1}^{n} \beta_k^j \boldsymbol{x}^k \tag{18}$$

where $\hat{\boldsymbol{x}}^j$ denotes reconstruction, $\hat{\boldsymbol{z}}^j$ is the code estimate, $\boldsymbol{\beta}^j = \boldsymbol{G}^{-1}\tilde{\boldsymbol{Z}}^{\mathrm{T}}\hat{\boldsymbol{z}}^j \in \mathbb{R}^n$, and $\beta_k^j = \sum_{a=1}^{n} \boldsymbol{G}_{ka}^{-1}\langle\tilde{\boldsymbol{z}}^a, \hat{\boldsymbol{z}}^j\rangle$. We observe that the contribution of image $k$ into the reconstruction of the test image is a function of $\beta_k^j$, and the energy of $\beta_k^j$ itself depends on the whole training set, and $\boldsymbol{G}^{-1}$. (18) shows how each image is reconstructed as interpolation of the training images. Figure 9 shows this results, where images with high (green) $\beta_k^j$ contribution are similar to the test image and those with low (red) $\beta_k^j$ contribution are different. In addition, we can evaluate the overall quality of the reconstruction by looking into $\beta_k^j$ in (18). For example, we observed that for test MNIST, unnormalized $\beta_k^j$ corresponding to high contributing training images is above 1. However, for resized-CIFAR, unnormalized $\beta_k^j$ of high contributing training images are often half or an order of magnitude lower than the MNIST case. This informs us of a bad representation/reconstruction of CIFAR image by the trained network. From another perspective, we can write the new image as

$$\boldsymbol{x}^j = \tilde{\boldsymbol{D}}\hat{\boldsymbol{z}}^j = \sum_{k=1}^{n}(\boldsymbol{X}\boldsymbol{G}^{-1})_k\langle\tilde{\boldsymbol{z}}^k, \hat{\boldsymbol{z}}^j\rangle \tag{19}$$

i.e., the contribution of each training image for reconstruction is a function of their code similarity to the new image and properties of the Gram matrix of training set code similarities. Specifically, the relation rules the contribution of transformed image $k$ (i.e., $(\boldsymbol{X}\boldsymbol{G}^{-1})_k$) into reconstruction of the test image as a function of its code similarity $\langle\tilde{\boldsymbol{z}}^k, \hat{\boldsymbol{z}}^j\rangle$. In other words, (19) shows that training images with the highest code similarity to the representation of the new image have the highest contribution to its reconstruction. This interpretation is demonstrated in Figure 10. The training images with the highest code similarity (green) and the lowest

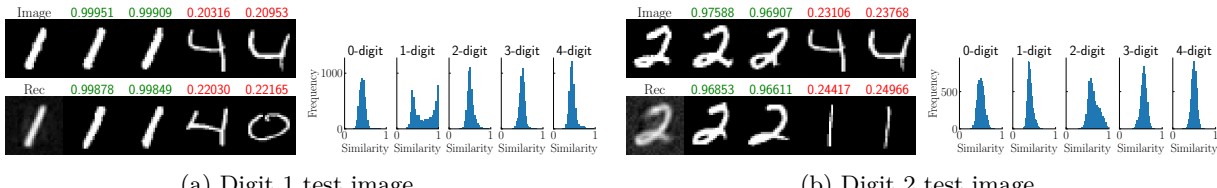

(a) Digit 1 test image.  (b) Digit 2 test image.

Figure 10: Contribution of images with code similarity into reconstruction of a new test image along with the histograms of the similarity of the test code to training codes from each class.

similarity (red) are shown. In addition, the figure demonstrates the histogram of the code similarity between the test image and the training set, grouped by their class digit. For example, for digit 1 test image, its code similarity to train images from class 1 are bimodal. This corresponds to 1 digits that are tilted to the left (low similarity) and right (high similarity). Moreover, for digit 2 test image, we observe that the histogram of images corresponding to digit 2 are shifted the most to the right (highest similarity) than the other classes.

## 6 Conclusions

This paper studied dictionary learning and analyzed the dynamics of unrolled sparse coding networks through a provable unrolled dictionary learning (PUDLE) framework. First, we provided a theoretical analysis of the forward pass for code recovery and dictionary learning. We discussed the bias introduced by $\ell_1$-based sparse coding in the forward pass, and how this affects the dictionary estimate in the backward pass. Second, we showed strategies to mitigate the propagation of this code bias into the backward pass; this is achieved by modification of the training loss function. We demonstrated that this bias could be further reduced and eliminated by decaying the regularization parameter within the unrolled layers. Additionally, we provided sufficient conditions on the data distribution and network to guarantee stability of backpropagated gradient computations. In the absence of such conditions, we proposed a modification to the loss function that resolves the gradient explosion and allows stable learning. In an image denoising task, we showed PUDLE outperforms the NOODL sparse coding scheme (Rambhatla et al., 2018). Motivated by interpretability as a popular feature for unrolled networks, we derived a mathematical relation between the network weights (dictionary) and the training set. We proved that the network weights live in the span of the training set, and constructed a relation between predictions of new input examples and the training set. The latter allows the user to extract images from the training set that are similar/dissimilar to the input image in representation/reconstruction.

### Acknowledgments

Demba Ba and Bahareh Tolooshams acknowledge the partial support of the ARO under grant number W911NF-16-1-0368. This is part of a collaboration between US DOD, UK MOD and UK Engineering and Physical Research Council (EPSRC) under the Multidisciplinary University Research Initiative. The authors would also like to thank the reviewers for their feedback, led to the improvement of the manuscript.

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

# A    Appendix - proofs

## A.1    Notation

Bold-lower-case and upper-case letters refer to vectors $\boldsymbol{d}$ and matrices $\boldsymbol{D}$. We use $\boldsymbol{d}_{(j)}$ to denote the $j^{th}$ element of the vector $\boldsymbol{d}$, and $\boldsymbol{D}_j$ is the $j^{th}$ column of the matrix $\boldsymbol{D}$. We denote the code estimate at unrolled layer $t$ by $\boldsymbol{z}_t$. $\lambda > 0$ is the regularization (sparsity-enforcing) parameter. $\sigma_{\max}(\boldsymbol{D})$ is the maximum singular value of $\boldsymbol{D}$. When taking the derivatives or norms w.r.t the matrix $\boldsymbol{D}$, we assume that $\boldsymbol{D}$ is vectorized. $\nabla_1 \mathcal{L}_{\boldsymbol{x}}(\boldsymbol{z}, \boldsymbol{D})$ and $\nabla_2 \mathcal{L}_{\boldsymbol{x}}(\boldsymbol{z}, \boldsymbol{D})$ are the first derivatives of the loss w.r.t $\boldsymbol{z}$ and $\boldsymbol{D}$, respectively. $\nabla_{11}^2 \mathcal{L}_{\boldsymbol{x}}(\boldsymbol{z}, \boldsymbol{D})$ is the second derivative of the loss w.r.t $\boldsymbol{z}$. $\nabla_{21}^2 \mathcal{L}_{\boldsymbol{x}}(\boldsymbol{z}, \boldsymbol{D})$ is the derivative of $\nabla_1 \mathcal{L}_{\boldsymbol{x}}(\boldsymbol{z}, \boldsymbol{D})$ w.r.t $\boldsymbol{D}$. The support of $\boldsymbol{z}$ is $\mathrm{supp}(\boldsymbol{z}) \triangleq \{j \colon \boldsymbol{z}_{(j)} \neq 0\}$.

## A.2    Basic definitions and Lemmas

We list four definitions used throughout the paper below.

**Definition A.1** ($\mu$-incoherence)**.** $\boldsymbol{D}$ *is $\mu$-incoherent, i.e., for every pair $(i,j)$ of columns, $|\langle \boldsymbol{D}_i, \boldsymbol{D}_j \rangle| \leq \mu/\sqrt{m}$.*

**Definition A.2** (($\delta, \kappa$)-closeness)**.** *Dictionary $\boldsymbol{D}$ is $\delta$-close to $\boldsymbol{D}^*$, i.e., there is a permutation $\pi$ and sign flip operator $u$ such that $\forall i \ \|u(i)\boldsymbol{D}_{\pi(i)} - \boldsymbol{D}_i^*\|_2 \leq \delta$. Additionally, $\|\boldsymbol{D} - \boldsymbol{D}^*\|_2 \leq \kappa \|\boldsymbol{D}^*\|_2$.*

**Definition A.3** (Lipschitz function)**.** *A function $f \colon \mathbb{R}^m \to \mathbb{R}^p$ is L-Lipschitz w.r.t a norm $\|\cdot\|$ if $\exists \ L > 0 \ s.t. \ \|f(a) - f(b)\| \leq L\|a - b\| \ \forall a, b \in \mathbb{R}^m$.*

**Definition A.4** (Lipschitz differentiable function)**.** *A twice differentiable function $f \colon \mathbb{R}^m \to \mathbb{R}^p$ is L-Lipschitz differentiable w.r.t a norm $\|\cdot\|$ iff $\exists \ L > 0 \ s.t. \ \|\nabla^2 f(a)\| \leq L \ \forall a \in \mathbb{R}^m$.*

**Definition A.5** (Strong convexity)**.** *A twice differentiable function $f \colon \mathbb{R}^m \to \mathbb{R}^p$ is strongly convex if $\exists \ \mu > 0 \ s.t. \ \nabla^2 f(a) \succeq \mu \boldsymbol{I}$.*

**Definition A.6** (Norm of subgradient). *For norms involving subgradents, we define* $\|\partial h(\boldsymbol{z})\| := \max_{\boldsymbol{v} \in \partial h(\boldsymbol{z})} \|\boldsymbol{v}\|$.

In the proof of the theorems, we use the strong convexity of the reconstruction loss after support selection and the bounded property of the Lipschitz mapping stated below.

**Lemma A.1** (Strong convexity of reconstruction loss). *Given the support selection (Proposition 4.1), $\boldsymbol{D}_S^T \boldsymbol{D}_S$ is full-rank. Thus, $\forall t > B, \mathcal{L}_{\boldsymbol{x}}(\boldsymbol{z}_t, \boldsymbol{D}) = \mathcal{L}_{\boldsymbol{x}}(\boldsymbol{z}_{t,S}, \boldsymbol{D}_S)$ is strongly convex (Definition A.5) in $\boldsymbol{z}$.*

**Lemma A.2** (Lipschitz mapping). *Given the recursion $\boldsymbol{z}_{t+1} = \Phi(\boldsymbol{z}_t) = \mathcal{P}_{\alpha\lambda}(\boldsymbol{z}_t - \alpha\nabla_1\mathcal{L}_{\boldsymbol{x}}(\boldsymbol{z}_t, \boldsymbol{D}))$, from Lemma A.1, there exist $B > 0$ such that loss $\mathcal{L}_{\boldsymbol{x}}(\boldsymbol{z}_t, \boldsymbol{D})$ is $\mu$-strongly convex $\forall t > B$. Hence, using Lemma 3.4,*

$$\|\nabla_1\Phi(\boldsymbol{z}_t, \boldsymbol{D})\|_2 = \|(\boldsymbol{I} - \alpha\nabla_{11}^2\mathcal{L}_{\boldsymbol{x}}(\boldsymbol{z}_t, \boldsymbol{D}))\partial\mathcal{P}_{\alpha\lambda}(\boldsymbol{z}_t)\|_2 \leq \rho \tag{20}$$

*where $\rho \triangleq c_{prox}(1 - \alpha\mu) < 1$.*

One key term, used in the proofs, is that $\boldsymbol{0} \in \nabla_1\mathcal{L}_{\boldsymbol{x}}(\hat{\boldsymbol{z}}, \boldsymbol{D}) + \partial h(\hat{\boldsymbol{z}})$ which is followed by the lasso optimality, i.e.,

**Lemma A.3** (Lasso optimality). *Lasso Karush-Kuhn-Tucker (KKT) optimality conditions are*

$$\hat{\boldsymbol{z}} \in \arg\min_{\boldsymbol{z} \in \mathbb{R}^p} f_{\boldsymbol{x}}(\boldsymbol{z}, \boldsymbol{D}) \Leftrightarrow \boldsymbol{D}^T(\boldsymbol{x} - \boldsymbol{D}\hat{\boldsymbol{z}}) \in \lambda\partial\|\hat{\boldsymbol{z}}\|_1, \text{ and } \partial|\hat{\boldsymbol{z}}_{(j)}| = \begin{cases} \{sign(\hat{\boldsymbol{z}}_{(j)})\} & \text{if } \hat{\boldsymbol{z}}_{(j)} \neq 0 \\ [-1, 1] & \text{if } \hat{\boldsymbol{z}}_{(j)} = 0 \end{cases}, \forall j \in \{1, 2, \ldots, p\}. \tag{21}$$

### A.3 Forward pass proof details

Given the $\mu$-incoherence of $\boldsymbol{D}^*$, and current dictionary closeness of $\delta_l$, we re-state Lemma 3.1 and proof it below. It shows that the current dictionary is $\mu_l$-close to $\boldsymbol{D}^*$.

**Lemma 3.1** ($\mu_l$-incoherent). *$\boldsymbol{D}^{(l)}$ is $\mu_l$-incoherent where $\mu_l = \mu + 2\sqrt{m}\delta_l$.*

*Proof.*

$$\langle\boldsymbol{D}_i^{(l)}, \boldsymbol{D}_j^{(l)}\rangle = \langle\boldsymbol{D}_i^*, \boldsymbol{D}_j^*\rangle - \langle\boldsymbol{D}_i^* - \boldsymbol{D}_i^{(l)}, \boldsymbol{D}_j^*\rangle - \langle\boldsymbol{D}_i^{(l)}, \boldsymbol{D}_j^* - \boldsymbol{D}_j^{(l)}\rangle$$
$$|\langle\boldsymbol{D}_i^{(l)}, \boldsymbol{D}_j^{(l)}\rangle| \leq \mu/\sqrt{m} + \|\boldsymbol{D}_i^* + \boldsymbol{D}_i^{(l)}\|_2\|\boldsymbol{D}_j^*\|_2 + \|\boldsymbol{D}_i^{(l)}\|_2\|\boldsymbol{D}_j^* - \boldsymbol{D}_j^{(l)}\|_2 \leq \mu/\sqrt{m} + 2\delta_l \tag{22}$$

$\blacksquare$

We re-state and proof the forward pass support recovery (Theorem 4.1). This shows that given proper initialization and under mild conditions, the support of the true code $\boldsymbol{z}^*$ is recovered with high probability in one iteration of the encoder.

**Theorem 4.1** (Forward pass support recovery). *Given Assumptions 3.3 and 3.4, suppose $\boldsymbol{D}^{(l)}$ is $\delta_l = \mathcal{O}^*(1/\sqrt{\log p})$ close to $\boldsymbol{D}^*$. If $s = \mathcal{O}^*(\sqrt{m}/\mu \log m)$, and $\mu = \mathcal{O}(\log m)$, then with probability of at least $1 - \epsilon_{supp\text{-}rec}^{(l)}$, the choice of $\lambda_0 = C_{\min}/4$ recovers the support of the code $\boldsymbol{z}^*$ in one encoder iteration, i.e., $sign(\mathcal{S}_{\alpha\lambda_0}(\alpha\boldsymbol{D}^{(l)T}\boldsymbol{x}) = sign(\boldsymbol{z}^*)$, where $\epsilon_{supp\text{-}rec}^{(l)} = 2p\exp\left(-\frac{C_{\min}^2}{\mathcal{O}^*(\delta_l^2)}\right)$.*

*Proof.* The code estimate after one iteration is $\boldsymbol{z}_1 = \mathcal{P}_{\alpha\lambda}(\alpha\boldsymbol{D}^{(l)T}\boldsymbol{x}) = sign(\boldsymbol{D}^{(l)T}\boldsymbol{x})\text{ReLU}(\alpha(|\boldsymbol{D}^{(l)T}\boldsymbol{D}^*\boldsymbol{z}^*| - \lambda_0))$. We focus on the positive entries. The analysis for negative entries is similar. Writting the relation for $i$-th entry,

$$\boldsymbol{z}_{1,(i)} = sign(\boldsymbol{D}^{(l)T}\boldsymbol{x})\text{ReLU}(\alpha(\sum_{j \in S^*}\langle\boldsymbol{D}_i^{(l)}, \boldsymbol{D}_j^*\rangle\boldsymbol{z}_{(j)}^* - \lambda_0))$$
$$= \text{ReLU}(\alpha(\langle\boldsymbol{D}_i^{(l)}, \boldsymbol{D}_i^*\rangle\boldsymbol{z}_{(i)}^* + \sum_{j \in S^*\backslash\{i\}}\langle\boldsymbol{D}_i^{(l)}, \boldsymbol{D}_j^*\rangle\boldsymbol{z}_{(j)}^* - \lambda_0)) \tag{23}$$

We focus on the term inside ReLU and discard $\alpha$, shared by all terms. We shows that under proper choice of $\lambda_0$, $\langle\boldsymbol{D}_i^{(l)}, \boldsymbol{D}_i^*\rangle\boldsymbol{z}_{(i)}^*$ is greater than $\lambda_0$ and $\boldsymbol{v}_i = \sum_{j \in S^*\backslash\{i\}}\langle\boldsymbol{D}_i^{(l)}, \boldsymbol{D}_j^*\rangle\boldsymbol{z}_{(j)}^*$ is small with respect to $\lambda_0$, hence

getting cancelled by ReLU. The small value of $\boldsymbol{v}_i$, compared to $\langle \boldsymbol{D}_i^{(l)}, \boldsymbol{D}_i^* \rangle \boldsymbol{z}_{(i)}^*$, results in $\text{sign}(\boldsymbol{D}^{(l)\text{T}}\boldsymbol{x})$ be equal to the $\text{sign}(\boldsymbol{D}^{(l)\text{T}}\boldsymbol{D}^*\boldsymbol{z}^*)$ which is equal to the sign of $\boldsymbol{z}^*$.

Given the current dictionary distance $\|\boldsymbol{D}_i^{(l)} - \boldsymbol{D}_i^*\|_2 \leq \delta_l$, we can find a lower bound on $\langle \boldsymbol{D}_i^{(l)}, \boldsymbol{D}_i^* \rangle \boldsymbol{z}_{(i)}^*$ as follows

$$\langle \boldsymbol{D}_i^{(l)}, \boldsymbol{D}_i^* \rangle = \frac{1}{2}(\|\boldsymbol{D}_i^*\|_2^2 + \|\boldsymbol{D}_i^{(l)}\|_2^2 - \|\boldsymbol{D}_i^{(l)} - \boldsymbol{D}_i^*\|_2^2) = 1 - \frac{1}{2}\|\boldsymbol{D}_i^{(l)} - \boldsymbol{D}_i^*\|_2^2$$

$$|\langle \boldsymbol{D}_i^{(l)}, \boldsymbol{D}_i^* \rangle| \geq 1 - \delta_l^2/2 \tag{24}$$

Hence, for $i \in S^*$

$$|\langle \boldsymbol{D}_i^{(l)}, \boldsymbol{D}_i^* \rangle \boldsymbol{z}_{(i)}^*| \geq (1 - \delta_l^2/2)C_{\min} \tag{25}$$

otherwise, it is 0. Given, $var(\boldsymbol{z}_{(i)}^*) = 1$ for $i \in S^*$, we find an upper bound on the variance $\boldsymbol{v}_i$ of as follows

$$var(\boldsymbol{v}_i) = \sum_{j \in S^* \setminus \{i\}} \langle \boldsymbol{D}_i^{(l)}, \boldsymbol{D}_j^* \rangle^2 = \sum_{j \in S^* \setminus \{i\}} (\langle \boldsymbol{D}_i^*, \boldsymbol{D}_j^* \rangle + \langle \boldsymbol{D}_i^{(l)} - \boldsymbol{D}_i^*, \boldsymbol{D}_j^* \rangle)^2$$

$$\leq \sum_{j \in S^* \setminus \{i\}} 2(\langle \boldsymbol{D}_i^*, \boldsymbol{D}_j^* \rangle^2 + \langle \boldsymbol{D}_i^{(l)} - \boldsymbol{D}_i^*, \boldsymbol{D}_j^* \rangle^2) \leq \sum_{j \in S^* \setminus \{i\}} (2\mu^2/m) + 2\|(\boldsymbol{D}_i^{(l)} - \boldsymbol{D}_i^*)^\text{T} \boldsymbol{D}_{S^* \setminus \{i\}}^*\|_2^2 \tag{26}$$

$$\leq (2s\mu^2/m) + 2\|(\boldsymbol{D}_i^{(l)} - \boldsymbol{D}_i^*)\|_2^2 \|\boldsymbol{D}_{S^* \setminus \{i\}}^*\|_2^2 \leq 2(s\mu^2/m + 4\delta_l^2) = \mathcal{O}^*(\delta_l^2)$$

where we used the Gershgorin Circle Theorem for the bound $\|\boldsymbol{D}_{S^* \setminus \{i\}}^*\|_2 \leq 2$. With the sub-Gaussian assumption on the coefficients $\boldsymbol{z}^*$, we get the following using Chernoff bound concerning $\boldsymbol{v}_i$.

$$P(|\boldsymbol{v}_i| \geq \frac{C_{\min}}{4}) \leq 2\exp\left(-\frac{C_{\min}^2}{4s\mu^2/m + 16\delta_l^2}\right) = 2\exp\left(-\frac{C_{\min}^2}{\mathcal{O}^*(\delta_l^2)}\right) \tag{27}$$

Taking a union bound over all indices $i \in [1, p]$ will result in

$$P(\max_i |\boldsymbol{v}_i| \geq \frac{C_{\min}}{4}) \leq 2p\exp\left(-\frac{C_{\min}^2}{\mathcal{O}^*(\delta_l^2)}\right) := \epsilon_{\text{supp-rec}}^{(l)} \tag{28}$$

Hence, we can set $\lambda_0 = C_{\min}/2$. $\blacksquare$

We re-state and prove the forward pass support preservation (Theorem 4.2).

**Theorem 4.2** (Forward pass support preservation). *Given Assumptions 3.3 and 3.4, suppose $\boldsymbol{D}^{(l)}$ is $\delta_l = \mathcal{O}^*(1/\log p)$ close to $\boldsymbol{D}^*$. If $s = \mathcal{O}^*(\sqrt{m}/\mu \log m)$, $\mu = \mathcal{O}(\log m)$, and the regularizer and step size are chosen such that $\lambda_t^{(l)} = \frac{\mu_l}{\sqrt{m}}\|\boldsymbol{z}^* - \boldsymbol{z}_t\|_1 + a_\gamma = \boldsymbol{\Omega}(\frac{s\log m}{\sqrt{m}})$ and $\alpha^{(l)} \leq 1 - \frac{2\lambda_t^{(l)} - (1 - \frac{\delta_l^2}{2})C_{\min}}{\lambda_{t-1}^{(l)}}$, then with probability of at least $1 - \epsilon_{supp-pres}^{(l)}$, the support, recovered at the first iteration, is preserved through the encoder iterations. We have $a_\gamma = \mathcal{O}(\sqrt{s}\delta_l)$ and $\epsilon_{supp-pres}^{(l)} := \epsilon_{supp-rec}^{(l)} + \epsilon_\gamma^{(l)} = 2p\exp\left(\frac{-C_{\min}^2}{\mathcal{O}^*(\delta_l^2)}\right) + 2s\exp\left(\frac{-1}{\mathcal{O}(\delta_l)}\right)$.*

*Proof.* Given current dictionary $\boldsymbol{D}^{(l)}$, in each iteration of the forward pass, we have $\boldsymbol{z}_{t+1} = \mathcal{P}_{\alpha\lambda}(\boldsymbol{z}_t + \alpha\boldsymbol{D}^\text{T}(\boldsymbol{D}^*\boldsymbol{z}^* - \boldsymbol{D}\boldsymbol{z}_t))$. We focus on the entires that are non-negative. Then procedure for negative code entries is similar. We follow similar steps as in (Rambhatla et al., 2018). We get

$$\boldsymbol{z}_{t+1,(j)} = \text{ReLU}((\boldsymbol{I} - \alpha\boldsymbol{D}^{(l)\text{T}}\boldsymbol{D}^{(l)})_{(j,:)}\boldsymbol{z}_t + \alpha(\boldsymbol{D}^{(l)\text{T}}\boldsymbol{D}^*)_{(j,:)}\boldsymbol{z}^* - \alpha\lambda_{t,j}^{(l)})$$

$$= \text{ReLU}((\boldsymbol{I} - \alpha\boldsymbol{D}^{(l)\text{T}}\boldsymbol{D}^{(l)})_{(j,:)}\boldsymbol{z}_t + \alpha((\boldsymbol{D}^{(l)} - \boldsymbol{D}^*)^\text{T}\boldsymbol{D}^*)_{(j,:)}\boldsymbol{z}^* + \alpha(\boldsymbol{D}^{*\text{T}}\boldsymbol{D}^*)_{(j,:)}\boldsymbol{z}^* - \alpha\lambda_{t,j}^{(l)})$$

$$= \text{ReLU}((1 - \alpha)\boldsymbol{z}_{t,(j)} - \alpha\sum_{i \neq j}\langle \boldsymbol{D}_j^{(l)}, \boldsymbol{D}_i^{(l)} \rangle \boldsymbol{z}_{t,(i)} + \alpha\langle (\boldsymbol{D}_j^{(l)} - \boldsymbol{D}_j^*), \boldsymbol{D}_j^* \rangle \boldsymbol{z}_{(j)}^*$$

$$+ \alpha\sum_{i \neq j}\langle \boldsymbol{D}_j^{(l)} - \boldsymbol{D}_j^*, \boldsymbol{D}_i^* \rangle \boldsymbol{z}_{(i)}^* + \alpha\boldsymbol{z}_{(j)}^* + \alpha\sum_{i \neq j}\langle \boldsymbol{D}_j^*, \boldsymbol{D}_i^* \rangle \boldsymbol{z}_{(i)}^* - \alpha\lambda_{t,j}^{(l)})$$

$$= \text{ReLU}((1 - \alpha)\boldsymbol{z}_{t,(j)} + \alpha(1 - \beta_j^{(l)})\boldsymbol{z}_{(j)}^* + \alpha\eta_{t,j}^{(l)} - \alpha\lambda_{t,j}^{(l)}) \tag{29}$$

where $\beta_j^{(l)} = \langle \boldsymbol{D}_j^* - \boldsymbol{D}_j^{(l)}, \boldsymbol{D}_j^* \rangle$, and $\eta_{t,j}^{(l)} = -\sum_{i \neq j} \langle \boldsymbol{D}_j^{(l)}, \boldsymbol{D}_i^{(l)} \rangle \boldsymbol{z}_{t,(i)} + (\langle \boldsymbol{D}_j^{(l)} - \boldsymbol{D}_j^*, \boldsymbol{D}_i^* \rangle + \langle \boldsymbol{D}_j^*, \boldsymbol{D}_i^* \rangle) \boldsymbol{z}_{(i)}^*$. With $\|\boldsymbol{D}_j^{(l)} - \boldsymbol{D}_j^*\|_2 \leq \delta_l$, $\beta_j^{(l)}$ can be bounded as follows

$$\beta_j^{(l)} = \langle \boldsymbol{D}_j^* - \boldsymbol{D}_j^{(l)}, \boldsymbol{D}_j^* \rangle \leq \delta_l^2/2 \tag{30}$$

where we used the relation $\|\boldsymbol{D}_j^{(l)} - \boldsymbol{D}_j^*\|_2^2 = 2(1 - \langle \boldsymbol{D}_j^{(l)}, \boldsymbol{D}_j^* \rangle)$. We re-write $\eta_{t,j}^{(l)}$ below

$$
\begin{aligned}
\eta_{t,j}^{(l)} &= -\sum_{i \neq j} \langle \boldsymbol{D}_j^{(l)}, \boldsymbol{D}_i^{(l)} \rangle \boldsymbol{z}_{t,(i)} + (\langle \boldsymbol{D}_j^{(l)} - \boldsymbol{D}_j^*, \boldsymbol{D}_i^* \rangle + \langle \boldsymbol{D}_j^*, \boldsymbol{D}_i^* \rangle) \boldsymbol{z}_{(i)}^* \\
&= -\sum_{i \neq j} \langle \boldsymbol{D}_j^{(l)}, \boldsymbol{D}_i^{(l)} \rangle \boldsymbol{z}_{t,(i)} + \sum_{i \neq j} (\langle \boldsymbol{D}_j^{(l)} - \boldsymbol{D}_j^*, \boldsymbol{D}_i^* \rangle + \langle \boldsymbol{D}_j^*, \boldsymbol{D}_i^* \rangle) \boldsymbol{z}_{(i)}^* + \sum_{i \neq j} \langle \boldsymbol{D}_j^{(l)}, \boldsymbol{D}_i^{(l)} \rangle \boldsymbol{z}_{(i)}^* - \sum_{i \neq j} \langle \boldsymbol{D}_j^{(l)}, \boldsymbol{D}_i^{(l)} \rangle \boldsymbol{z}_{(i)}^* \\
&= \sum_{i \neq j} \langle \boldsymbol{D}_j^{(l)}, \boldsymbol{D}_i^{(l)} \rangle (\boldsymbol{z}_{(i)}^* - \boldsymbol{z}_{t,(i)}) + \sum_{i \neq j} (\langle \boldsymbol{D}_j^{(l)} - \boldsymbol{D}_j^*, \boldsymbol{D}_i^* \rangle + \langle \boldsymbol{D}_j^*, \boldsymbol{D}_i^* \rangle - \langle \boldsymbol{D}_j^{(l)}, \boldsymbol{D}_i^{(l)} \rangle) \boldsymbol{z}_{(i)}^* \\
&= \sum_{i \neq j} \langle \boldsymbol{D}_j^{(l)}, \boldsymbol{D}_i^{(l)} \rangle (\boldsymbol{z}_{(i)}^* - \boldsymbol{z}_{t,(i)}) + \sum_{i \neq j} \langle \boldsymbol{D}_j^{(l)}, \boldsymbol{D}_i^* - \boldsymbol{D}_i^{(l)} \rangle \boldsymbol{z}_{(i)}^* \\
&= \sum_{i \neq j} \langle \boldsymbol{D}_j^{(l)}, \boldsymbol{D}_i^{(l)} \rangle (\boldsymbol{z}_{(i)}^* - \boldsymbol{z}_{t,(i)}) + \gamma_j^{(l)}
\end{aligned}
\tag{31}
$$

where $\gamma_j^{(l)} = \sum_{i \neq j} \langle \boldsymbol{D}_j^{(l)}, \boldsymbol{D}_i^* - \boldsymbol{D}_i^{(l)} \rangle \boldsymbol{z}_{(i)}^*$. Given the sub-Gaussian entries of the code $\boldsymbol{z}^*$, we provide a bound on the variance of $\gamma_j^{(l)}$ below:

$$var(\gamma_j^{(l)}) = \sum_{i \neq j} \langle \boldsymbol{D}_j^{(l)}, \boldsymbol{D}_i^* - \boldsymbol{D}_i^{(l)} \rangle^2 \leq s\delta_l^2 \tag{32}$$

Now, using Chernoff bound on the sub-Gaussian code entries, we get

$$P(|\gamma_j^{(l)}| > a) \leq 2\exp\left(\frac{-a^2}{2s\delta_l^2}\right) \tag{33}$$

To bound all the terms in the support, for $j \in S^*$, we have

$$P(\max |\gamma_j^{(l)}| > a_\gamma) \leq \epsilon_\gamma^{(l)} \tag{34}$$

where $\epsilon_\gamma^{(l)} = 2s\exp\left(\frac{-a_\gamma^2}{2s\delta_l^2}\right)$. Let $a_\gamma = \mathcal{O}(\sqrt{s}\delta_l)$, then $\epsilon_\gamma^{(l)} = 2s\exp\left(\frac{-1}{\mathcal{O}(\delta_l)}\right)$. The above analysis states that with probability of at least $1 - \epsilon_\gamma^{(l)}$, $|\gamma_j^{(l)}| \leq a_\gamma = \mathcal{O}(\sqrt{s}\delta_l)$. Next, we write the recursion for when the support is identified (see Theorem 4.1). For the code at iteration $T$, we have

$$
\begin{aligned}
\boldsymbol{z}_{T,(j)} &= (1-\alpha)^T \boldsymbol{z}_{0,(j)} + \boldsymbol{z}_{(j)}^* \sum_{t=1}^T \alpha(1 - \beta_j^{(l)})(1-\alpha)^{T-t} + \sum_{t=1}^T \alpha(\eta_{t-1,j}^{(l)} - \lambda_{t-1,j}^{(l)})(1-\alpha)^{T-t} \\
&= (1-\alpha)^T \boldsymbol{z}_{0,(j)} + \boldsymbol{z}_{(j)}^*(1 - \beta_j^{(l)})(1 - (1-\alpha)^T) + \sum_{t=1}^T \alpha(\eta_{t-1,j}^{(l)} - \lambda_{t-1,j}^{(l)})(1-\alpha)^{T-t} \\
&= \boldsymbol{z}_{(j)}^*(1 - \beta_j^{(l)}) + (1-\alpha)^T(\boldsymbol{z}_{0,(j)} - \boldsymbol{z}_{(j)}^*(1 - \beta_j^{(l)})) + \sum_{t=1}^T \alpha(\eta_{t-1,j}^{(l)} - \lambda_{t-1,j}^{(l)})(1-\alpha)^{T-t} \\
&= \boldsymbol{z}_{(j)}^*(1 - \beta_j^{(l)}) + \zeta_{T,j}^{(l)}
\end{aligned}
\tag{35}
$$

where $\zeta_{T,j}^{(l)} = (1-\alpha)^T(\boldsymbol{z}_{0,(j)} - \boldsymbol{z}_{(j)}^*(1 - \beta_j^{(l)})) + \sum_{t=1}^T \alpha(\eta_{t-1,j}^{(l)} - \lambda_{t-1,j}^{(l)})(1-\alpha)^{T-t}$. With the support correctly identified at iteration $t-1$, we show that the support is preserved at iteration $t$. With $\|\boldsymbol{z}^* - \boldsymbol{z}_t\|_1 = \mathcal{O}(s)$, for each $j \in S^*$, we have

$$\eta_{t,j}^{(l)} = \sum_{i \neq j} \langle \boldsymbol{D}_j^{(l)}, \boldsymbol{D}_i^{(l)} \rangle (\boldsymbol{z}_{(i)}^* - \boldsymbol{z}_{t,(i)}) + \gamma_j^{(l)} \leq \frac{\mu_l}{\sqrt{m}}\|\boldsymbol{z}^* - \boldsymbol{z}_t\|_1 + a_\gamma = \mathcal{O}\left(\frac{s\log m}{\sqrt{m}}\right) \tag{36}$$

We make sure the regularizer is chosen such that

$$\lambda_t \geq \frac{\mu_l}{\sqrt{m}}\|\boldsymbol{z}^* - \boldsymbol{z}_t\|_1 + a_\gamma \tag{37}$$

We see that the larger the code error and coherence between the columns of the current dictionary, the larger $\lambda_t$ should be. This is to suppress the noise component in the code recursion and make sure no false support is introduced. Furthermore, we want $\lambda_t$ to be lower than half of the signal component, i.e.,

$$\alpha\lambda_t \leq \frac{1-\alpha}{2}\boldsymbol{z}_{t,(j)} + \frac{\alpha}{2}(1-\beta_j^{(l)})\boldsymbol{z}_{(j)}^*, \forall j \in S^*$$
$$\alpha\lambda_t \leq \frac{1-\alpha}{2}\boldsymbol{z}_t^{\min} + \frac{\alpha}{2}(1-\frac{\delta_l^2}{2})C_{\min} \tag{38}$$

where $\boldsymbol{z}_t^{\min} = \min_j \boldsymbol{z}_{t,(j)}$. We further shrink the upper bound, given the code from previous iteration $t-1$ (i.e., $\alpha\lambda_{t-1} \leq \boldsymbol{z}_t^{\min}$). Hence, we want the regularizer to follow

$$\alpha\lambda_t \leq \frac{1-\alpha}{2}\alpha\lambda_{t-1} + \frac{\alpha}{2}(1-\frac{\delta_l^2}{2})C_{\min}$$
$$\lambda_t \leq \frac{1-\alpha}{2}\lambda_{t-1} + \frac{1}{2}(1-\frac{\delta_l^2}{2})C_{\min} \tag{39}$$

This condition is to make sure the identified supports are not killed in the recursion. We use the condition to set the step size $\alpha$. We get

$$\alpha \leq 1 - \frac{2\lambda_t - (1-\frac{\delta_l^2}{2})C_{\min}}{\lambda_{t-1}} \tag{40}$$

Hence, $\lambda_t = \boldsymbol{\Omega}(\frac{s\log m}{\sqrt{m}})$ and $\alpha$ should be chosen sufficiently small such that the condition above is met. We denote $\epsilon_{\text{supp-pres}}^{(l)} := \epsilon_{\text{supp-rec}}^{(l)} + \epsilon_\gamma^{(l)} = 2p\exp(\frac{-C_{\min}^2}{\mathcal{O}^*(\delta_l^2)}) + 2s\exp(\frac{-1}{\mathcal{O}(\delta_l)})$. Hence, with probability of at least $1 - \epsilon_{\text{supp-pres}}^{(l)}$, the support, recovered at the first iteration, is preserved through the encoder iterations. ∎

Theorem 4.1 and Theorem 4.2 allow to achieve linear convergence in the forward pass right after the first encoder iteration, i.e., $B = 1$. With support recovery at first iteration and its preservation, we now re-state the forward pass code convergence (Theorem 4.3).

**Theorem 4.4** (Local forward pass code convergence). *Given the encoder* $\boldsymbol{z}_{t+1} = \Phi(\boldsymbol{z}_t, \boldsymbol{D})$, *Assumption 3.6, Lemmas 3.2, A.1 and A.2, then* $\exists \rho < 1, B > 0$ *s.t.* $\|\boldsymbol{z}_t - \hat{\boldsymbol{z}}\|_2 \leq \mathcal{O}(\rho^t) \; \forall t > B$, *where* $\hat{\boldsymbol{z}}$ *is the unique minimizer of lasso* (1). *Furthermore, given Theorem 4.1 and Theorem 4.2,* $B = 1$.

*Proof.* Given the support selection at iteration $B$, from Lemma A.1, we have $\nabla_{11}^2\mathcal{L}_{\boldsymbol{x}}(\boldsymbol{z}_t, \boldsymbol{D}) \succeq \mu\boldsymbol{I}$ restricted to the support for $t > B$. Then, from Lemma A.2, we get

$$\|\nabla_1\Phi(\boldsymbol{z}_t, \boldsymbol{D})\|_2 = \|(\boldsymbol{I} - \alpha\nabla_{11}^2\mathcal{L}_{\boldsymbol{x}}(\boldsymbol{z}_t, \boldsymbol{D}))\partial\mathcal{P}_{\alpha\lambda}(\boldsymbol{z}_t)\|_2 \leq \rho$$

where $\rho \triangleq c_{\text{prox}}(1 - \alpha\mu) < 1$. Hence, using fixed-point property (Lemma 3.2)

$$\exists \, B > 0, \text{ s.t. } \|\boldsymbol{z}_{t+1} - \hat{\boldsymbol{z}}\|_2 = \|\Phi(\boldsymbol{z}_t) - \Phi(\hat{\boldsymbol{z}})\|_2 \leq \rho\|\boldsymbol{z}_t - \hat{\boldsymbol{z}}\|_2 \; \forall t > B$$

where $\hat{\boldsymbol{z}} = \arg\min f_{\boldsymbol{x}}(\boldsymbol{z}, \boldsymbol{D})$. Unrolling the recursion,

$$\|\boldsymbol{z}_t - \hat{\boldsymbol{z}}\|_2 \leq \rho^{t-B}\|\boldsymbol{z}_B - \hat{\boldsymbol{z}}\|_2.$$

∎

**Theorem 4.4** (Global forward pass code error with variable $\lambda_t$). *Given Assumptions 3.3 and 3.4, suppose* $\boldsymbol{D}^{(l)}$ *is* $\mu_l$-*incoherent and* $\delta_l = \mathcal{O}^*(1/\log p)$ *close to* $\boldsymbol{D}^*$. *If* $s = \mathcal{O}^*(\sqrt{m}/\mu\log m)$, $\mu = \mathcal{O}(\log m)$, *and the regularizer*

and step size are chosen such that $\lambda_t^{(l)} = \frac{\mu_l}{\sqrt{m}} \|\boldsymbol{z}^* - \boldsymbol{z}_t\|_1 + a_\gamma = \boldsymbol{\Omega}(\frac{s \log m}{\sqrt{m}})$ and $\alpha^{(l)} \leq 1 - \frac{2\lambda_t^{(l)} - (1 - \frac{\delta_l^2}{2})C_{\min}}{\lambda_{t-1}^{(l)}}$, then with probability of at least $1 - \epsilon_{supp\text{-}pres}^{(l)}$, for $j \in supp(\boldsymbol{z}^*)$, the code coefficient error is

$$|\boldsymbol{z}_{t,(j)}^{(l)} - \boldsymbol{z}_{(j)}^*| \leq \mathcal{O}(\sqrt{s\|\boldsymbol{D}_j^{(l)} - \boldsymbol{D}_j^*\|_2} + e_{t,j}^{(l)unroll}) \tag{8}$$

and

$$\boldsymbol{z}_{T,(j)} = \boldsymbol{z}_{(j)}^*(1 - \beta_j^{(l)}) + \zeta_{T,j}^{(l)} \tag{9}$$

where $e_{t,j}^{(l)unroll} := 2(s-1)t\alpha\frac{\mu_l}{\sqrt{m}}\max_i |\boldsymbol{z}_{0,(i)}^{(l)} - \boldsymbol{z}_{(i)}^*|\delta_{\alpha,t-1} + |\boldsymbol{z}_{0,(j)}^{(l)} - \boldsymbol{z}_{(j)}^*|\delta_{\alpha,t}$, $\delta_{\alpha,t} := (1 - \alpha + 2\alpha\frac{\mu_l}{\sqrt{m}})^t$, $|\zeta_{T,j}^{(l)}| = \mathcal{O}(a_\gamma)$ with $a_\gamma = \mathcal{O}(\sqrt{s\delta_l})$, $\beta_j^{(l)} = \langle \boldsymbol{D}_j^* - \boldsymbol{D}_j^{(l)}, \boldsymbol{D}_j^* \rangle \leq \frac{\delta_l^2}{2}$ and $\epsilon_{supp\text{-}pres}^{(l)} := \epsilon_{supp\text{-}rec}^{(l)} + \epsilon_\gamma^{(l)} = 2p\exp(\frac{-C_{\min}^2}{\mathcal{O}^*(\delta_l^2)}) + 2s\exp(\frac{-1}{\mathcal{O}(\delta_l)})$. With appropriately large $t$, $|\boldsymbol{z}_{t,(j)}^{(l)} - \boldsymbol{z}_{(j)}^*| = \mathcal{O}(\sqrt{s\|\boldsymbol{D}_j^{(l)} - \boldsymbol{D}_j^*\|_2})$.

*Proof.* We define $\tilde{\eta}_{t,j}^{(l)} := \sum_{i \neq j} |\langle \boldsymbol{D}_j^{(l)}, \boldsymbol{D}_i^{(l)} \rangle| \|\boldsymbol{z}_{(i)}^* - \boldsymbol{z}_{t,(i)}| + \gamma_j^{(l)}$ and upper bound it as

$$\tilde{\eta}_{t,j}^{(l)} \leq \frac{\mu_l}{\sqrt{m}}\sum_{i \neq j} E_{t,i} + \gamma_j^{(l)} \tag{41}$$

where $E_{t,i} := |\boldsymbol{z}_{(i)}^* - \boldsymbol{z}_{t,(i)}|$. Given (21), we re-write the code recursion

$$\boldsymbol{z}_{t+1,(j)} = \mathcal{P}_{\alpha\lambda_{t,j}^{(l)}}((1-\alpha)\boldsymbol{z}_{t,(j)} + \alpha(1 - \beta_j^{(l)})\boldsymbol{z}_{(j)}^* + \alpha\eta_{t,j}^{(l)})$$

$$\in (1-\alpha)\boldsymbol{z}_{t,(j)} + \alpha(1 - \beta_j^{(l)})\boldsymbol{z}_{(j)}^* + \alpha(\eta_{t,j}^{(l)} - \lambda_{t,j}^{(l)}\partial|\boldsymbol{z}_{t+1,(j)}|) \tag{42}$$

$$E_{t+1,j} = |\boldsymbol{z}_{t+1,(j)} - \boldsymbol{z}_{(j)}^*| \leq (1-\alpha)E_{t,j} + \alpha\beta_j^{(l)}|\boldsymbol{z}_{(j)}^*| + \alpha(\tilde{\eta}_{t,j}^{(l)} + \lambda_{t,j}^{(l)})$$

Opening up the recursion, we get

$$E_{t+1,j} \leq E_{0,j}\prod_{q=0}^{t}(1-\alpha) + \beta_j^{(l)}|\boldsymbol{z}_{(j)}^*|\sum_{a=1}^{t+1}\alpha\prod_{q=a}^{t+1}(1-\alpha) + \sum_{a=1}^{t+1}\alpha(\tilde{\eta}_{a-1,j}^{(l)} + \lambda_{a-1,j}^{(l)})\prod_{q=a}^{t+1}(1-\alpha) \tag{43}$$

where we define $\prod_{q=a}^{a}(1-\alpha) = 1$. Using the upper bound from (41) and $\lambda_{t,j}^{(l)}$ from Theorem 4.2, we get

$$E_{t+1,j} \leq v_{t+1,j} + 2\frac{\mu_l}{\sqrt{m}}\alpha\sum_{a=1}^{t+1}\sum_{i \neq j}E_{a-1,i}\prod_{q=a}^{t+1}(1-\alpha) \tag{44}$$

where $v_{t+1,j} := E_{0,j}\prod_{q=0}^{t}(1-\alpha) + (\beta_j^{(l)}|\boldsymbol{z}_{(j)}^*| + 2\gamma_j^{(l)})\sum_{a=1}^{t+1}\alpha\prod_{q=a}^{t+1}(1-\alpha)$. We now derive the general upper bound on $E_{t+1,j}$ as follows

$$E_{1,i_1} \leq v_{1,i_1} + 2\alpha\frac{\mu_l}{\sqrt{m}}\sum_{i_2 \neq i_1}E_{0,i_2} \tag{45}$$

For $E_{2,i_1}$, we have

$$E_{2,i_1} \leq v_{2,i_1} + 2\alpha\frac{\mu_l}{\sqrt{m}}(\sum_{i_2 \neq i_1}E_{1,i_2} + \sum_{i_2 \neq i_1}E_{0,i_2}(1-\alpha)) \tag{46}$$

Substituting $E_{1,i_1}$,

$$E_{2,i_1} \leq v_{2,i_1} + 2\alpha\frac{\mu_l}{\sqrt{m}}(\sum_{i_2 \neq i_1}(v_{1,i_2} + 2\alpha\frac{\mu_l}{\sqrt{m}}\sum_{i_3 \neq i_2}E_{0,i_3}) + \sum_{i_2 \neq i_1}E_{0,i_2}(1-\alpha)) \tag{47}$$

For $E_{3,i_1}$, we have

$$E_{3,i_1} \leq v_{3,i_1} + 2\alpha\frac{\mu_l}{\sqrt{m}}\sum_{a=1}^{3}\sum_{i_2 \neq i_1}E_{a-1,i_2}(1-\alpha)^{3-a} \tag{48}$$

Unrolling the recursion,

$$
\begin{aligned}
E_{3,i_1} \leq{} & v_{3,i_1} + 2\alpha\frac{\mu_l}{\sqrt{m}}\sum_{i_2 \neq i_1} v_{2,i_2} + 2\alpha\frac{\mu_l}{\sqrt{m}}\left((1-\alpha)\sum_{i_2 \neq i_1} v_{1,i_2} + 2\alpha\frac{\mu_l}{\sqrt{m}}\sum_{i_3 \neq i_2} v_{1,i_3}\right) \\
& + 2\alpha\frac{\mu_l}{\sqrt{m}}\left((1-\alpha)^2\sum_{i_2 \neq i_1} E_{0,i_2} + 2(1-\alpha)(2\alpha\frac{\mu_l}{\sqrt{m}})\sum_{i_3 \neq i_2, i_1} E_{0,i_3} + (2\alpha\frac{\mu_l}{\sqrt{m}})^2\sum_{i_4 \neq i_3, i_2, i_1} E_{0,i_4}\right)
\end{aligned}
\tag{49}
$$

Given above, we can write up the relation as

$$
\begin{aligned}
E_{3,i_1} \leq{} & v_{3,i_1} + 2\alpha\frac{\mu_l}{\sqrt{m}}\left((s-1)v_{2,i} + v_{1,i}((1-\alpha)(s-1) + 2\alpha\frac{\mu_l}{\sqrt{m}}(s-2))\right) \\
& + 2\alpha\frac{\mu_l}{\sqrt{m}}E_{0,i}\left((s-1)(1-\alpha)^2 + 2(1-\alpha)(2\alpha\frac{\mu_l}{\sqrt{m}})(s-2) + (2\alpha\frac{\mu_l}{\sqrt{m}})^2(s-3)\right) \\
\leq{} & v_{3,i_1} + (s-1)2\alpha\frac{\mu_l}{\sqrt{m}}\left(v_{2,i} + v_{1,i}((1-\alpha) + 2\alpha\frac{\mu_l}{\sqrt{m}})\right) \\
& + 2(s-1)\alpha\frac{\mu_l}{\sqrt{m}}E_{0,i}\left((1-\alpha)^2 + 2(1-\alpha)(2\alpha\frac{\mu_l}{\sqrt{m}}) + (2\alpha\frac{\mu_l}{\sqrt{m}})^2\right)
\end{aligned}
\tag{50}
$$

Following similar steps for $E_{4,i_1}$, we get,

$$
\begin{aligned}
E_{4,i_1} \leq{} & v_{4,i_1} + (s-1)2\alpha\frac{\mu_l}{\sqrt{m}}\left(v_{3,i} + v_{2,i}(1-\alpha + 2\alpha\frac{\mu_l}{\sqrt{m}}) + v_{1,i}((1-\alpha)^2 + 2(1-\alpha)(2\alpha\frac{\mu_l}{\sqrt{m}}) + (2\alpha\frac{\mu_l}{\sqrt{m}})^2)\right) \\
& + 2(s-1)\alpha\frac{\mu_l}{\sqrt{m}}E_{0,i}\left((1-\alpha)^3 + 3(1-\alpha)(2\alpha\frac{\mu_l}{\sqrt{m}})^2 + 3(1-\alpha)^2(2\alpha\frac{\mu_l}{\sqrt{m}}) + (2\alpha\frac{\mu_l}{\sqrt{m}})^3\right)
\end{aligned}
\tag{51}
$$

This leads to the term

$$
\begin{aligned}
E_{4,i_1} \leq{} & v_{4,i_1} + (s-1)2\alpha\frac{\mu_l}{\sqrt{m}}\left(v_{3,i} + v_{2,i}(1-\alpha + 2\alpha\frac{\mu_l}{\sqrt{m}})^1 + v_{1,i}(1-\alpha + 2\alpha\frac{\mu_l}{\sqrt{m}})^2\right) \\
& + 2(s-1)\alpha\frac{\mu_l}{\sqrt{m}}E_{0,i}(1-\alpha + 2\alpha\frac{\mu_l}{\sqrt{m}})^3
\end{aligned}
\tag{52}
$$

Hence, the general term for code error at $t$ layer is

$$
E_{t+1,j} \leq v_{t+1,j} + 2(s-1)\alpha\frac{\mu_l}{\sqrt{m}}\sum_{a=1}^{t} v_{a,\max}(1-\alpha + 2\alpha\frac{\mu_l}{\sqrt{m}})^{t-a} + 2(s-1)\alpha\frac{\mu_l}{\sqrt{m}}E_{0,\max}(1-\alpha + 2\alpha\frac{\mu_l}{\sqrt{m}})^t
\tag{53}
$$

where for $j$ in the support, we define the upper bounds $v_{a,j} \leq v_{a,\max}$ and $E_{0,j} \leq E_{0,\max}$. Next, we define $(1-\alpha)^t \leq \delta_{\alpha,t} := (1-\alpha + 2\alpha\frac{\mu_l}{\sqrt{m}})^t$, and use it to find an upper bound on the expression $\sum_{a=1}^{t} v_{a,\max}(1-\alpha + 2\alpha\frac{\mu_l}{\sqrt{m}})^{t-a}$. We have

$$
v_{t,j} = E_{0,j}(1-\alpha)^t + (\beta_j^{(l)}|\boldsymbol{z}_{(j)}^*| + 2\gamma_j^{(l)})\sum_{k=1}^{t}\alpha(1-\alpha)^{t-k+1}
\tag{54}
$$

We bound the expression

$$
\begin{aligned}
\sum_{a=1}^{t} v_{a,j}(1-\alpha + 2\alpha\frac{\mu_l}{\sqrt{m}})^{t-a} &\leq \sum_{a=1}^{t}\left(E_{0,j}(1-\alpha)^a + (\beta_j^{(l)}|\boldsymbol{z}_{(j)}^*| + 2\gamma_j^{(l)})\sum_{k=1}^{a}\alpha(1-\alpha)^{a-k+1}\right)(1-\alpha + 2\alpha\frac{\mu_l}{\sqrt{m}})^{t-a} \\
&\leq E_{0,j}t\delta_{\alpha,t} + (\beta_j^{(l)}|\boldsymbol{z}_{(j)}^*| + 2\gamma_j^{(l)})\sum_{a=1}^{t}(1-\alpha + 2\alpha\frac{\mu_l}{\sqrt{m}})^{t-a}\sum_{k=1}^{a}\alpha(1-\alpha)^{a-k+1}
\end{aligned}
\tag{55}
$$

Using sum of geometric series, we write $\sum_{a=1}^{t}(1-\alpha+2\alpha\frac{\mu_l}{\sqrt{m}})^{t-a} = \frac{1-(1-\alpha+2\alpha\frac{\mu_l}{\sqrt{m}})^t}{\alpha-2\alpha\frac{\mu_l}{\sqrt{m}}} \leq \frac{1}{\alpha(1-2\frac{\mu_l}{\sqrt{m}})}$. Hence, using (30) and (34), with probability of at least $1 - \epsilon_\gamma^{(l)}$, we have

$$\sum_{a=1}^{t} v_{a,\max}(1-\alpha+2\alpha\frac{\mu_l}{\sqrt{m}})^{t-a} \leq E_{0,\max}t\delta_{\alpha,t} + \frac{1}{\alpha(1-2\frac{\mu_l}{\sqrt{m}})}(\frac{\delta_l^2}{2}|\boldsymbol{z}_{\max}^*| + 2a_\gamma) \tag{56}$$

Hence, we bound the code error on the coefficients as following

$$E_{t+1,j} \leq v_{t+1,j} + 2(s-1)\frac{\mu_l}{\sqrt{m}}(\frac{1}{(1-2\frac{\mu_l}{\sqrt{m}})}(\frac{\delta_l^2}{2}|\boldsymbol{z}_{\max}^*| + 2a_\gamma)) + 2(t+1)(s-1)\alpha\frac{\mu_l}{\sqrt{m}}E_{0,\max}\delta_{\alpha,t} \tag{57}$$

Next, we further simplify the first term

$$v_{t+1,j} = E_{0,j}(1-\alpha)^{t+1} + (\beta_j^{(l)}|\boldsymbol{z}_{(j)}^*| + 2\gamma_j^{(l)})\sum_{k=1}^{t+1}\alpha(1-\alpha)^{t-k+1} \leq E_{0,j}\delta_{\alpha,t+1} + \frac{\delta_l^2}{2}|\boldsymbol{z}_{\max}^*| + 2a_\gamma \tag{58}$$

Substituting the above upper bound into the upper bound for $E_{t+1,j}$, we get

$$E_{t+1,j} \leq E_{0,j}\delta_{\alpha,t+1} + (1+2(s-1)\kappa_l)(\frac{\delta_l^2}{2}|\boldsymbol{z}_{\max}^*| + 2a_\gamma) + 2(t+1)(s-1)\alpha\frac{\mu_l}{\sqrt{m}}E_{0,\max}\delta_{\alpha,t} \tag{59}$$

where $\kappa_l := \frac{\mu_l}{\sqrt{m}}(\frac{1}{(1-2\frac{\mu_l}{\sqrt{m}})}$. Given $s = \mathcal{O}^*(\sqrt{m}/\mu\log m)$, we have $(s-1)\kappa_l < 1$. Hence, with probability of at least $1 - \epsilon_\gamma^{(l)}$, we have

$$E_{t,j} \leq \mathcal{O}(a_\gamma) + 2(s-1)t\alpha\frac{\mu_l}{\sqrt{m}}E_{0,\max}\delta_{\alpha,t-1} + E_{0,j}\delta_{\alpha,t}$$
$$|\boldsymbol{z}_{t,(j)}^{(l)} - \boldsymbol{z}_{(j)}^*| \leq \mathcal{O}(\sqrt{s\|\boldsymbol{D}_j^{(l)} - \boldsymbol{D}_j^*\|_2} + e_{t,j}^{(l)\text{unroll}}) \tag{60}$$

where $e_{t,j}^{(l)\text{unroll}} := 2(s-1)t\alpha\frac{\mu_l}{\sqrt{m}}\max_i|\boldsymbol{z}_{0,(i)}^{(l)} - \boldsymbol{z}_{(i)}^*|\delta_{\alpha,t-1} + |\boldsymbol{z}_{0,(j)}^{(l)} - \boldsymbol{z}_{(j)}^*|\delta_{\alpha,t}$. With appropriately large unrolled layer $t$, $e_{t,j}^{(l)\text{unroll}} \approx 0$. Hence, for the code error on non-zero coefficients, we get

$$|\boldsymbol{z}_{t,(j)}^{(l)} - \boldsymbol{z}_{(j)}^*| = \mathcal{O}(\sqrt{s\|\boldsymbol{D}_j^{(l)} - \boldsymbol{D}_j^*\|_2}) \tag{61}$$

for large enough $t$. Now, we try to prove the relation for $\boldsymbol{z}_{T,(j)}$. For shrinkage, we re-write (35)

$$\boldsymbol{z}_{T,(j)} \in \boldsymbol{z}_{(j)}^*(1-\beta_j^{(l)}) + \zeta_{T,j}^{(l)} \tag{62}$$

where $\zeta_{T,j}^{(l)} = \kappa_{T,j}^{(l)} + \sum_{t=1}^{T}\alpha(\eta_{t-1,j}^{(l)} - \lambda_{t-1,j}^{(l)}\partial|\boldsymbol{z}_{t,(j)}|)(1-\alpha)^{T-t}$ and $\kappa_{T,j}^{(l)} := (1-\alpha)^T(\boldsymbol{z}_{0,(j)} - \boldsymbol{z}_{(j)}^*(1-\beta_j^{(l)}))$. $\kappa_{T,j}^{(l)}$ decays very fast as $T$ increases. Hence, we bound the second term. We substitute $\eta_{t,j}^{(l)} = \sum_{i\neq j}\langle\boldsymbol{D}_j^{(l)}, \boldsymbol{D}_i^{(l)}\rangle(\boldsymbol{z}_{(i)}^* - \boldsymbol{z}_{t,(i)}) + \gamma_j^{(l)}$ in $\zeta_{T,j}^{(l)}$.

$$\begin{aligned}
\zeta_{T,j}^{(l)} &\in \kappa_{T,j}^{(l)} + \sum_{t=1}^{T}\alpha(\eta_{t-1,j}^{(l)} - \lambda_{t-1,j}^{(l)}\partial|\boldsymbol{z}_{t,(j)}|)(1-\alpha)^{T-t} \\
&\in \kappa_{T,j}^{(l)} + \sum_{t=1}^{T}\alpha(\sum_{i\neq j}\langle\boldsymbol{D}_j^{(l)}, \boldsymbol{D}_i^{(l)}\rangle(\boldsymbol{z}_{(i)}^* - \boldsymbol{z}_{t-1,(i)}) + 2\gamma_j^{(l)} - \lambda_{t-1,j}^{(l)}\partial|\boldsymbol{z}_{t,(j)}|)(1-\alpha)^{T-t} \\
&\leq \kappa_{T,j}^{(l)} + 2\gamma_j^{(l)}\sum_{t=1}^{T}\alpha(1-\alpha)^{T-t} + 2\alpha\frac{\mu_l}{\sqrt{m}}\sum_{t=1}^{T}\sum_{i\neq j}E_{t-1,j}(1-\alpha)^{T-t} \\
&\leq \kappa_{T,j}^{(l)} + 2\gamma_j^{(l)} + 2(s-1)\alpha\frac{\mu_l}{\sqrt{m}}\sum_{t=1}^{T}E_{t-1,j}(1-\alpha)^{T-t}
\end{aligned} \tag{63}$$

Given above, we find an upper bound on $E_{t-1,j}(1-\alpha)^{T-t}$ below. From analysis in Theorem 4.4, we have

$$
\begin{aligned}
E_{t-1,j} &\leq v_{t-1,j} + 2(s-1)\alpha\frac{\mu_l}{\sqrt{m}}\sum_{a=1}^{t-2}v_{a,\max}(1-\alpha+2\alpha\frac{\mu_l}{\sqrt{m}})^{t-a-2} \\
&\quad + 2(s-1)\alpha\frac{\mu_l}{\sqrt{m}}E_{0,\max}(1-\alpha+2\alpha\frac{\mu_l}{\sqrt{m}})^{t-2} \\
E_{t-1,j}(1-\alpha)^{T-t} &\leq v_{t-1,j}(1-\alpha)^{T-t} + 2(s-1)\alpha\frac{\mu_l}{\sqrt{m}}\sum_{a=1}^{t-2}v_{a,\max}(1-\alpha+2\alpha\frac{\mu_l}{\sqrt{m}})^{t-a-2}(1-\alpha)^{T-t} \\
&\quad + 2(s-1)\alpha\frac{\mu_l}{\sqrt{m}}E_{0,\max}(1-\alpha+2\alpha\frac{\mu_l}{\sqrt{m}})^{t-2}(1-\alpha)^{T-t}
\end{aligned}
\tag{64}
$$

Re-write the first term,

$$
\begin{aligned}
v_{t-1,j}(1-\alpha)^{T-t} &= (E_{0,j}(1-\alpha)^{t-1} + (\beta_j^{(l)}|\boldsymbol{z}_{(j)}^*| + 2\gamma_j^{(l)})\sum_{k=1}^{t-1}\alpha(1-\alpha)^{t-k-1})(1-\alpha)^{T-t} \\
&= E_{0,j}(1-\alpha)^{T-1} + (\beta_j^{(l)}|\boldsymbol{z}_{(j)}^*| + 2\gamma_j^{(l)})\sum_{k=1}^{t-1}\alpha(1-\alpha)^{T-k-1}
\end{aligned}
\tag{65}
$$

Similarly,

$$
v_{a,\max} = E_{0,\max}(1-\alpha)^a + (\beta_{\max}^{(l)}|\boldsymbol{z}_{\max}^*| + 2\gamma_{\max}^{(l)})\sum_{k=1}^{a}\alpha(1-\alpha)^{a-k}
\tag{66}
$$

We write

$$
\begin{aligned}
\sum_{a=1}^{t-2}v_{a,\max}(1-\alpha+2\alpha\frac{\mu_l}{\sqrt{m}})^{t-a-2} &= \sum_{a=1}^{t-2}E_{0,\max}(1-\alpha)^a(1-\alpha+2\alpha\frac{\mu_l}{\sqrt{m}})^{t-a-2} \\
&\quad + \sum_{a=1}^{t-2}(\beta_{\max}^{(l)}|\boldsymbol{z}_{\max}^*| + 2\gamma_{\max}^{(l)})\sum_{k=1}^{a}\alpha(1-\alpha)^{a-k}(1-\alpha+2\alpha\frac{\mu_l}{\sqrt{m}})^{t-a-2} \\
&\leq \sum_{a=1}^{t-2}E_{0,\max}(1-\alpha+2\alpha\frac{\mu_l}{\sqrt{m}})^{t-2} \\
&\quad + (\beta_{\max}^{(l)}|\boldsymbol{z}_{\max}^*| + 2\gamma_{\max}^{(l)})\sum_{a=1}^{t-2}(1-\alpha+2\alpha\frac{\mu_l}{\sqrt{m}})^{t-a-2}
\end{aligned}
\tag{67}
$$

Hence,

$$
\begin{aligned}
(1-\alpha)^{T-t}\sum_{a=1}^{t-2}v_{a,\max}(1-\alpha+2\alpha\frac{\mu_l}{\sqrt{m}})^{t-a-2} &\leq \sum_{a=1}^{t-2}E_{0,\max}(1-\alpha+2\alpha\frac{\mu_l}{\sqrt{m}})^{t-2}(1-\alpha)^{T-t} \\
&\quad + (\beta_{\max}^{(l)}|\boldsymbol{z}_{\max}^*| + 2\gamma_{\max}^{(l)})\sum_{a=1}^{t-2}(1-\alpha+2\alpha\frac{\mu_l}{\sqrt{m}})^{t-a-2}(1-\alpha)^{T-t} \\
&\leq (t-2)E_{0,\max}(1-\alpha+2\alpha\frac{\mu_l}{\sqrt{m}})^{T-2} \\
&\quad + (\beta_{\max}^{(l)}|\boldsymbol{z}_{\max}^*| + 2\gamma_{\max}^{(l)})\frac{(1-\alpha)^{T-t}}{\alpha(1-2\frac{\mu_l}{\sqrt{m}})}
\end{aligned}
\tag{68}
$$

Combining all terms, we get

$$
\begin{aligned}
E_{t-1,j}(1-\alpha)^{T-t} &\le v_{t-1,j}(1-\alpha)^{T-t} + 2(s-1)\alpha\frac{\mu_l}{\sqrt{m}}\sum_{a=1}^{t-2} v_{a,\max}(1-\alpha+2\alpha\frac{\mu_l}{\sqrt{m}})^{t-a-2}(1-\alpha)^{T-t} \\
&\quad + 2(s-1)\alpha\frac{\mu_l}{\sqrt{m}}E_{0,\max}(1-\alpha+2\alpha\frac{\mu_l}{\sqrt{m}})^{t-2}(1-\alpha)^{T-t} \\
&\le E_{0,j}(1-\alpha)^{T-1} + 2(s-1)\alpha\frac{\mu_l}{\sqrt{m}}(t-1)E_{0,\max}\delta_{\alpha,T-2} \\
&\quad + (\beta_{\max}^{(l)}|\mathbf{z}_{\max}^*| + 2\gamma_{\max}^{(l)})\left(\sum_{k=1}^{t-1}\alpha(1-\alpha)^{T-k-1} + 2(s-1)\kappa_l(1-\alpha)^{T-t}\right) \\
&\le (1+2(s-1)\alpha\frac{\mu_l}{\sqrt{m}}(t-1))E_{0,\max}\delta_{\alpha,T-2} \\
&\quad + (\beta_{\max}^{(l)}|\mathbf{z}_{\max}^*| + 2\gamma_{\max}^{(l)})\left(\sum_{k=1}^{t-1}\alpha(1-\alpha)^{T-k-1} + 2(s-1)\kappa_l(1-\alpha)^{T-t}\right)
\end{aligned}
\tag{69}
$$

where $\kappa_l = \frac{\frac{\mu_l}{\sqrt{m}}}{(1-2\frac{\mu_l}{\sqrt{m}})}$. Moreover, we bound

$$
\sum_{t=1}^{T}\sum_{k=1}^{t-1}\alpha(1-\alpha)^{T-k-1} = \sum_{t=1}^{T}\alpha(1-\alpha)^{T-t}\frac{1-(1-\alpha)^{t-1}}{\alpha} \le \sum_{t=1}^{T}(1-\alpha)^{T-t} \le \frac{1}{\alpha}
\tag{70}
$$

Finally, we are ready to write the bound for $\zeta_{T,j}^{(l)}$

$$
\begin{aligned}
|\zeta_{T,j}^{(l)}| &\le \kappa_{T,j}^{(l)} + 2|\gamma_j^{(l)}| + 2(s-1)\frac{\mu_l}{\sqrt{m}}(\beta_{\max}^{(l)}|\mathbf{z}_{\max}^*| + 2\gamma_{\max}^{(l)})(1+2(s-1)\kappa_l) \\
&\quad + \sum_{t=1}^{T} 2(s-1)\alpha\frac{\mu_l}{\sqrt{m}}((1+2(s-1)\alpha\frac{\mu_l}{\sqrt{m}}(t-1))E_{0,\max}\delta_{\alpha,T-2})
\end{aligned}
\tag{71}
$$

Given $|\gamma_j^{(l)}| = a_\gamma$ with probability of $1-\epsilon_\gamma^{(l)}$ where $a_\gamma = \sqrt{s\delta_l}$ and $s = \mathcal{O}^*(\sqrt{m}/\mu\log m)$, we will have

$$
|\zeta_{T,j}^{(l)}| \le \mathcal{O}(\sqrt{s\|\mathbf{D}_j^{(l)} - \mathbf{D}_j^*\|_2})
\tag{72}
$$

∎

**Theorem 4.5** (Global forward pass code error with fixed $\lambda_t$). *Given Assumptions 3.3 and 3.4, suppose $\mathbf{D}^{(l)}$ is $\mu_l$-incoherent and $\delta_l = \mathcal{O}^*(1/\log p)$ close to $\mathbf{D}^*$. If $s = \mathcal{O}^*(\sqrt{m}/\mu\log m)$, $\mu = \mathcal{O}(\log m)$, and the regularizer and step size are chosen such that $\lambda_t^{(l)} = \lambda^{fixed} = \frac{\mu_l}{\sqrt{m}}\|\mathbf{z}^* - \mathbf{z}_0\|_1 + a_\gamma = \mathbf{\Omega}(\frac{s\log m}{\sqrt{m}})$ and $\alpha^{(l)} \le 1 - \frac{2\lambda_t^{(l)} - (1-\frac{\delta_l^2}{2})C_{\min}}{\lambda_{t-1}^{(l)}}$, then with probability of at least $1 - \epsilon_{supp-pres}^{(l)}$, for $j \in supp(\mathbf{z}^*)$, the code coefficient error is*

$$
|\mathbf{z}_{t,(j)}^{(l)} - \mathbf{z}_{(j)}^*| \le \mathcal{O}(\sqrt{s\|\mathbf{D}_j^{(l)} - \mathbf{D}_j^*\|_2} + e_{t,j}^{(l)unroll,fixed} + \lambda^{fixed})
\tag{10}
$$

*and*

$$
\mathbf{z}_{T,(j)} = \mathbf{z}_{(j)}^*(1-\beta_j^{(l)}) + \zeta_{T,j}^{(l)}
\tag{11}
$$

*where $e_{t,j}^{(l)unroll,\ fixed} := (s-1)t\alpha\frac{\mu_l}{\sqrt{m}}\max_i|\mathbf{z}_{0,(i)}^{(l)} - \mathbf{z}_{(i)}^*|\delta_{\alpha,t-1}^{fixed} + |\mathbf{z}_{0,(j)}^{(l)} - \mathbf{z}_{(j)}^*|\delta_{\alpha,t}^{fixed}$, $\delta_{\alpha,t}^{fixed} := (1-\alpha+\alpha\frac{\mu_l}{\sqrt{m}})^t$, $|\zeta_{T,j}^{(l)}| = \mathcal{O}(a_\gamma + \lambda^{fixed})$ with $a_\gamma = \mathcal{O}(\sqrt{s\delta_l})$, $\beta_j^{(l)} = \langle\mathbf{D}_j^* - \mathbf{D}_j^{(l)}, \mathbf{D}_j^*\rangle \le \frac{\delta_l^2}{2}$, and $\epsilon_{supp-pres}^{(l)} := \epsilon_{supp-rec}^{(l)} + \epsilon_\gamma^{(l)} = 2p\exp\left(\frac{-C_{\min}^2}{\mathcal{O}^*(\delta_l^2)}\right) + 2s\exp\left(\frac{-1}{\mathcal{O}(\delta_l)}\right)$. With appropriately large $t$, $|\mathbf{z}_{t,(j)}^{(l)} - \mathbf{z}_{(j)}^*| = \mathcal{O}(\sqrt{s\|\mathbf{D}_j^{(l)} - \mathbf{D}_j^*\|_2} + \lambda^{fixed})$.*

*Proof.* We denote the regularization used in all layers $\lambda_1^j = \lambda_1^j = \cdots = \lambda_1^p = \lambda^{\text{fixed}}$. We assume that there exists such $\lambda^{\text{fixed}}$ that meets the lower bounds of regularization and also allow to pick an $\alpha > 0$ according to Theorem 4.2. We define $\tilde{\eta}_{t,j}^{(l)} \coloneqq \sum_{i \neq j} |\langle \boldsymbol{D}_j^{(l)}, \boldsymbol{D}_i^{(l)} \rangle| \|\boldsymbol{z}_{(i)}^* - \boldsymbol{z}_{t,(i)}| + \gamma_j^{(l)}$ and upper bound it as

$$\tilde{\eta}_{t,j}^{(l)} \leq \frac{\mu_l}{\sqrt{m}} \sum_{i \neq j} E_{t,i} + \gamma_j^{(l)} \tag{73}$$

where $E_{t,i} \coloneqq |\boldsymbol{z}_{(i)}^* - \boldsymbol{z}_{t,(i)}|$. Given (21), we re-write the code recursion

$$\boldsymbol{z}_{t+1,(j)} = \mathcal{P}_{\alpha \lambda_{t,j}^{(l)}}((1-\alpha)\boldsymbol{z}_{t,(j)} + \alpha(1 - \beta_j^{(l)})\boldsymbol{z}_{(j)}^* + \alpha \eta_{t,j}^{(l)})$$

$$\in (1-\alpha)\boldsymbol{z}_{t,(j)} + \alpha(1-\beta_j^{(l)})\boldsymbol{z}_{(j)}^* + \alpha(\eta_{t,j}^{(l)} - \lambda_{t,j}^{(l)} \partial |\boldsymbol{z}_{t+1,(j)}|) \tag{74}$$

$$E_{t+1,j} = |\boldsymbol{z}_{t+1,(j)} - \boldsymbol{z}_{(j)}^*| \leq (1-\alpha)E_{t,j} + \alpha \beta_j^{(l)} |\boldsymbol{z}_{(j)}^*| + \alpha(\tilde{\eta}_{t,j}^{(l)} + \lambda_{t,j}^{(l)})$$

Opening up the recursion, we get

$$E_{t+1,j} \leq E_{0,j} \prod_{q=0}^{t}(1-\alpha) + \beta_j^{(l)} |\boldsymbol{z}_{(j)}^*| \sum_{a=1}^{t+1} \alpha \prod_{q=a}^{t+1}(1-\alpha) + \sum_{a=1}^{t+1} \alpha(\tilde{\eta}_{a-1,j}^{(l)} + \lambda_{a-1,j}^{(l)}) \prod_{q=a}^{t+1}(1-\alpha) \tag{75}$$

where we define $\prod_{q=a}^{a}(1-\alpha) = 1$. Using the upper bounds from (73) and $\lambda_{t,j}^{(l)}$ from Theorem 4.2, we get

$$E_{t+1,j} \leq v_{t+1,j} + r_{t+1,j} + \sum_{a=1}^{t+1} \alpha \frac{\mu_l}{\sqrt{m}} \sum_{i \neq j} E_{a-1,i} \prod_{q=a}^{t+1}(1-\alpha) \tag{76}$$

where $v_{t+1,j} \coloneqq E_{0,j} \prod_{q=0}^{t}(1-\alpha) + (\beta_j^{(l)} |\boldsymbol{z}_{(j)}^*| + \gamma_j^{(l)}) \sum_{a=1}^{t+1} \alpha \prod_{q=a}^{t+1}(1-\alpha)$ and $r_{t+1,j} \coloneqq \sum_{a=1}^{t+1} \alpha \lambda_{a-1,j}^{(l)} \prod_{q=a}^{t+1}(1-\alpha)$. Following similar steps in Theorem 4.4, we now derive the general upper bound on $E_{t+1,j}$ as follows

$$E_{1,i_1} \leq v_{1,i_1} + r_{1,i_1} + \alpha \frac{\mu_l}{\sqrt{m}} \sum_{i_2 \neq i_1} E_{0,i_2} \tag{77}$$

For $E_{2,i_1}$, we have

$$E_{2,i_1} \leq v_{2,i_1} + r_{2,i_1} + \alpha \frac{\mu_l}{\sqrt{m}} \left( \sum_{i_2 \neq i_1} E_{1,i_2} + \sum_{i_2 \neq i_1} E_{0,i_2}(1-\alpha) \right) \tag{78}$$

Substituting $E_{1,i_1}$,

$$E_{2,i_1} \leq v_{2,i_1} + r_{2,i_1} + \alpha \frac{\mu_l}{\sqrt{m}} \left( \sum_{i_2 \neq i_1} (v_{1,i_2} + r_{1,i_2} + \alpha \frac{\mu_l}{\sqrt{m}} \sum_{i_3 \neq i_2} E_{0,i_3}) + \sum_{i_2 \neq i_1} E_{0,i_2}(1-\alpha) \right) \tag{79}$$

For $E_{3,i_1}$, we have

$$E_{3,i_1} \leq v_{3,i_1} + r_{3,i_1} + \alpha \frac{\mu_l}{\sqrt{m}} \sum_{a=1}^{3} \sum_{i_2 \neq i_1} E_{a-1,i_2}(1-\alpha)^{3-a} \tag{80}$$

Unrolling the recursion,

$$E_{3,i_1} \leq v_{3,i_1} + r_{3,i_1} + \alpha \frac{\mu_l}{\sqrt{m}} \sum_{i_2 \neq i_1} (v_{2,i_2} + r_{2,i_2}) + \alpha \frac{\mu_l}{\sqrt{m}} \left( (1-\alpha) \sum_{i_2 \neq i_1} (v_{1,i_2} + r_{1,i_2}) + \alpha \frac{\mu_l}{\sqrt{m}} \sum_{i_3 \neq i_2} (v_{1,i_3} + r_{1,i_3}) \right)$$

$$+ \alpha \frac{\mu_l}{\sqrt{m}} \left( (1-\alpha)^2 \sum_{i_2 \neq i_1} E_{0,i_2} + 2(1-\alpha)(\alpha \frac{\mu_l}{\sqrt{m}}) \sum_{i_3 \neq i_2, i_1} E_{0,i_3} + (\alpha \frac{\mu_l}{\sqrt{m}})^2 \sum_{i_4 \neq i_3, i_2, i_1} E_{0,i_4} \right) \tag{81}$$

Given above, we can write up the relation as

$$
\begin{aligned}
E_{3,i_1} \leq{} & v_{3,i_1} + r_{3,i_1} + \alpha\frac{\mu_l}{\sqrt{m}}\left((s-1)(v_{2,i}+r_{2,i}) + (v_{1,i}+r_{1,i})((1-\alpha)(s-1)+\alpha\frac{\mu_l}{\sqrt{m}}(s-2))\right) \\
& + \alpha\frac{\mu_l}{\sqrt{m}}E_{0,i}\left((s-1)(1-\alpha)^2 + 2(1-\alpha)(\alpha\frac{\mu_l}{\sqrt{m}})(s-2) + (\alpha\frac{\mu_l}{\sqrt{m}})^2(s-3)\right) \\
\leq{} & v_{3,i_1} + r_{3,i_1} + (s-1)\alpha\frac{\mu_l}{\sqrt{m}}\left(v_{2,i}+r_{2,i} + (v_{1,i}+r_{1,i})((1-\alpha)+\alpha\frac{\mu_l}{\sqrt{m}})\right) \\
& + (s-1)\alpha\frac{\mu_l}{\sqrt{m}}E_{0,i}\left((1-\alpha)^2 + 2(1-\alpha)(\alpha\frac{\mu_l}{\sqrt{m}}) + (\alpha\frac{\mu_l}{\sqrt{m}})^2\right)
\end{aligned}
\tag{82}
$$

We denote $u_{t+1,i} := v_{t+1,i} + r_{t+1,i}$ and following similar steps for $E_{4,i_1}$, we get,

$$
\begin{aligned}
E_{4,i_1} \leq{} & u_{4,i_1} + (s-1)\alpha\frac{\mu_l}{\sqrt{m}}\left(u_{3,i} + u_{2,i}(1-\alpha+\alpha\frac{\mu_l}{\sqrt{m}}) + u_{1,i}((1-\alpha)^2 + 2(1-\alpha)(\alpha\frac{\mu_l}{\sqrt{m}}) + (\alpha\frac{\mu_l}{\sqrt{m}})^2)\right) \\
& + (s-1)\alpha\frac{\mu_l}{\sqrt{m}}E_{0,i}\left((1-\alpha)^3 + 3(1-\alpha)(\alpha\frac{\mu_l}{\sqrt{m}})^2 + 3(1-\alpha)^2(\alpha\frac{\mu_l}{\sqrt{m}}) + (\alpha\frac{\mu_l}{\sqrt{m}})^3\right)
\end{aligned}
\tag{83}
$$

This leads to the term

$$
\begin{aligned}
E_{4,i_1} \leq{} & u_{4,i_1} + (s-1)\alpha\frac{\mu_l}{\sqrt{m}}\left(u_{3,i} + u_{2,i}(1-\alpha+\alpha\frac{\mu_l}{\sqrt{m}})^1 + u_{1,i}(1-\alpha+\alpha\frac{\mu_l}{\sqrt{m}})^2\right) \\
& + (s-1)\alpha\frac{\mu_l}{\sqrt{m}}E_{0,i}(1-\alpha+\alpha\frac{\mu_l}{\sqrt{m}})^3
\end{aligned}
\tag{84}
$$

Hence, the general term for code error at $t$ layer is

$$
E_{t+1,j} \leq u_{t+1,j} + (s-1)\alpha\frac{\mu_l}{\sqrt{m}}\sum_{a=1}^{t}(v_{a,\max}+r_{a,\max})(1-\alpha+\alpha\frac{\mu_l}{\sqrt{m}})^{t-a} + (s-1)\alpha\frac{\mu_l}{\sqrt{m}}E_{0,\max}(1-\alpha+\alpha\frac{\mu_l}{\sqrt{m}})^t
\tag{85}
$$

where for $j$ in the support, we define the upper bounds $v_{a,j} \leq v_{a,\max}$, $r_{a,j} \leq r_{a,\max}$, and $E_{0,j} \leq E_{0,\max}$. Next, we define $(1-\alpha)^t \leq \delta^{\text{fixed}}_{\alpha,t} := (1-\alpha+\alpha\frac{\mu_l}{\sqrt{m}})^t$, and use it to find an upper bound on the two expressions $\sum_{a=1}^{t} v_{a,\max}(1-\alpha+\alpha\frac{\mu_l}{\sqrt{m}})^{t-a}$ and $\sum_{a=1}^{t} r_{a,\max}(1-\alpha+\alpha\frac{\mu_l}{\sqrt{m}})^{t-a}$. The following bound can be achieved similar to the steps in Theorem 4.4

$$
\sum_{a=1}^{t} v_{a,\max}(1-\alpha+\alpha\frac{\mu_l}{\sqrt{m}})^{t-a} \leq E_{0,\max}t\delta^{\text{fixed}}_{\alpha,t} + \frac{1}{\alpha(1-\frac{\mu_l}{\sqrt{m}})}(\frac{\delta_l^2}{2}|z^*_{\max}| + a_\gamma)
\tag{86}
$$

Hence, we focus on $\sum_{a=1}^{t} r_{a,\max}(1-\alpha+\alpha\frac{\mu_l}{\sqrt{m}})^{t-a}$ next. First, we rewrite

$$
r_{t+1,j} = \sum_{a=1}^{t+1}\alpha\lambda^{(l)}_{a-1,j}\prod_{q=a}^{t+1}(1-\alpha)
\tag{87}
$$

We replace all $\lambda^{(l)}_{t,j}$ with a fixed one $\lambda^{\text{fixed}}$, and write

$$
\begin{aligned}
\sum_{a=1}^{t} r_{a,j}(1-\alpha+\alpha\frac{\mu_l}{\sqrt{m}})^{t-a} &\leq \sum_{a=1}^{t}\lambda^{\text{fixed}}\sum_{k=1}^{a}\alpha(1-\alpha)^{a-k+1})(1-\alpha+\alpha\frac{\mu_l}{\sqrt{m}})^{t-a} \\
&\leq \lambda^{\text{fixed}}\sum_{a=1}^{t}(1-\alpha+\alpha\frac{\mu_l}{\sqrt{m}})^{t-a}\sum_{k=1}^{a}\alpha(1-\alpha)^{a-k+1}
\end{aligned}
\tag{88}
$$

Using sum of geometric series, we write $\sum_{a=1}^{t}(1-\alpha+\alpha\frac{\mu_l}{\sqrt{m}})^{t-a} = \frac{1-(1-\alpha+\alpha\frac{\mu_l}{\sqrt{m}})^t}{\alpha-\alpha\frac{\mu_l}{\sqrt{m}}} \leq \frac{1}{\alpha(1-\frac{\mu_l}{\sqrt{m}})}$. Hence, we get

$$
\sum_{a=1}^{t} r_{a,\max}(1-\alpha+\alpha\frac{\mu_l}{\sqrt{m}})^{t-a} \leq \lambda^{\text{fixed}}\frac{1}{\alpha(1-\frac{\mu_l}{\sqrt{m}})}
\tag{89}
$$

Hence, we bound the code error on the coefficients as following

$$E_{t+1,j} \le u_{t+1,j} + (s-1)\frac{\mu_l}{\sqrt{m}}\left(\frac{1}{(1-\frac{\mu_l}{\sqrt{m}})}\left(\frac{\delta_l^2}{2}|z_{\max}^*| + a_\gamma + \lambda^{\text{fixed}}\right)\right) + (t+1)(s-1)\alpha\frac{\mu_l}{\sqrt{m}}E_{0,\max}\delta_{\alpha,t}^{\text{fixed}} \quad (90)$$

Next, we further simplify the first term. From before, we have

$$v_{t+1,j} = E_{0,j}(1-\alpha)^{t+1} + (\beta_j^{(l)}|z_{(j)}^*| + \gamma_j^{(l)})\sum_{k=1}^{t+1}\alpha(1-\alpha)^{t-k+1} \le E_{0,j}\delta_{\alpha,t+1}^{\text{fixed}} + \frac{\delta_l^2}{2}|z_{\max}^*| + a_\gamma \quad (91)$$

and

$$r_{t+1,j} = \alpha\lambda^{\text{fixed}}\sum_{k=1}^{t+1}(1-\alpha)^{t-k+1} \le \lambda^{\text{fixed}} \quad (92)$$

Substituting the above upper bound into the upper bound for $E_{t+1,j}$, we get

$$E_{t+1,j} \le E_{0,j}\delta_{\alpha,t+1}^{\text{fixed}} + (1 + (s-1)\kappa_l^{\text{fixed}})\left(\frac{\delta_l^2}{2}|z_{\max}^*| + a_\gamma + \lambda^{\text{fixed}}\right) + (t+1)(s-1)\alpha\frac{\mu_l}{\sqrt{m}}E_{0,\max}\delta_{\alpha,t}^{\text{fixed}} \quad (93)$$

where $\kappa_l^{\text{fixed}} := \frac{\mu_l}{\sqrt{m}}\left(\frac{1}{(1-\frac{\mu_l}{\sqrt{m}})}\right)$. Given $s = \mathcal{O}^*(\sqrt{m}/\mu\log m)$, we have $(s-1)\kappa_l^{\text{fixed}} < 1$. Hence, with probability of at least $1 - \epsilon_\gamma^{(l)}$, we have

$$E_{t,j} \le \mathcal{O}(a_\gamma + \lambda^{\text{fixed}}) + (s-1)t\alpha\frac{\mu_l}{\sqrt{m}}E_{0,\max}\delta_{\alpha,t-1}^{\text{fixed}} + E_{0,j}\delta_{\alpha,t}^{\text{fixed}}$$
$$|z_{t,(j)}^{(l)} - z_{(j)}^*| \le \mathcal{O}(\sqrt{s\|D_j^{(l)} - D_j^*\|_2} + \lambda^{\text{fixed}} + e_{t,j}^{(l)\text{unroll, fixed}}) \quad (94)$$

where $e_{t,j}^{(l)\text{unroll, fixed}} := (s-1)t\alpha\frac{\mu_l}{\sqrt{m}}\max_i|z_{0,(i)}^{(l)} - z_{(i)}^*|\delta_{\alpha,t-1}^{\text{fixed}} + |z_{0,(j)}^{(l)} - z_{(j)}^*|\delta_{\alpha,t}^{\text{fixed}}$. With appropriately large unrolled layer $t$, $e_{t,j}^{(l)\text{unroll, fixed}} \approx 0$. Hence, for the code error on non-zero coefficients, we get

$$|z_{t,(j)}^{(l)} - z_{(j)}^*| = \mathcal{O}(\sqrt{s\|D_j^{(l)} - D_j^*\|_2} + \lambda^{\text{fixed}}) \quad (95)$$

for large enough $t$. Now, we provide the relation for $z_{T,(j)}$. We re-write (35)

$$z_{T,(j)} \in z_{(j)}^*(1 - \beta_j^{(l)}) + \zeta_{T,j}^{(l)} \quad (96)$$

where $\zeta_{T,j}^{(l)} = \kappa_{T,j}^{(l)} + \sum_{t=1}^T\alpha(\eta_{t-1,j}^{(l)} - \lambda_{t-1,j}^{(l)}\partial|z_{t,(j)}|)(1-\alpha)^{T-t}$ and $\kappa_{T,j}^{(l)} := (1-\alpha)^T(z_{0,(j)} - z_{(j)}^*(1-\beta_j^{(l)}))$. $\kappa_{T,j}^{(l)}$ decays very fast as $T$ increases. Hence, we bound the second term. We substitute $\eta_{t,j}^{(l)} = \sum_{i \ne j}\langle D_j^{(l)}, D_i^{(l)}\rangle(z_{(i)}^* - z_{t,(i)}) + \gamma_j^{(l)}$ in $\zeta_{T,j}^{(l)}$.

$$\begin{aligned}
\zeta_{T,j}^{(l)} &\in \kappa_{T,j}^{(l)} + \sum_{t=1}^T\alpha(\eta_{t-1,j}^{(l)} - \lambda_{t-1,j}^{(l)}\partial|z_{t,(j)}|)(1-\alpha)^{T-t} \\
&\in \kappa_{T,j}^{(l)} + \sum_{t=1}^T\alpha\left(\sum_{i \ne j}\langle D_j^{(l)}, D_i^{(l)}\rangle(z_{(i)}^* - z_{t-1,(i)}) + \gamma_j^{(l)} - \lambda_{t-1,j}^{(l)}\partial|z_{t,(j)}|\right)(1-\alpha)^{T-t} \\
&\le \kappa_{T,j}^{(l)} + (\gamma_j^{(l)} + \lambda^{\text{fixed}})\sum_{t=1}^T\alpha(1-\alpha)^{T-t} + \alpha\frac{\mu_l}{\sqrt{m}}\sum_{t=1}^T\sum_{i \ne j}E_{t-1,j}(1-\alpha)^{T-t} \\
&\le \kappa_{T,j}^{(l)} + \gamma_j^{(l)} + \lambda^{\text{fixed}} + (s-1)\alpha\frac{\mu_l}{\sqrt{m}}\sum_{t=1}^T E_{t-1,j}(1-\alpha)^{T-t}
\end{aligned} \quad (97)$$

Given above, we find an upper bound on $E_{t-1,j}(1-\alpha)^{T-t}$ below. From analysis in [Theorem 4.4](#), we have

$$
E_{t-1,j} \le v_{t-1,j} + (s-1)\alpha\frac{\mu_l}{\sqrt{m}}\sum_{a=1}^{t-2}v_{a,\max}(1-\alpha+\alpha\frac{\mu_l}{\sqrt{m}})^{t-a-2}
$$
$$
+ (s-1)\alpha\frac{\mu_l}{\sqrt{m}}E_{0,\max}(1-\alpha+\alpha\frac{\mu_l}{\sqrt{m}})^{t-2}
$$
$$
E_{t-1,j}(1-\alpha)^{T-t} \le v_{t-1,j}(1-\alpha)^{T-t} + 2(s-1)\alpha\frac{\mu_l}{\sqrt{m}}\sum_{a=1}^{t-2}v_{a,\max}(1-\alpha+\alpha\frac{\mu_l}{\sqrt{m}})^{t-a-2}(1-\alpha)^{T-t}
$$
$$
+ (s-1)\alpha\frac{\mu_l}{\sqrt{m}}E_{0,\max}(1-\alpha+\alpha\frac{\mu_l}{\sqrt{m}})^{t-2}(1-\alpha)^{T-t}
\tag{98}
$$

Re-write the first term,

$$
v_{t-1,j}(1-\alpha)^{T-t} = (E_{0,j}(1-\alpha)^{t-1} + (\beta_j^{(l)}|\boldsymbol{z}_{(j)}^*| + \gamma_j^{(l)} + \lambda^{\text{fixed}})\sum_{k=1}^{t-1}\alpha(1-\alpha)^{t-k-1})(1-\alpha)^{T-t}
$$
$$
= E_{0,j}(1-\alpha)^{T-1} + (\beta_j^{(l)}|\boldsymbol{z}_{(j)}^*| + \gamma_j^{(l)} + \lambda^{\text{fixed}})\sum_{k=1}^{t-1}\alpha(1-\alpha)^{T-k-1}
\tag{99}
$$

Similarly,

$$
v_{a,\max} = E_{0,\max}(1-\alpha)^a + (\beta_{\max}^{(l)}|\boldsymbol{z}_{\max}^*| + \gamma_{\max}^{(l)} + \lambda^{\text{fixed}})\sum_{k=1}^{a}\alpha(1-\alpha)^{a-k}
\tag{100}
$$

We write

$$
\sum_{a=1}^{t-2}v_{a,\max}(1-\alpha+\alpha\frac{\mu_l}{\sqrt{m}})^{t-a-2} = \sum_{a=1}^{t-2}E_{0,\max}(1-\alpha)^a(1-\alpha+\alpha\frac{\mu_l}{\sqrt{m}})^{t-a-2}
$$
$$
+ \sum_{a=1}^{t-2}(\beta_{\max}^{(l)}|\boldsymbol{z}_{\max}^*| + \gamma_{\max}^{(l)} + \lambda^{\text{fixed}})\sum_{k=1}^{a}\alpha(1-\alpha)^{a-k}(1-\alpha+\alpha\frac{\mu_l}{\sqrt{m}})^{t-a-2}
$$
$$
\le \sum_{a=1}^{t-2}E_{0,\max}(1-\alpha+\alpha\frac{\mu_l}{\sqrt{m}})^{t-2}
$$
$$
+ (\beta_{\max}^{(l)}|\boldsymbol{z}_{\max}^*| + \gamma_{\max}^{(l)} + \lambda^{\text{fixed}})\sum_{a=1}^{t-2}(1-\alpha+\alpha\frac{\mu_l}{\sqrt{m}})^{t-a-2}
\tag{101}
$$

Hence,

$$
(1-\alpha)^{T-t}\sum_{a=1}^{t-2}v_{a,\max}(1-\alpha+\alpha\frac{\mu_l}{\sqrt{m}})^{t-a-2} \le \sum_{a=1}^{t-2}E_{0,\max}(1-\alpha+\alpha\frac{\mu_l}{\sqrt{m}})^{t-2}(1-\alpha)^{T-t}
$$
$$
+ (\beta_{\max}^{(l)}|\boldsymbol{z}_{\max}^*| + \gamma_{\max}^{(l)} + \lambda^{\text{fixed}})\sum_{a=1}^{t-2}(1-\alpha+\alpha\frac{\mu_l}{\sqrt{m}})^{t-a-2}(1-\alpha)^{T-t}
$$
$$
\le (t-2)E_{0,\max}(1-\alpha+\alpha\frac{\mu_l}{\sqrt{m}})^{T-2}
$$
$$
+ (\beta_{\max}^{(l)}|\boldsymbol{z}_{\max}^*| + \gamma_{\max}^{(l)} + \lambda^{\text{fixed}})\frac{(1-\alpha)^{T-t}}{\alpha(1-\frac{\mu_l}{\sqrt{m}})}
\tag{102}
$$

Combining all terms, we get

$$
\begin{aligned}
E_{t-1,j}(1-\alpha)^{T-t} &\le v_{t-1,j}(1-\alpha)^{T-t} + (s-1)\alpha\frac{\mu_l}{\sqrt{m}}\sum_{a=1}^{t-2}v_{a,\max}(1-\alpha+\alpha\frac{\mu_l}{\sqrt{m}})^{t-a-2}(1-\alpha)^{T-t} \\
&\quad + (s-1)\alpha\frac{\mu_l}{\sqrt{m}}E_{0,\max}(1-\alpha+\alpha\frac{\mu_l}{\sqrt{m}})^{t-2}(1-\alpha)^{T-t} \\
&\le E_{0,j}(1-\alpha)^{T-1} + (s-1)\alpha\frac{\mu_l}{\sqrt{m}}(t-1)E_{0,\max}\delta_{\alpha,T-2} \\
&\quad + (\beta_{\max}^{(l)}|z_{\max}^*| + \gamma_{\max}^{(l)} + \lambda^{\text{fixed}})\left(\sum_{k=1}^{t-1}\alpha(1-\alpha)^{T-k-1} + (s-1)\kappa_l(1-\alpha)^{T-t}\right) \\
&\le (1+(s-1)\alpha\frac{\mu_l}{\sqrt{m}}(t-1))E_{0,\max}\delta_{\alpha,T-2} \\
&\quad + (\beta_{\max}^{(l)}|z_{\max}^*| + \gamma_{\max}^{(l)} + \lambda^{\text{fixed}})\left(\sum_{k=1}^{t-1}\alpha(1-\alpha)^{T-k-1} + (s-1)\kappa_l(1-\alpha)^{T-t}\right)
\end{aligned}
\tag{103}
$$

where $\kappa_l^{\text{fixed}} = \frac{\frac{\mu_l}{\sqrt{m}}}{(1-\frac{\mu_l}{\sqrt{m}})}$. Moreover, we bound

$$
\sum_{t=1}^{T}\sum_{k=1}^{t-1}\alpha(1-\alpha)^{T-k-1} = \sum_{t=1}^{T}\alpha(1-\alpha)^{T-t}\frac{1-(1-\alpha)^{t-1}}{\alpha} \le \sum_{t=1}^{T}(1-\alpha)^{T-t} \le \frac{1}{\alpha}
\tag{104}
$$

Finally, we are ready to write the bound for $\zeta_{T,j}^{(l)}$

$$
\begin{aligned}
|\zeta_{T,j}^{(l)}| &\le \kappa_{T,j}^{(l)} + |\gamma_j^{(l)}| + \lambda^{\text{fixed}} + (s-1)\frac{\mu_l}{\sqrt{m}}(\beta_{\max}^{(l)}|z_{\max}^*| + \gamma_{\max}^{(l)} + \lambda^{\text{fixed}})(1+(s-1)\kappa_l^{\text{fixed}}) \\
&\quad + \sum_{t=1}^{T}(s-1)\alpha\frac{\mu_l}{\sqrt{m}}((1+(s-1)\alpha\frac{\mu_l}{\sqrt{m}}(t-1))E_{0,\max}\delta_{\alpha,T-2})
\end{aligned}
\tag{105}
$$

Given $|\gamma_j^{(l)}| = a_\gamma$ with probability of $1-\epsilon_\gamma^{(l)}$ where $a_\gamma = \sqrt{s\delta_l}$ and $s = \mathcal{O}^*(\sqrt{m}/\mu\log m)$, we will have

$$
|\zeta_{T,j}^{(l)}| \le \mathcal{O}(\sqrt{s}\|D_j^{(l)} - D_j^*\|_2 + \lambda^{\text{fixed}})
\tag{106}
$$

$\blacksquare$

We now re-state the forward pass Jacobian (Theorem 4.7) convergence.

**Theorem 4.7** (Local forward pass Jacobian convergence). *Given the recursion $z_{t+1} = \Phi(z_t, D)$, and $\hat{z}$ the unique minimizer of lasso with Jacobian $\hat{J}$, then $\exists\ \rho < 1, B > 0$ s.t. $\|J_t - \hat{J}\|_2 \le \mathcal{O}(t\rho^t)\ \forall t > B$. Furthermore, given Theorem 4.1 and Theorem 4.2, $B = 1$.*

*Proof.* Differentiating the recursion,

$$
J_{t+1} = \nabla_1\Phi(z_t, D)^{\mathrm{T}}J_t + \nabla_2\Phi(z_t, D)^{\mathrm{T}}.
$$

Similarly,

$$
\hat{J} = \nabla_1\Phi(\hat{z}, D)^{\mathrm{T}}\hat{J} + \nabla_2\Phi(\hat{z}, D)^{\mathrm{T}}
$$

where $\hat{z}$ is a minimizer of lasso and fixed-point of the mapping (see Lemma 3.2). Subtract the terms

$$
J_{t+1} - \hat{J} = \nabla_1\Phi(z_t, D)^{\mathrm{T}}(J_t - \hat{J}) + (\nabla_1\Phi(z_t, D) - \nabla_1\Phi(\hat{z}, D))^{\mathrm{T}}\hat{J} + (\nabla_2\Phi(z_t, D) - \nabla_2\Phi(\hat{z}, D))^{\mathrm{T}}
$$

Given the Lipschitz properties of $\mathcal{L}$ and $h$, we can further get the upper bounds on $\|\nabla_1 \Phi(\boldsymbol{a}, \boldsymbol{D}) - \nabla_1 \Phi(\boldsymbol{b}, \boldsymbol{D})\|_2 \leq L_{\Phi_1} \|\boldsymbol{b} - \boldsymbol{a}\|_2$ and $\|\nabla_2 \Phi(\boldsymbol{a}, \boldsymbol{D}) - \nabla_2 \Phi(\boldsymbol{b}, \boldsymbol{D})\|_2 \leq L_{\Phi_2} \|\boldsymbol{b} - \boldsymbol{a}\|_2$. Hence, with upper bound on the norm of Jacobian (Assumption 4.1), there exists $B > 0$ such that $\forall t > B$

$$\begin{aligned}
\|\boldsymbol{J}_{t+1} - \hat{\boldsymbol{J}}\|_2 &\leq \|\nabla_1 \Phi(\boldsymbol{z}_t, \boldsymbol{D})\|_2 \|\boldsymbol{J}_t - \hat{\boldsymbol{J}}\|_2 + \|\nabla_1 \Phi(\boldsymbol{z}_t, \boldsymbol{D}) - \nabla_1 \Phi(\hat{\boldsymbol{z}}, \boldsymbol{D})\|_2 \|\hat{\boldsymbol{J}}\|_2 \\
&\quad + \|\nabla_2 \Phi(\boldsymbol{z}_t, \boldsymbol{D}) - \nabla_2 \Phi(\hat{\boldsymbol{z}}, \boldsymbol{D})\|_2 \\
&\leq \rho \|\boldsymbol{J}_t - \hat{\boldsymbol{J}}\|_2 + c \|\boldsymbol{z}_t - \hat{\boldsymbol{z}}\|_2
\end{aligned}$$

where $c \triangleq M_J L_{\Phi_1} + L_{\Phi_2}$. Hence,

$$\|\boldsymbol{J}_{t+1} - \hat{\boldsymbol{J}}\|_2 \leq \rho \|\boldsymbol{J}_t - \hat{\boldsymbol{J}}\|_2 + \mathcal{O}(\rho^t).$$

Unrolling the recursion,

$$\|\boldsymbol{J}_{t+1} - \hat{\boldsymbol{J}}\|_2 \leq \mathcal{O}((t+1)\rho^t).$$

∎

**Theorem 4.6** (Global forward pass code error)**.** *Let $\hat{\boldsymbol{z}}$ be the fixed-point of the encoder with iterations $\boldsymbol{z}_{t+1} = \Phi(\boldsymbol{z}_t, \boldsymbol{D})$. Given Assumption 3.6, Lemmas 3.2, A.1 and A.2, we have $\|\hat{\boldsymbol{z}} - \boldsymbol{z}^*\|_2 \leq \mathcal{O}(\|\boldsymbol{D} - \boldsymbol{D}^*\|_2 + \hat{\delta}^*)$, where $\hat{\delta}^* = \|\hat{\boldsymbol{z}}^* - \boldsymbol{z}^*\|_2$, $\hat{\boldsymbol{z}}$ is the unique minimizer of lasso (1) given the dictionary $\boldsymbol{D}$, $\hat{\boldsymbol{z}}^*$ is the unique minimizer of lasso (1) given the dictionary $\boldsymbol{D}^*$, and $\boldsymbol{z}^*$ is the ground-truth code.*

*Proof.* We first find the error between $\hat{\boldsymbol{z}}$ and $\hat{\boldsymbol{z}}^*$ which is the unique minimizer of lasso (1) given the true dictionary $\boldsymbol{D}^*$. Using fixed-point property (Lemma 3.2), we get

$$\|\hat{\boldsymbol{z}} - \hat{\boldsymbol{z}}^*\|_2 = \|\Phi(\hat{\boldsymbol{z}}, \boldsymbol{D}) - \Phi(\hat{\boldsymbol{z}}^*, \boldsymbol{D}^*)\|_2 \leq \|\Phi(\hat{\boldsymbol{z}}, \boldsymbol{D}) - \Phi(\hat{\boldsymbol{z}}^*, \boldsymbol{D})\|_2 + \|\Phi(\hat{\boldsymbol{z}}^*, \boldsymbol{D}) - \Phi(\hat{\boldsymbol{z}}^*, \boldsymbol{D}^*)\|_2 \tag{107}$$

Using the $\mu$-strongly convexity of $\mathcal{L}_{\boldsymbol{x}}(\boldsymbol{z}_t, \boldsymbol{D})$ on the support, and $L_{21}$ Lipschitz constants of $\nabla_{21}^2 \mathcal{L}_{\boldsymbol{x}}(\boldsymbol{z}, \boldsymbol{D})$, we upper bound the term as follows:

$$\|\hat{\boldsymbol{z}} - \hat{\boldsymbol{z}}^*\|_2 \leq \rho \|\hat{\boldsymbol{z}} - \hat{\boldsymbol{z}}^*\|_2 + \alpha L_{21} c_{\text{prox}} \|\boldsymbol{D} - \boldsymbol{D}^*\|_2 \tag{108}$$

Where $\rho \triangleq c_{\text{prox}}(1 - \alpha\mu) < 1$. Denote $q \triangleq \frac{\alpha c_{\text{prox}} L_{21}}{1 - \rho}$ which can be made to be small with proper choice of step size $\alpha$.

$$\|\hat{\boldsymbol{z}} - \hat{\boldsymbol{z}}^*\|_2 \leq q \|\boldsymbol{D} - \boldsymbol{D}^*\|_2 \tag{109}$$

Hence, we get the following code error

$$\|\hat{\boldsymbol{z}} - \boldsymbol{z}^*\|_2 \leq \|\hat{\boldsymbol{z}} - \hat{\boldsymbol{z}}^*\|_2 + \|\hat{\boldsymbol{z}}^* - \boldsymbol{z}^*\|_2 \leq q \|\boldsymbol{D} - \boldsymbol{D}^*\|_2 + \hat{\delta}^* \leq \mathcal{O}(\|\boldsymbol{D} - \boldsymbol{D}^*\|_2 + \hat{\delta}^*) \tag{110}$$

∎

**Theorem 4.8** (Global forward pass Jacobian error)**.** *Let $\hat{\boldsymbol{z}}$ be the fixed-point of the encoder with iterations $\boldsymbol{z}_{t+1} = \Phi(\boldsymbol{z}_t, \boldsymbol{D})$. Given Assumption 3.6, Lemmas 3.2, A.1 and A.2, we have $\|\hat{\boldsymbol{J}} - \boldsymbol{J}^*\|_2 \leq \mathcal{O}(\|\boldsymbol{D} - \boldsymbol{D}^*\|_2 + \hat{\delta}_J^*)$, where $\hat{\delta}_J^* := \|\hat{\boldsymbol{J}}^* - \boldsymbol{J}^*\|_2$, and $\hat{\boldsymbol{J}}$, $\hat{\boldsymbol{J}}^*$ and $\boldsymbol{J}^*$ are Jacobians corresponding to $\hat{\boldsymbol{z}}$, $\hat{\boldsymbol{z}}^*$ and $\boldsymbol{z}^*$.*

*Proof.* First, we define $\boldsymbol{J}^*$. For $\boldsymbol{z}^*$, we define the mapping function $\boldsymbol{z} \to \mathcal{L}_{\boldsymbol{x}}(\boldsymbol{z}, \boldsymbol{D})$, where $\boldsymbol{z}^*(\boldsymbol{D})$ is its minimizer evaluated at $\boldsymbol{D}^*$, i.e., $\nabla_1 \mathcal{L}_{\boldsymbol{x}}(\boldsymbol{z}^*, \boldsymbol{D}^*) = \boldsymbol{D}^{*\mathrm{T}}(\boldsymbol{D}^*\boldsymbol{z}^* - \boldsymbol{x}) = 0$ given the generative model ($\boldsymbol{x} = \boldsymbol{D}^*\boldsymbol{z}^*$). Hence, we define the Jacobian $\boldsymbol{J}^* = \frac{\partial \boldsymbol{z}^*(\boldsymbol{D})}{\partial \boldsymbol{D}}|_{\boldsymbol{D}=\boldsymbol{D}^*}$. From implicit function theorem, we get

$$\boldsymbol{J}^{*+} \nabla_{11}^2 \mathcal{L}_{\boldsymbol{x}}(\boldsymbol{z}^*, \boldsymbol{D}^*) + \nabla_{21}^2 \mathcal{L}_{\boldsymbol{x}}(\boldsymbol{z}^*, \boldsymbol{D}^*) = \boldsymbol{0}$$

which is later used in the global backward pass analysis. Alternatively, if $\nabla_{11}^2 \mathcal{L}_{\boldsymbol{x}}(\boldsymbol{z}^*, \boldsymbol{D}^*)$ is invertible, then we can compute $\boldsymbol{J}^{*+}$ as follows:

$$\boldsymbol{J}^{*+} = -\nabla_{21}^2 \mathcal{L}_{\boldsymbol{x}}(\boldsymbol{z}^*, \boldsymbol{D}^*) \nabla_{11}^2 \mathcal{L}_{\boldsymbol{x}}(\boldsymbol{z}^*, \boldsymbol{D}^*)^{-1}$$

The Jacobian w.r.t row $i$ of the dictionary is

$$\boldsymbol{J}^*_{(i,:)} = -(\boldsymbol{D}^{*\mathrm{T}}_{S^*}\boldsymbol{D}^*_{S^*})^{-1}(\boldsymbol{D}^*_{i,:}\boldsymbol{z}^{*\mathrm{T}} + (\boldsymbol{D}^{*\mathrm{T}}_{i,:}\boldsymbol{z}^* - \boldsymbol{x}_i)\boldsymbol{I}_p)_{S^*}$$

on the support $S^*$ of $\boldsymbol{z}^*$. Outside of the support, it is zero. Now, given the recursion $\boldsymbol{z}_{t+1} = \Phi(\boldsymbol{z}_t, \boldsymbol{D})$, we differentiate the recursion,

$$\boldsymbol{J}_{t+1} = \nabla_1\Phi(\boldsymbol{z}_t, \boldsymbol{D})^{\mathrm{T}}\boldsymbol{J}_t + \nabla_2\Phi(\boldsymbol{z}_t, \boldsymbol{D})^{\mathrm{T}}.$$

Hence, we have

$$\hat{\boldsymbol{J}} = \nabla_1\Phi(\hat{\boldsymbol{z}}, \boldsymbol{D})^{\mathrm{T}}\hat{\boldsymbol{J}} + \nabla_2\Phi(\hat{\boldsymbol{z}}, \boldsymbol{D})^{\mathrm{T}}$$

$$\hat{\boldsymbol{J}}^* = \nabla_1\Phi(\hat{\boldsymbol{z}}^*, \boldsymbol{D}^*)^{\mathrm{T}}\hat{\boldsymbol{J}}^* + \nabla_2\Phi(\hat{\boldsymbol{z}}^*, \boldsymbol{D}^*)^{\mathrm{T}}$$

where $\hat{\boldsymbol{J}}^*$ is the Jacobian of $\hat{\boldsymbol{z}}^*$. Then, following similar step to Theorem 4.7, we can write

$$\hat{\boldsymbol{J}} - \hat{\boldsymbol{J}}^* = \nabla_1\Phi(\hat{\boldsymbol{z}}, \boldsymbol{D})^{\mathrm{T}}(\hat{\boldsymbol{J}} - \hat{\boldsymbol{J}}^*) + (\nabla_1\Phi(\hat{\boldsymbol{z}}, \boldsymbol{D}) - \nabla_1\Phi(\hat{\boldsymbol{z}}^*, \boldsymbol{D}^*))^{\mathrm{T}}\hat{\boldsymbol{J}}^* + (\nabla_2\Phi(\hat{\boldsymbol{z}}, \boldsymbol{D}) - \nabla_2\Phi(\hat{\boldsymbol{z}}^*, \boldsymbol{D}^*))^{\mathrm{T}}$$

With respect to $\boldsymbol{D}$, we denote the Lipschitz constants of $\nabla_1\Phi(\hat{\boldsymbol{z}}, \boldsymbol{D})$ and $\nabla_2\Phi(\hat{\boldsymbol{z}}, \boldsymbol{D})$ with $L_{\Phi_{1D}}$ and $L_{\Phi_{2D}}$, respectively. Then,

$$\|\hat{\boldsymbol{J}} - \hat{\boldsymbol{J}}^*\|_2 \le \rho\|\hat{\boldsymbol{J}} - \hat{\boldsymbol{J}}^*\|_2 + c\|\hat{\boldsymbol{z}} - \hat{\boldsymbol{z}}^*\|_2 + c_D\|\hat{\boldsymbol{D}} - \boldsymbol{D}^*\|_2$$

where $c \triangleq M_J L_{\Phi_1} + L_{\Phi_2}$ and $c_D \triangleq M_J L_{\Phi_{1D}} + L_{\Phi_{2D}}$. Given the global forward pass code error, we get

$$\|\hat{\boldsymbol{J}} - \hat{\boldsymbol{J}}^*\|_2 \le q_z\|\hat{\boldsymbol{z}} - \hat{\boldsymbol{z}}^*\|_2 + q_D\|\boldsymbol{D} - \boldsymbol{D}^*\|_2 \le (q_D + q_z q)\|\boldsymbol{D} - \boldsymbol{D}^*\|_2 \tag{111}$$

where $q_z \triangleq \frac{c}{1-\rho}$, $q_D \triangleq \frac{c_D}{1-\rho}$. Hence, we get

$$\|\hat{\boldsymbol{J}} - \boldsymbol{J}^*\|_2 \le \|\hat{\boldsymbol{J}} - \hat{\boldsymbol{J}}^*\|_2 + \|\hat{\boldsymbol{J}}^* - \boldsymbol{J}^*\|_2 \le \mathcal{O}(\|\boldsymbol{D} - \boldsymbol{D}^*\|_2 + \hat{\delta}^*_J) \tag{112}$$

where we denote $\hat{\delta}^*_J := \|\hat{\boldsymbol{J}}^* - \boldsymbol{J}^*\|_2$ ∎

## A.4 Local backward pass proof details

In each update of the dictionary, we bound the gradient approximations as function of unrolling $t$ (Theorem 4.9). This shows that $\boldsymbol{g}_t^{\text{ae-lasso}}$ converges faster than $\boldsymbol{g}_t^{\text{dec}}$ and $\boldsymbol{g}_t^{\text{ae-ls}}$, and the latter is a biased estimator of $\hat{\boldsymbol{g}}$. This is followed by Theorem 4.9 showing the order magnitude of the bounds is indeed tight.

**Theorem 4.9** (Local convergence of gradients). *Given the forward pass convergence results (Theorems 4.3 and 4.7), $\exists\, \rho < 1, B > 0$ such that $\forall t > B$, the errors of gradients defined in Algorithm 2 w.r.t $\hat{\boldsymbol{g}}$ (4) satisfy*

$$\|\boldsymbol{g}_t^{dec} - \hat{\boldsymbol{g}}\|_2 \le \mathcal{O}(\rho^t)$$
$$\|\boldsymbol{g}_t^{ae\text{-}lasso} - \hat{\boldsymbol{g}}\|_2 \le \mathcal{O}(t\rho^{2t}) \tag{12}$$
$$\|\boldsymbol{g}_t^{ae\text{-}ls} - \hat{\boldsymbol{g}}\|_2 \le \mathcal{O}(t\rho^{2t} + M_J\lambda\sqrt{s}).$$

*Moreover, the order of upper bounds is tight (see Lemma A.4).*

*Proof.* For $\boldsymbol{g}_t^{\text{dec}}$, with the infinite fresh samples, we have $\lim_{n\to\infty}\frac{1}{n}\sum_{i=1}^n \nabla_2\mathcal{L}_{\boldsymbol{x}^i}(\boldsymbol{z}_t^i, \boldsymbol{D}) = \mathbb{E}_{\boldsymbol{x}\in\mathcal{X}}[\nabla_2\mathcal{L}_{\boldsymbol{x}}(\boldsymbol{z}_t, \boldsymbol{D})]$ a.s. Based on Lemma 3.3, we get

$$\begin{aligned}\|\boldsymbol{g}_t^{\text{dec}} - \hat{\boldsymbol{g}}\|_2 &= \|\mathbb{E}_{\boldsymbol{x}\in\mathcal{X}}[\nabla_2\mathcal{L}_{\boldsymbol{x}}(\boldsymbol{z}_t, \boldsymbol{D})] - \mathbb{E}_{\boldsymbol{x}\in\mathcal{X}}[\nabla_2\mathcal{L}_{\boldsymbol{x}}(\hat{\boldsymbol{z}}, \boldsymbol{D})]\|_2 \\ &\le \mathbb{E}_{\boldsymbol{x}\in\mathcal{X}}[\|\nabla_2\mathcal{L}_{\boldsymbol{x}}(\boldsymbol{z}_t, \boldsymbol{D}) - \nabla_2\mathcal{L}_{\boldsymbol{x}}(\hat{\boldsymbol{z}}, \boldsymbol{D})\|_2] \le \mathbb{E}_{\boldsymbol{x}\in\mathcal{X}}[L_2\|\boldsymbol{z}_t - \hat{\boldsymbol{z}}\|_2] \le \mathcal{O}(\rho^t).\end{aligned} \tag{113}$$

Similarly, for $\boldsymbol{g}_t^{\text{ae-lasso}}$ and $\boldsymbol{g}_t^{\text{ae-ls}}$, we replace the sample mean for gradient computations with expectation in their limit. We re-write the gradient estimation error as following

$$\begin{aligned}\boldsymbol{g}_t^{\text{ae-lasso}} - \hat{\boldsymbol{g}} &= \mathbb{E}_{\boldsymbol{x}\in\mathcal{X}}[Q(\hat{\boldsymbol{z}}, \boldsymbol{J}_t)(\boldsymbol{z}_t - \hat{\boldsymbol{z}})] + \mathbb{E}_{\boldsymbol{x}\in\mathcal{X}}[Q_t^{21}(\hat{\boldsymbol{z}})] + \mathbb{E}_{\boldsymbol{x}\in\mathcal{X}}[\boldsymbol{J}_t Q_t^{\text{lasso-11}}(\hat{\boldsymbol{z}})] \\ \boldsymbol{g}_t^{\text{ae-ls}} - \hat{\boldsymbol{g}} &= \mathbb{E}_{\boldsymbol{x}\in\mathcal{X}}[Q(\hat{\boldsymbol{z}}, \boldsymbol{J}_t)(\boldsymbol{z}_t - \hat{\boldsymbol{z}})] + \mathbb{E}_{\boldsymbol{x}\in\mathcal{X}}[Q_t^{21}(\hat{\boldsymbol{z}})] + \mathbb{E}_{\boldsymbol{x}\in\mathcal{X}}[\boldsymbol{J}_t Q_t^{\text{ls-11}}(\hat{\boldsymbol{z}})]\end{aligned} \tag{114}$$

where

$$
\begin{aligned}
Q_t^{21}(\boldsymbol{z}) &\triangleq \nabla_2 \mathcal{L}_{\boldsymbol{x}}(\boldsymbol{z}_t, \boldsymbol{D}) - \nabla_2 \mathcal{L}_{\boldsymbol{x}}(\boldsymbol{z}, \boldsymbol{D}) - \nabla_{21}^2 \mathcal{L}_{\boldsymbol{x}}(\boldsymbol{z}, \boldsymbol{D})(\boldsymbol{z}_t - \boldsymbol{z}) \\
Q_t^{\text{lasso-11}}(\boldsymbol{z}) &\triangleq \nabla_1 \mathcal{L}_{\boldsymbol{x}}(\boldsymbol{z}_t, \boldsymbol{D}) + \partial h(\boldsymbol{z}_t) - \nabla_{11}^2 \mathcal{L}_{\boldsymbol{x}}(\boldsymbol{z}, \boldsymbol{D})(\boldsymbol{z}_t - \boldsymbol{z}) \\
Q_t^{\text{ls-11}}(\boldsymbol{z}) &\triangleq \nabla_1 \mathcal{L}_{\boldsymbol{x}}(\boldsymbol{z}_t, \boldsymbol{D}) - \nabla_{11}^2 \mathcal{L}_{\boldsymbol{x}}(\boldsymbol{z}, \boldsymbol{D})(\boldsymbol{z}_t - \boldsymbol{z}) \\
Q(\boldsymbol{z}, \boldsymbol{J}) &\triangleq \boldsymbol{J}^+ \nabla_{11}^2 \mathcal{L}_{\boldsymbol{x}}(\boldsymbol{z}, \boldsymbol{D}) + \nabla_{21}^2 \mathcal{L}_{\boldsymbol{x}}(\boldsymbol{z}, \boldsymbol{D}).
\end{aligned} \tag{115}
$$

We provide bounds on the above in Lemma A.5. Hence, it suffices to bound the terms on the *r.h.s* as follows:

$$
\|\boldsymbol{g}_t^{\text{ae-lasso}} - \hat{\boldsymbol{g}}\|_2 \leq \mathbb{E}_{\boldsymbol{x} \in \mathcal{X}} \left[ L_1 \|\boldsymbol{J}_t - \hat{\boldsymbol{J}}\|_2 \|\boldsymbol{z}_t - \hat{\boldsymbol{z}}\|_2 + (L_{21}/2) \|\boldsymbol{z}_t - \hat{\boldsymbol{z}}\|_2^2 + M_J(L_{11}/2) \|\boldsymbol{z}_t - \hat{\boldsymbol{z}}\|_2^2 \right]. \tag{116}
$$

Using the convergence errors from the forward pass (Theorems 4.3 and 4.7),

$$
\|\boldsymbol{g}_t^{\text{ae-lasso}} - \hat{\boldsymbol{g}}\|_2 \leq L_1 \mathcal{O}(t\rho^{2t}) + (L_{21}/2 + M_J(L_{11}/2)) \mathcal{O}(\rho^{2t}) = \mathcal{O}(t\rho^{2t}). \tag{117}
$$

Similarly,

$$
\|\boldsymbol{g}_t^{\text{ae-ls}} - \hat{\boldsymbol{g}}\|_2 \leq \mathbb{E}_{\boldsymbol{x} \in \mathcal{X}} \left[ L_1 \|\boldsymbol{J}_t - \hat{\boldsymbol{J}}\|_2 \|\boldsymbol{z}_t - \hat{\boldsymbol{z}}\|_2 + (L_{21}/2) \|\boldsymbol{z}_t - \hat{\boldsymbol{z}}\|_2^2 + M_J((L_{11}/2) \|\boldsymbol{z}_t - \hat{\boldsymbol{z}}\|_2^2 + \|\partial h(\hat{\boldsymbol{z}})\|_2) \right]. \tag{118}
$$

Using the convergence errors from the forward pass (Theorems 4.3 and 4.7),

$$
\|\boldsymbol{g}_t^{\text{ae-ls}} - \hat{\boldsymbol{g}}\|_2 \leq L_1 \mathcal{O}(t\rho^{2t}) + (L_{21}/2 + M_J L_{11}/2) \mathcal{O}(\rho^{2t}) + M_J \|\partial h(\hat{\boldsymbol{z}})\|_2 = \mathcal{O}(t\rho^{2t} + M_J \lambda \sqrt{s}). \tag{119}
$$

$\blacksquare$

**Lemma A.4** (Tight local bound)**.** *The order magnitude of the upper bounds in Theorem 4.9 is tight.*

*Proof.* It is sufficient to show that there exist an example such that its forward pass code and Jacobian convergences are $\mathcal{O}(\rho^t)$ and $\mathcal{O}(t\rho^t)$, respectively. The following example confirms this. Without loss of generality, let $\boldsymbol{z}^*$ be 1-sparse and non-negative, $\boldsymbol{D} = \boldsymbol{D}^*$ and $\boldsymbol{D}_j = \boldsymbol{0}$ for $j \neq i$. The loss function is $\frac{1}{2}\|\boldsymbol{D}_i^* \boldsymbol{z}_{(i)}^* - \boldsymbol{D}_i \boldsymbol{z}_{(i)}\|_2^2 + \lambda |\boldsymbol{z}_{(i)}|$. Given the support recovery after first iteration, the encoder forward pass implements $\boldsymbol{z}_{t+1,(i)} = \boldsymbol{z}_{t,(i)} - \alpha(\boldsymbol{D}_i^{\mathrm{T}}(\boldsymbol{D}_i \boldsymbol{z}_{t,(i)} - \boldsymbol{D}_i^* \boldsymbol{z}_{(i)}^*) + \lambda) = (1-\alpha)\boldsymbol{z}_{t,(i)} + \alpha(\boldsymbol{z}_{(i)}^* - \lambda)$. Hence, the forward pass convergences are

$$
\boldsymbol{z}_{t,(i)} = (1-\alpha)^t \boldsymbol{z}_0 + \sum_{k=1}^{t} \alpha(1-\alpha)^{t-k}(\boldsymbol{z}_{(i)}^* - \lambda) = (1-\alpha)^t \boldsymbol{z}_0 + (1 - (1-\alpha)^t)(\boldsymbol{z}_{(i)}^* - \lambda) \tag{120}
$$
$$
\boldsymbol{z}_{t,(i)} - \hat{\boldsymbol{z}}_{(i)} = \rho^t(\boldsymbol{z}_0 - \boldsymbol{z}_{(i)}^* + \lambda) = \mathcal{O}(\rho^t)
$$

and

$$
\boldsymbol{J}_{t,(i)} = \boldsymbol{J}_{t-1,(i)} - \alpha(\boldsymbol{J}_{t-1,(i)} + 2\boldsymbol{D}_i \boldsymbol{z}_{t,(i)} - \boldsymbol{D}_i^* \boldsymbol{z}_{(i)}^*) = \rho \boldsymbol{J}_{t-1,(i)} + \mathcal{O}(\rho^t) + \hat{\boldsymbol{J}}_{(i)} \tag{121}
$$
$$
\boldsymbol{J}_{t,(i)} - \hat{\boldsymbol{J}}_{(i)} = \rho^t \boldsymbol{J}_{0,(i)} + \sum_{k=1}^{t} \mathcal{O}(\rho^t) = \mathcal{O}(t\rho^t)
$$

where $\rho = 1 - \alpha$, $\hat{\boldsymbol{z}}_{(i)} = \boldsymbol{z}_{(i)}^* - \lambda$, and $\hat{\boldsymbol{J}}_{(i)} = \alpha(2\boldsymbol{D}_i \hat{\boldsymbol{z}}_{(i)} - \boldsymbol{D}_i^* \boldsymbol{z}_{(i)}^*)$ $\blacksquare$

**Lemma A.5** (Local bounds)**.** *From local gradient errors in Theorem 4.9, the following are satisfied*

$$
\begin{aligned}
\|Q_t^{21}(\hat{\boldsymbol{z}})\|_2 &\leq (L_{21}/2) \|\boldsymbol{z}_t - \hat{\boldsymbol{z}}\|_2^2, \qquad & \|Q_t^{lasso-11}(\hat{\boldsymbol{z}})\|_2 &\leq (L_{11}/2) \|\boldsymbol{z}_t - \hat{\boldsymbol{z}}\|_2^2 \\
\|Q(\hat{\boldsymbol{z}}, \boldsymbol{J}_t)\|_2 &\leq L_1 \|\boldsymbol{J}_t - \hat{\boldsymbol{J}}\|_2, & \|Q_t^{ls-11}(\hat{\boldsymbol{z}})\|_2 &\leq (L_{11}/2) \|\boldsymbol{z}_t - \hat{\boldsymbol{z}}\|_2^2 + \|\partial h(\hat{\boldsymbol{z}})\|_2.
\end{aligned} \tag{122}
$$

*Proof.* For $Q_t^{21}(\hat{\boldsymbol{z}})$, given convexity of $\nabla_1 \mathcal{L}_{\boldsymbol{x}}(\boldsymbol{z}, \boldsymbol{D})$ and its domain (Assumption 3.1) and Lemma 3.3, we achieve the quadratic upper bound. For $Q_t^{\text{lasso-11}}(\hat{\boldsymbol{z}})$, we add and subtract $\nabla_1 \mathcal{L}_{\boldsymbol{x}}(\hat{\boldsymbol{z}}, \boldsymbol{D})$, and then use quadratic

upper bound. At line four, given Lemma A.3, we use $\mathbf{0} \in \nabla_1 \mathcal{L}_{\boldsymbol{x}}(\hat{\boldsymbol{z}}, \boldsymbol{D}) + \partial h(\hat{\boldsymbol{z}})$ and assume that $\boldsymbol{z}_t$ recovers the sign entries of $\hat{\boldsymbol{z}}$.

$$
\begin{aligned}
\|Q_t^{\text{lasso-11}}\|_2 &= \|\nabla_1 \mathcal{L}_{\boldsymbol{x}}(\boldsymbol{z}_t, \boldsymbol{D}) + \partial h(\boldsymbol{z}_t) - \nabla_{11}^2 \mathcal{L}_{\boldsymbol{x}}(\hat{\boldsymbol{z}}, \boldsymbol{D})(\boldsymbol{z}_t - \hat{\boldsymbol{z}})\| \\
&= \|\nabla_1 \mathcal{L}_{\boldsymbol{x}}(\boldsymbol{z}_t, \boldsymbol{D}) - \nabla_1 \mathcal{L}_{\boldsymbol{x}}(\hat{\boldsymbol{z}}, \boldsymbol{D}) + \nabla_1 \mathcal{L}_{\boldsymbol{x}}(\hat{\boldsymbol{z}}, \boldsymbol{D}) + \partial h(\boldsymbol{z}_t) - \nabla_{11}^2 \mathcal{L}_{\boldsymbol{x}}(\hat{\boldsymbol{z}}, \boldsymbol{D})(\boldsymbol{z}_t - \hat{\boldsymbol{z}})\| \\
&\leq (L_{11}/2)\|\boldsymbol{z}_t - \hat{\boldsymbol{z}}\|_2^2 + \|\partial h(\boldsymbol{z}_t) + \nabla_1 \mathcal{L}_{\boldsymbol{x}}(\hat{\boldsymbol{z}}, \boldsymbol{D})\|_2 \\
&\leq (L_{11}/2)\|\boldsymbol{z}_t - \hat{\boldsymbol{z}}\|_2^2 + \|\partial h(\boldsymbol{z}_t) - \partial h(\hat{\boldsymbol{z}})\|_2 \leq (L_{11}/2)\|\boldsymbol{z}_t - \hat{\boldsymbol{z}}\|_2^2.
\end{aligned}
\tag{123}
$$

Similarly,

$$
\begin{aligned}
\|Q_t^{\text{ls-11}}\|_2 &= \|\nabla_1 \mathcal{L}_{\boldsymbol{x}}(\boldsymbol{z}_t, \boldsymbol{D}) - \nabla_{11}^2 \mathcal{L}_{\boldsymbol{x}}(\hat{\boldsymbol{z}}, \boldsymbol{D})(\boldsymbol{z}_t - \hat{\boldsymbol{z}})\| \\
&= \|\nabla_1 \mathcal{L}_{\boldsymbol{x}}(\boldsymbol{z}_t, \boldsymbol{D}) - \nabla_1 \mathcal{L}_{\boldsymbol{x}}(\hat{\boldsymbol{z}}, \boldsymbol{D}) + \nabla_1 \mathcal{L}_{\boldsymbol{x}}(\hat{\boldsymbol{z}}, \boldsymbol{D}) - \nabla_{11}^2 \mathcal{L}_{\boldsymbol{x}}(\hat{\boldsymbol{z}}, \boldsymbol{D})(\boldsymbol{z}_t - \hat{\boldsymbol{z}})\| \\
&\leq (L_{11}/2)\|\boldsymbol{z}_t - \hat{\boldsymbol{z}}\|_2^2 + \|\nabla_1 \mathcal{L}_{\boldsymbol{x}}(\hat{\boldsymbol{z}}, \boldsymbol{D})\|_2 \leq (L_{11}/2)\|\boldsymbol{z}_t - \hat{\boldsymbol{z}}\|_2^2 + \|\partial h(\hat{\boldsymbol{z}})\|_2.
\end{aligned}
\tag{124}
$$

For $Q(\hat{\boldsymbol{z}}, \boldsymbol{J}_t)$, from implicit function theorem, $Q(\hat{\boldsymbol{z}}, \hat{\boldsymbol{J}}) = \mathbf{0}$ under the support $S$ of $\hat{\boldsymbol{z}}$ that is identified by $\boldsymbol{z}_t$. To prove this, consider the minimizer $\hat{\boldsymbol{z}}(\boldsymbol{D})$. We have $\mathbf{0} \in \nabla_1 f(\hat{\boldsymbol{z}}, \boldsymbol{D})$, hence, we get $\mathbf{0} \in \hat{\boldsymbol{J}}(\boldsymbol{D})\nabla_{11}^2 f(\hat{\boldsymbol{z}}, \boldsymbol{D}) + \nabla_{21}^2 f(\hat{\boldsymbol{z}}, \boldsymbol{D})$. Given the support recovery, the relation $\hat{\boldsymbol{J}}(\boldsymbol{D})(\nabla_{11}^2 \mathcal{L}_{\boldsymbol{x}}(\hat{\boldsymbol{z}}, \boldsymbol{D}) \odot \mathbf{1}_{S^*}) + \nabla_{21}^2 \mathcal{L}_{\boldsymbol{x}}(\hat{\boldsymbol{z}}, \boldsymbol{D}) \odot \mathbf{1}_{S^*} = \mathbf{0}$ also holds which is equivalent to $Q(\hat{\boldsymbol{z}}, \hat{\boldsymbol{J}})$ under the support. To show this, given the recursion $\boldsymbol{z}_{t+1} = \Phi(\boldsymbol{z}_t, \boldsymbol{D})$, we differentiate it and get $\boldsymbol{J}_{t+1} = \nabla_1 \Phi(\boldsymbol{z}_t, \boldsymbol{D})\boldsymbol{J}_t + \nabla_2 \Phi(\boldsymbol{z}_t, \boldsymbol{D})$. Given the support recovery and fixed-point property, we can write

$$
\begin{aligned}
\hat{\boldsymbol{J}} &= \mathbf{1}_S \odot (\hat{\boldsymbol{J}} - \alpha \nabla_{11}^2 \mathcal{L}_{\boldsymbol{x}}(\hat{\boldsymbol{z}}, \boldsymbol{D})^{\mathrm{T}} \hat{\boldsymbol{J}}) + \mathbf{1}_S \odot (-\alpha \nabla_{21}^2 \mathcal{L}_{\boldsymbol{x}}(\hat{\boldsymbol{z}}, \boldsymbol{D})^{\mathrm{T}}) \\
\hat{\boldsymbol{J}} - \mathbf{1}_S \odot \hat{\boldsymbol{J}} &= -\hat{\boldsymbol{J}} \alpha \nabla_{11}^2 \mathcal{L}_{\boldsymbol{x}}(\hat{\boldsymbol{z}}, \boldsymbol{D})^{\mathrm{T}} \odot \mathbf{1}_S - \alpha \nabla_{21}^2 \mathcal{L}_{\boldsymbol{x}}(\hat{\boldsymbol{z}}, \boldsymbol{D})^{\mathrm{T}} \odot \mathbf{1}_S \\
\mathbf{0} &= \hat{\boldsymbol{J}}^+ (\nabla_{11}^2 \mathcal{L}(\hat{\boldsymbol{z}}, \boldsymbol{D}) \odot \mathbf{1}_S) + \nabla_{21}^2 \mathcal{L}_{\boldsymbol{x}}(\hat{\boldsymbol{z}}, \boldsymbol{D}) \odot \mathbf{1}_S
\end{aligned}
\tag{125}
$$

If the term $(\nabla_{11}^2 \mathcal{L}(\boldsymbol{z}_t, \boldsymbol{D}) \odot \mathbf{1}_S)$ is invertible, then we can write

$$
\hat{\boldsymbol{J}}^+ = -\nabla_{21}^2 \mathcal{L}_{\boldsymbol{x}}(\hat{\boldsymbol{z}}, \boldsymbol{D}) \odot \mathbf{1}_S (\nabla_{11}^2 \mathcal{L}(\hat{\boldsymbol{z}}, \boldsymbol{D}) \odot \mathbf{1}_S)^{-1}
\tag{126}
$$

For the Jacobian corresponding to row $i$ of the dictionary, we get

$$
\hat{\boldsymbol{J}}_{(i,:)} = -(\boldsymbol{D}_S^{\mathrm{T}} \boldsymbol{D}_S)^{-1}(\boldsymbol{D}_{i,:}\hat{\boldsymbol{z}}^{\mathrm{T}} + (\boldsymbol{D}_{i,:}^{\mathrm{T}}\hat{\boldsymbol{z}} - \boldsymbol{x}_i)\boldsymbol{I}_p)_S
\tag{127}
$$

on the support. Outside of the support $S$, the Jacobian is zero. This proof is similarly provided by Malézieux et al. (2022). Hence, we can use $\nabla_{21}^2 \mathcal{L}_{\boldsymbol{x}}(\hat{\boldsymbol{z}}, \boldsymbol{D}) = -\hat{\boldsymbol{J}}^+ \nabla_{11}^2 \mathcal{L}_{\boldsymbol{x}}(\hat{\boldsymbol{z}}, \boldsymbol{D})$ under the support $S$ in the following

$$
\begin{aligned}
\|Q(\hat{\boldsymbol{z}}, \boldsymbol{J}_t)\|_2 &= \|\boldsymbol{J}_t^+ \nabla_{11}^2 \mathcal{L}_{\boldsymbol{x}}(\hat{\boldsymbol{z}}, \boldsymbol{D}) + \nabla_{21}^2 \mathcal{L}_{\boldsymbol{x}}(\hat{\boldsymbol{z}}, \boldsymbol{D})\|_2 = \|\boldsymbol{J}_t^+ \nabla_{11}^2 \mathcal{L}_{\boldsymbol{x}}(\hat{\boldsymbol{z}}, \boldsymbol{D}) - \hat{\boldsymbol{J}}^+ \nabla_{11}^2 \mathcal{L}_{\boldsymbol{x}}(\hat{\boldsymbol{z}}, \boldsymbol{D})\|_2 \\
&\leq \|(\boldsymbol{J}_t - \hat{\boldsymbol{J}})^+ \nabla_{11}^2 \mathcal{L}_{\boldsymbol{x}}(\hat{\boldsymbol{z}}, \boldsymbol{D})\|_2 \leq L_1 \|\boldsymbol{J}_t - \hat{\boldsymbol{J}}\|_2.
\end{aligned}
\tag{128}
$$

∎

## A.5 Global backward pass proof details

We re-state and proof Theorem 4.10 as follows:

**Theorem 4.10** (Global error of gradients). *Given the convergence results from the forward pass, (Theorems 4.6 and 4.8), the errors of gradients defined in Algorithm 2 w.r.t global direction $\boldsymbol{g}^*$ (defined in (5)) satisfy*

$$
\begin{aligned}
\|\boldsymbol{g}_\infty^{ae\text{-}lasso} - \boldsymbol{g}^*\|_2 &\leq \mathcal{O}(\|\boldsymbol{D} - \boldsymbol{D}^*\|_2^2 + \|\boldsymbol{D} - \boldsymbol{D}^*\|_2 + \hat{\delta}^* + \hat{\delta}_J^* + M_J \lambda \sqrt{s}) \\
\|\boldsymbol{g}_\infty^{ae\text{-}ls} - \boldsymbol{g}^*\|_2 &\leq \mathcal{O}(\|\boldsymbol{D} - \boldsymbol{D}^*\|_2^2 + \|\boldsymbol{D} - \boldsymbol{D}^*\|_2 + \hat{\delta}^* + \hat{\delta}_J^*).
\end{aligned}
\tag{13}
$$

*Proof.* For $\boldsymbol{g}_t^{\text{dec}}$, we compute the gradient in their limit assuming infinite fresh samples $\lim_{n \to \infty} \frac{1}{n} \sum_{i=1}^n \nabla_2 \mathcal{L}_{\boldsymbol{x}^i}(\boldsymbol{z}_t^i, \boldsymbol{D}) = \mathbb{E}_{\boldsymbol{x} \in \mathcal{X}} [\nabla_2 \mathcal{L}_{\boldsymbol{x}}(\boldsymbol{z}_t, \boldsymbol{D})]$ a.s.. Similar to Theorem 4.9, we re-write the errors of gradients $\boldsymbol{g}_t^{\text{ae-lasso}}$ and $\boldsymbol{g}_t^{\text{ae-ls}}$ as following

$$
\begin{aligned}
\boldsymbol{g}_t^{\text{ae-lasso}} - \boldsymbol{g}^* &= \mathbb{E}_{\boldsymbol{x} \in \mathcal{X}} [Q(\boldsymbol{z}^*, \boldsymbol{J}_t)(\boldsymbol{z}_t - \boldsymbol{z}^*)] + \mathbb{E}_{\boldsymbol{x} \in \mathcal{X}} [Q_t^{21}(\boldsymbol{z}^*)] + \mathbb{E}_{\boldsymbol{x} \in \mathcal{X}} [\boldsymbol{J}_t Q_t^{\text{lasso-11}}(\boldsymbol{z}^*)] \\
\boldsymbol{g}_t^{\text{ae-ls}} - \boldsymbol{g}^* &= \mathbb{E}_{\boldsymbol{x} \in \mathcal{X}} [Q(\boldsymbol{z}^*, \boldsymbol{J}_t)(\boldsymbol{z}_t - \boldsymbol{z}^*)] + \mathbb{E}_{\boldsymbol{x} \in \mathcal{X}} [Q_t^{21}(\boldsymbol{z}^*)] + \mathbb{E}_{\boldsymbol{x} \in \mathcal{X}} [\boldsymbol{J}_t Q_t^{\text{ls-11}}(\boldsymbol{z}^*)].
\end{aligned}
\tag{129}
$$

where $Q_t^{21}(\boldsymbol{z})$, $Q_t^{\text{lasso-11}}(\boldsymbol{z})$, $Q_t^{\text{ls-11}}(\boldsymbol{z})$, and $Q(\boldsymbol{z}, \boldsymbol{J})$ are defined as in Theorem 4.9. Given Assumption 4.1 and Lemma A.6, we find an upper bound on the *r.h.s* of the gradient errors as follows:

$$
\begin{aligned}
\|\boldsymbol{g}_t^{\text{ae-lasso}} - \boldsymbol{g}^*\|_2 \leq\ & \mathbb{E}_{\boldsymbol{x} \in \mathcal{X}}\ [(L_1\|\boldsymbol{J}_t - \boldsymbol{J}^*\|_2 + M_J L_{11D} L_{21D}\|\boldsymbol{D} - \boldsymbol{D}^*\|_2)\|\boldsymbol{z}_t - \boldsymbol{z}^*\|_2 + (L_{21}/2)\|\boldsymbol{z}_t - \boldsymbol{z}^*\|_2^2] \\
& + \mathbb{E}_{\boldsymbol{x} \in \mathcal{X}}\ [M_J(L_{11}/2)\|\boldsymbol{z}_t - \boldsymbol{z}^*\|_2^2 + M_J\|\partial h(\boldsymbol{z}_t)\|_2 + L_{1D}\|\boldsymbol{D} - \boldsymbol{D}^*\|_2] \\
\leq\ & \mathbb{E}_{\boldsymbol{x} \in \mathcal{X}}\ [L_1(\|\boldsymbol{J}_t - \hat{\boldsymbol{J}}\|_2 + \|\hat{\boldsymbol{J}} - \boldsymbol{J}^*\|_2 + M_J L_{11D} L_{21D}\|\boldsymbol{D} - \boldsymbol{D}^*\|_2)(\|\boldsymbol{z}_t - \hat{\boldsymbol{z}}\|_2 + \|\hat{\boldsymbol{z}} - \boldsymbol{z}^*\|_2)] \\
& + \mathbb{E}_{\boldsymbol{x} \in \mathcal{X}}\ [(L_{21}/2)(\|\boldsymbol{z}_t - \hat{\boldsymbol{z}}\|_2^2 + \|\hat{\boldsymbol{z}} - \boldsymbol{z}^*\|_2^2) + L_{1D}\|\boldsymbol{D} - \boldsymbol{D}^*\|_2] \\
& + \mathbb{E}_{\boldsymbol{x} \in \mathcal{X}}\ [M_J(L_{11}/2)(\|\boldsymbol{z}_t - \hat{\boldsymbol{z}}\|_2^2 + \|\hat{\boldsymbol{z}} - \boldsymbol{z}^*\|_2^2) + M_J\|\partial h(\boldsymbol{z}_t)\|_2]
\end{aligned}
\tag{130}
$$

Similarly,

$$
\begin{aligned}
\|\boldsymbol{g}_t^{\text{ae-ls}} - \boldsymbol{g}^*\|_2 \leq\ & \mathbb{E}_{\boldsymbol{x} \in \mathcal{X}}\ [(L_1\|\boldsymbol{J}_t - \boldsymbol{J}^*\|_2 + M_J L_{11D} L_{21D}\|\boldsymbol{D} - \boldsymbol{D}^*\|_2)\|\boldsymbol{z}_t - \boldsymbol{z}^*\|_2 + (L_{21}/2)\|\boldsymbol{z}_t - \boldsymbol{z}^*\|_2^2] \\
& + \mathbb{E}_{\boldsymbol{x} \in \mathcal{X}}\ [M_J(L_{11}/2)\|\boldsymbol{z}_t - \boldsymbol{z}^*\|_2^2 + L_{1D}\|\boldsymbol{D} - \boldsymbol{D}^*\|_2] \\
\leq\ & \mathbb{E}_{\boldsymbol{x} \in \mathcal{X}}\ [L_1(\|\boldsymbol{J}_t - \hat{\boldsymbol{J}}\|_2 + \|\hat{\boldsymbol{J}} - \boldsymbol{J}^*\|_2 + M_J L_{11D} L_{21D}\|\boldsymbol{D} - \boldsymbol{D}^*\|_2)(\|\boldsymbol{z}_t - \hat{\boldsymbol{z}}\|_2 + \|\hat{\boldsymbol{z}} - \boldsymbol{z}^*\|_2)] \\
& + \mathbb{E}_{\boldsymbol{x} \in \mathcal{X}}\ [(L_{21}/2)(\|\boldsymbol{z}_t - \hat{\boldsymbol{z}}\|_2^2 + \|\hat{\boldsymbol{z}} - \boldsymbol{z}^*\|_2^2)] \\
& + \mathbb{E}_{\boldsymbol{x} \in \mathcal{X}}\ [M_J(L_{11}/2)(\|\boldsymbol{z}_t - \hat{\boldsymbol{z}}\|_2^2 + \|\hat{\boldsymbol{z}} - \boldsymbol{z}^*\|_2^2) + L_{1D}\|\boldsymbol{D} - \boldsymbol{D}^*\|_2].
\end{aligned}
\tag{131}
$$

Using the convergence errors from the forward pass (Theorems 4.3 and 4.7),

$$
\begin{aligned}
\|\boldsymbol{g}_t^{\text{ae-lasso}} - \boldsymbol{g}^*\|_2 \leq\ & L_1 \mathcal{O}(t\rho^{2t} + (\|\boldsymbol{D} - \boldsymbol{D}^*\|_2 + \hat{\delta}^*)t\rho^t + \rho^t(\|\boldsymbol{D} - \boldsymbol{D}^*\|_2 + \hat{\delta}_J^*) \\
& + L_1 \mathcal{O}(\|\boldsymbol{D} - \boldsymbol{D}^*\|_2 + \hat{\delta}^*)(\|\boldsymbol{D} - \boldsymbol{D}^*\|_2 + \hat{\delta}_J^*)) \\
& + M_J L_{11D} L_{21D}\|\boldsymbol{D} - \boldsymbol{D}^*\|_2(\rho^t + (\|\boldsymbol{D} - \boldsymbol{D}^*\|_2 + \hat{\delta}^*)) \\
& + (L_{21}/2 + M_J L_{11}/2)\,\mathcal{O}(\rho^t + \|\boldsymbol{D} - \boldsymbol{D}^*\|_2 + \hat{\delta}^*) + \mathcal{O}(\|\boldsymbol{D} - \boldsymbol{D}^*\|_2) + M_J\|\partial h(\boldsymbol{z}_t)\|_2
\end{aligned}
\tag{132}
$$

Hence,

$$
\begin{aligned}
\|\boldsymbol{g}_\infty^{\text{ae-lasso}} - \boldsymbol{g}^*\|_2 &\leq \mathcal{O}((\|\boldsymbol{D} - \boldsymbol{D}^*\|_2 + \hat{\delta}_J^*)(\|\boldsymbol{D} - \boldsymbol{D}^*\|_2 + \hat{\delta}^* + 1) + M_J \lambda \sqrt{s}) \\
&= \mathcal{O}(\|\boldsymbol{D} - \boldsymbol{D}^*\|_2^2 + \|\boldsymbol{D} - \boldsymbol{D}^*\|_2 + \|\boldsymbol{D} - \boldsymbol{D}^*\|_2(\hat{\delta}^* + \hat{\delta}_J^*) + \hat{\delta}^* + \hat{\delta}_J^* + M_J \lambda \sqrt{s})
\end{aligned}
\tag{133}
$$

Similarly,

$$
\|\boldsymbol{g}_\infty^{\text{ae-ls}} - \boldsymbol{g}^*\|_2 \leq \mathcal{O}(\|\boldsymbol{D} - \boldsymbol{D}^*\|_2^2 + \|\boldsymbol{D} - \boldsymbol{D}^*\|_2 + \|\boldsymbol{D} - \boldsymbol{D}^*\|_2(\hat{\delta}^* + \hat{\delta}_J^*) + \hat{\delta}^* + \hat{\delta}_J^*)
\tag{134}
$$

∎

**Lemma A.6** (Global bounds). *From global gradient errors in Theorem 4.10, the following are satisfied*

$$
\begin{aligned}
\|Q_t^{21}(\boldsymbol{z}^*)\|_2 &\leq (L_{21}/2)\|\boldsymbol{z}_t - \boldsymbol{z}^*\|_2^2 \\
\|Q_t^{lasso\text{-}11}(\boldsymbol{z}^*)\|_2 &\leq (L_{11}/2)\|\boldsymbol{z}_t - \boldsymbol{z}^*\|_2^2 + L_{1D}\|\boldsymbol{D} - \boldsymbol{D}^*\|_2 + \|\partial h(\boldsymbol{z}_t)\|_2 \\
\|Q_t^{ls\text{-}11}(\boldsymbol{z}^*)\|_2 &\leq (L_{11}/2)\|\boldsymbol{z}_t - \boldsymbol{z}^*\|_2^2 + L_{1D}\|\boldsymbol{D} - \boldsymbol{D}^*\|_2 \\
\|Q(\boldsymbol{z}^*, \boldsymbol{J}_t)\|_2 &\leq L_1\|\boldsymbol{J}_t - \boldsymbol{J}^*\|_2 + M_J L_{11D} L_{21D}\|\boldsymbol{D} - \boldsymbol{D}^*\|_2.
\end{aligned}
\tag{135}
$$

*Proof.* For $Q_t^{21}(\boldsymbol{z}^*)$, we achieve the quadratic bound using convexity of $\nabla_1 \mathcal{L}_{\boldsymbol{x}}(\boldsymbol{z}, \boldsymbol{D})$ and its domain (Assumption 3.1) and Lemma 3.3. For $Q_t^{\text{lasso-11}}(\boldsymbol{z}^*)$, we add and subtract $\nabla_1 \mathcal{L}_{\boldsymbol{x}}(\boldsymbol{z}^*, \boldsymbol{D})$, and use quadratic upper bound similar to Lemma A.5. At line four, we use $\boldsymbol{0} \in \nabla_1 \mathcal{L}_{\boldsymbol{x}}(\boldsymbol{z}^*, \boldsymbol{D}^*)$ (Lemma A.3) and assume that $\boldsymbol{z}_t$

recovers the sign entries of $\boldsymbol{z}^*$ (see Theorem 4.1 and Theorem 4.2).

$$
\begin{aligned}
\|Q_t^{\text{lasso-11}}(\boldsymbol{z}^*)\|_2 &= \|\nabla_1 \mathcal{L}_{\boldsymbol{x}}(\boldsymbol{z}_t, \boldsymbol{D}) + \partial h(\boldsymbol{z}_t) - \nabla_{11}^2 \mathcal{L}_{\boldsymbol{x}}(\boldsymbol{z}^*, \boldsymbol{D})(\boldsymbol{z}_t - \boldsymbol{z}^*)\|_2 \\
&= \|\nabla_1 \mathcal{L}_{\boldsymbol{x}}(\boldsymbol{z}_t, \boldsymbol{D}) - \nabla_1 \mathcal{L}_{\boldsymbol{x}}(\boldsymbol{z}^*, \boldsymbol{D}) + \nabla_1 \mathcal{L}_{\boldsymbol{x}}(\boldsymbol{z}^*, \boldsymbol{D}) + \partial h(\boldsymbol{z}_t) - \nabla_{11}^2 \mathcal{L}_{\boldsymbol{x}}(\boldsymbol{z}^*, \boldsymbol{D})(\boldsymbol{z}_t - \boldsymbol{z}^*)\|_2 \\
&\le (L_{11}/2)\|\boldsymbol{z}_t - \boldsymbol{z}^*\|_2^2 + \|\partial h(\boldsymbol{z}_t) + \nabla_1 \mathcal{L}_{\boldsymbol{x}}(\boldsymbol{z}^*, \boldsymbol{D})\|_2 \\
&\le (L_{11}/2)\|\boldsymbol{z}_t - \boldsymbol{z}^*\|_2^2 + \|\partial h(\boldsymbol{z}_t) + \nabla_1 \mathcal{L}_{\boldsymbol{x}}(\boldsymbol{z}^*, \boldsymbol{D}) - \nabla_1 \mathcal{L}_{\boldsymbol{x}}(\boldsymbol{z}^*, \boldsymbol{D}^*)\|_2 \\
&\le (L_{11}/2)\|\boldsymbol{z}_t - \boldsymbol{z}^*\|_2^2 + L_{1D}\|\boldsymbol{D} - \boldsymbol{D}^*\|_2 + \|\partial h(\boldsymbol{z}_t)\|_2.
\end{aligned}
\tag{136}
$$

Similarly,

$$
\begin{aligned}
\|Q_t^{\text{ls-11}}(\boldsymbol{z}^*)\|_2 &= \|\nabla_1 \mathcal{L}_{\boldsymbol{x}}(\boldsymbol{z}_t, \boldsymbol{D}) - \nabla_{11}^2 \mathcal{L}_{\boldsymbol{x}}(\boldsymbol{z}^*, \boldsymbol{D})(\boldsymbol{z}_t - \boldsymbol{z}^*)\|_2 \\
&= \|\nabla_1 \mathcal{L}_{\boldsymbol{x}}(\boldsymbol{z}_t, \boldsymbol{D}) - \nabla_1 \mathcal{L}_{\boldsymbol{x}}(\boldsymbol{z}^*, \boldsymbol{D}) + \nabla_1 \mathcal{L}_{\boldsymbol{x}}(\boldsymbol{z}^*, \boldsymbol{D}) - \nabla_{11}^2 \mathcal{L}_{\boldsymbol{x}}(\boldsymbol{z}^*, \boldsymbol{D})(\boldsymbol{z}_t - \boldsymbol{z}^*)\|_2 \\
&\le (L_{11}/2)\|\boldsymbol{z}_t - \boldsymbol{z}^*\|_2^2 + \|\nabla_1 \mathcal{L}_{\boldsymbol{x}}(\boldsymbol{z}^*, \boldsymbol{D})\|_2 \le (L_{11}/2)\|\boldsymbol{z}_t - \boldsymbol{z}^*\|_2^2 + \|\nabla_1 \mathcal{L}_{\boldsymbol{x}}(\boldsymbol{z}^*, \boldsymbol{D}) - \nabla_1 \mathcal{L}_{\boldsymbol{x}}(\boldsymbol{z}^*, \boldsymbol{D}^*)\|_2 \\
&\le (L_{11}/2)\|\boldsymbol{z}_t - \boldsymbol{z}^*\|_2^2 + L_{1D}\|\boldsymbol{D} - \boldsymbol{D}^*\|_2.
\end{aligned}
\tag{137}
$$

For $Q(\boldsymbol{z}^*, \boldsymbol{J}_t)$, from implicit function theorem, $Q(\boldsymbol{z}^*, \boldsymbol{J}^*) = 0$ for $\boldsymbol{D}$ evaluated at $\boldsymbol{D}^*$. Hence, we can use $\nabla_{21}^2 \mathcal{L}_{\boldsymbol{x}}(\boldsymbol{z}^*, \boldsymbol{D}^*) = -\boldsymbol{J}^{*+}\nabla_{11}^2 \mathcal{L}_{\boldsymbol{x}}(\boldsymbol{z}^*, \boldsymbol{D}^*)$ in the following

$$
\begin{aligned}
\|Q(\boldsymbol{z}^*, \boldsymbol{J}_t)\|_2 &= \|\boldsymbol{J}_t^+ \nabla_{11}^2 \mathcal{L}_{\boldsymbol{x}}(\boldsymbol{z}^*, \boldsymbol{D}) + \nabla_{21}^2 \mathcal{L}_{\boldsymbol{x}}(\boldsymbol{z}^*, \boldsymbol{D}) - \nabla_{21}^2 \mathcal{L}_{\boldsymbol{x}}(\boldsymbol{z}^*, \boldsymbol{D}^*) + \nabla_{21}^2 \mathcal{L}_{\boldsymbol{x}}(\boldsymbol{z}^*, \boldsymbol{D}^*)\|_2 \\
&= \|\boldsymbol{J}_t^+ \nabla_{11}^2 \mathcal{L}_{\boldsymbol{x}}(\boldsymbol{z}^*, \boldsymbol{D}) - \boldsymbol{J}^{*+} \nabla_{11}^2 \mathcal{L}_{\boldsymbol{x}}(\boldsymbol{z}^*, \boldsymbol{D}^*)\|_2 + \|\nabla_{21}^2 \mathcal{L}_{\boldsymbol{x}}(\boldsymbol{z}^*, \boldsymbol{D}) - \nabla_{21}^2 \mathcal{L}_{\boldsymbol{x}}(\boldsymbol{z}^*, \boldsymbol{D}^*)\|_2 \\
&= \|\boldsymbol{J}_t^+ \nabla_{11}^2 \mathcal{L}_{\boldsymbol{x}}(\boldsymbol{z}^*, \boldsymbol{D}) - \boldsymbol{J}_t^+ \nabla_{11}^2 \mathcal{L}_{\boldsymbol{x}}(\boldsymbol{z}^*, \boldsymbol{D}^*)\|_2 \\
&\quad + \|\boldsymbol{J}_t^+ \nabla_{11}^2 \mathcal{L}_{\boldsymbol{x}}(\boldsymbol{z}^*, \boldsymbol{D}^*) - \boldsymbol{J}^{*+} \nabla_{11}^2 \mathcal{L}_{\boldsymbol{x}}(\boldsymbol{z}^*, \boldsymbol{D}^*)\|_2 + L_{21D}\|\boldsymbol{D} - \boldsymbol{D}^*\|_2 \\
&= M_J L_{11D}\|\boldsymbol{D} - \boldsymbol{D}^*\|_2 + \|\boldsymbol{J}_t^+ \nabla_{11}^2 \mathcal{L}_{\boldsymbol{x}}(\boldsymbol{z}^*, \boldsymbol{D}^*) - \boldsymbol{J}^{*+} \nabla_{11}^2 \mathcal{L}_{\boldsymbol{x}}(\boldsymbol{z}^*, \boldsymbol{D}^*)\|_2 + L_{21D}\|\boldsymbol{D} - \boldsymbol{D}^*\|_2 \\
&\le \|(\boldsymbol{J}_t^+ - \boldsymbol{J}^{*+})\nabla_{11}^2 \mathcal{L}_{\boldsymbol{x}}(\boldsymbol{z}^*, \boldsymbol{D})\|_2 + M_J L_{11D} L_{21D}\|\boldsymbol{D} - \boldsymbol{D}^*\|_2 \\
&\le L_1 \|\boldsymbol{J}_t - \boldsymbol{J}^*\|_2 + M_J L_{11D} L_{21D}\|\boldsymbol{D} - \boldsymbol{D}^*\|_2.
\end{aligned}
\tag{138}
$$

∎

**Theorem 4.11** (Dictionary learning with variable $\lambda_t$). *Given Assumptions 3.3 and 3.4, suppose $\boldsymbol{D}^{(l)}$ is $\mu_l$-incoherent and $(\delta_l, 2)$-close to $\boldsymbol{D}^*$ with $\delta_l = \mathcal{O}^*(1/\log p)$. If $s = \mathcal{O}(\sqrt{m})$, $\mu = \mathcal{O}(\log m)$, learning rate is $\eta = \mathcal{O}(\frac{p}{s(1-\delta_l^2/2)})$, and the regularizer and step size are set according to Theorem 4.4, then for any dictionary update $l$ using $\boldsymbol{g}_T^{dec}$, with probability of at least $1 - \epsilon_{supp\text{-}pres}^{(l)}$,*

$$
\|\boldsymbol{D}_j^{(l+1)} - \boldsymbol{D}_j^*\|_2^2 \le (1 - \psi)\|\boldsymbol{D}_j^{(l)} - \boldsymbol{D}_j^*\|_2^2
\tag{15}
$$

*where $\epsilon_{supp\text{-}pres}^{(l)} := \epsilon_{supp\text{-}rec}^{(l)} + \epsilon_\gamma^{(l)} = 2p \exp\left(\frac{-C_{\min}^2}{\mathcal{O}^*(\delta_l^2)}\right) + 2s \exp\left(\frac{-1}{\mathcal{O}(\delta_l)}\right)$.*

*Proof.* In this proof, we study $g_{T,j}^{\text{dec}}$, and for ease of notation we drop the superscript.

$$
\boldsymbol{g}_{T,j}^{(l)} = \mathbb{E}[\mathbf{1}_{\boldsymbol{z}_{T,(j)} \ne 0} \boldsymbol{z}_{T,(j)}(\boldsymbol{D}^{(l)} \boldsymbol{z}_T - \boldsymbol{x})] = \mathbb{E}[\mathbf{1}_{\boldsymbol{z}_{(j)}^* \ne 0} \boldsymbol{z}_{T,(j)}(\boldsymbol{D}^{(l)} \boldsymbol{z}_T - \boldsymbol{x})] + \gamma
\tag{139}
$$

where $\gamma = \mathbb{E}[(\mathbf{1}_{\boldsymbol{z}_{T,(j)} \ne 0} - \mathbf{1}_{\boldsymbol{z}_{(j)}^* \ne 0}) \boldsymbol{z}_{T,(j)}(\boldsymbol{D}^{(l)} \boldsymbol{z}_T - \boldsymbol{x})]$. We have the event $\mathbf{1}_{\boldsymbol{z}_{T,(j)} \ne 0} - \mathbf{1}_{\boldsymbol{z}_{(j)}^* \ne 0} = 0$ happening with probability of $1 - \epsilon_{\text{supp-pres}}^{(l)}$, and $\epsilon_{\text{supp-pres}}^{(l)}$ decreases with decrease in $\delta_l$. Hence, $\gamma$ gets smaller. We write

$$
\boldsymbol{g}_{T,j}^{(l)} = \mathbb{E}[\mathbf{1}_{\boldsymbol{z}_{(j)}^* \ne 0} \boldsymbol{z}_{T,(j)}(\boldsymbol{D}^{(l)} \boldsymbol{z}_T - \boldsymbol{x})] + \gamma
\tag{140}
$$

where $B_S^{(l)}$ is an diagonal matrix with $\beta_j^{(l)}$ for $j \in S$ as entries. For $j \notin S$, $\mathbf{1}_{\boldsymbol{z}_{(j)}^* \ne 0} = 0$, which results in $\boldsymbol{g}_{T,j}^{(l)} = 0$. Hence, we only focus on $j \in S$ where $\mathbf{1}_{\boldsymbol{z}_{(j)}^* \ne 0} = 1$. We condition on the support and re-write the

gradient as

$$
\begin{aligned}
\boldsymbol{g}_{T,j}^{(l)} &= \mathbb{E}[\boldsymbol{z}_{T,(j)}(\boldsymbol{D}^{(l)}\boldsymbol{z}_T - \boldsymbol{x})] + \gamma \\
&= \mathbb{E}[\mathbb{E}[\boldsymbol{z}_{T,(j)}[\boldsymbol{D}_S^{(l)}(\boldsymbol{I} - B_S^{(l)})\boldsymbol{z}_{(S)}^* + \boldsymbol{D}_S^{(l)}\zeta_{T,S}^{(l)} - \boldsymbol{D}_S^*\boldsymbol{z}_{(S)}^*] \mid S] + \gamma \\
&= \mathbb{E}[\boldsymbol{D}_S^{(l)}(\boldsymbol{I} - B_S^{(l)})\mathbb{E}[\boldsymbol{z}_{T,(j)}\boldsymbol{z}_{(S)}^* \mid S]] - \mathbb{E}[\boldsymbol{D}_S^* E[\boldsymbol{z}_{T,(j)}\boldsymbol{z}_{(S)}^* \mid S]] + \mathbb{E}[\boldsymbol{D}_S^{(l)}\mathbb{E}[\boldsymbol{z}_{T,(j)}\zeta_{T,S}^{(l)}] \mid S] + \gamma \\
&= \mathbb{E}[\boldsymbol{D}_S^{(l)}(\boldsymbol{I} - B_S^{(l)})\mathbb{E}[(\boldsymbol{z}_{(j)}^*(1 - \beta_j^{(l)}) + \zeta_{T,j}^{(l)})\boldsymbol{z}_{(S)}^* \mid S]] - \mathbb{E}[\boldsymbol{D}_S^* E[(\boldsymbol{z}_{(j)}^*(1 - \beta_j^{(l)}) + \zeta_{T,j}^{(l)})\boldsymbol{z}_{(S)}^* \mid S]] \\
&\quad + \mathbb{E}[\boldsymbol{D}_S^{(l)}\mathbb{E}[(\boldsymbol{z}_{(j)}^*(1 - \beta_j^{(l)}) + \zeta_{T,j}^{(l)})\zeta_{T,S}^{(l)}] \mid S] + \gamma \\
&= \mathbb{E}[\boldsymbol{D}_j^{(l)}(1 - \beta_j^{(l)})^2] - \mathbb{E}[\boldsymbol{D}_j^*(1 - \beta_j^{(l)})] + \gamma \\
&\quad + \mathbb{E}[\boldsymbol{D}_S^{(l)}(\boldsymbol{I} - B_S^{(l)})E[\boldsymbol{z}_{(S)}^*\zeta_{T,j}^{(l)} \mid S]] - \mathbb{E}[\boldsymbol{D}_S^* E[\boldsymbol{z}_{(S)}^*\zeta_{T,j}^{(l)} \mid S]] + \mathbb{E}[\boldsymbol{D}_S^{(l)}\mathbb{E}[(\boldsymbol{z}_{(j)}^*(1 - \beta_j^{(l)}) + \zeta_{T,j}^{(l)})\zeta_{T,S}^{(l)} \mid S]]
\end{aligned}
$$
(141)

where in the last line, we use the fact that $\mathbb{E}[\boldsymbol{z}_{(j)}^* \mid j \in S] = 0$ and $\mathbb{E}[\boldsymbol{z}_{(S)}^*\boldsymbol{z}_{(S)}^{*\mathrm{T}} \mid S] = \boldsymbol{I}$. Computing the expectation, we get

$$
\boldsymbol{g}_{T,j}^{(l)} = p_j\boldsymbol{D}_j^{(l)}(1 - \beta_j^{(l)})^2 - p_j\boldsymbol{D}_j^*(1 - \beta_j^{(l)}) + U_{T,j}^{(l)} + \gamma = p_j(1 - \beta_j^{(l)})\left((1 - \beta_j^{(l)})\boldsymbol{D}_j^{(l)} - \boldsymbol{D}_j^*\right) + U_{T,j}^{(l)} + \gamma
$$
(142)

where $U_{T,j}^{(l)} = \mathbb{E}[\boldsymbol{D}_S^{(l)}(\boldsymbol{I} - B_S^{(l)})E[\boldsymbol{z}_{(S)}^*\zeta_{T,j}^{(l)} \mid S]] - \mathbb{E}[\boldsymbol{D}_S^* E[\boldsymbol{z}_{(S)}^*\zeta_{T,j}^{(l)} \mid S]] + \mathbb{E}[\boldsymbol{D}_S^{(l)}\mathbb{E}[(\boldsymbol{z}_{(j)}^*(1 - \beta_j^{(l)}) + \zeta_{T,j}^{(l)})\zeta_{T,S}^{(l)} \mid S]]$. Given this gradient, we now find a bound on $U_{T,j}^{(l)}$.

$$
U_{T,j}^{(l)} = \mathbb{E}[\boldsymbol{D}_S^{(l)}(\boldsymbol{I} - B_S^{(l)})E[\boldsymbol{z}_{(S)}^*\zeta_{T,j}^{(l)} \mid S]] - \mathbb{E}[\boldsymbol{D}_S^*\mathbb{E}[\boldsymbol{z}_{(S)}^*\zeta_{T,j}^{(l)} \mid S]] + \mathbb{E}[\boldsymbol{D}_S^{(l)}\mathbb{E}[(\boldsymbol{z}_{(j)}^*(1 - \beta_j^{(l)}) + \zeta_{T,j}^{(l)})\zeta_{T,S}^{(l)} \mid S]]
$$
(143)

First, we bound $\mathbb{E}[\boldsymbol{z}_{(i)}^*\zeta_{T,j}^{(l)} \mid S]]$ as following.

$$
\begin{aligned}
\mathbb{E}[\boldsymbol{z}_{(i)}^*\zeta_{T,j}^{(l)} \mid S]] &\leq \sum_{t=1}^{T}\alpha(1 - \alpha)^{T-t}\mathbb{E}[2\gamma_j^{(l)}\boldsymbol{z}_{(i)}^* \mid S] + \frac{\mu_l}{\sqrt{m}}\sum_{t=1}^{T}\alpha(1 - \alpha)^{T-t}\sum_{k \neq j}\mathbb{E}[E_{t-1,k}\mathrm{sign}(\boldsymbol{z}_{(k)}^* - \boldsymbol{z}_{t-1,(k)})\boldsymbol{z}_{(i)}^* \mid S] \\
&\quad + \frac{\mu_l}{\sqrt{m}}\sum_{t=1}^{T}\alpha(1 - \alpha)^{T-t}\sum_{k \neq j}\mathbb{E}[E_{t-1,k}\mathrm{sign}(\boldsymbol{z}_{(k)}^* - \boldsymbol{z}_{t-1,(k)})\mathrm{sign}(\boldsymbol{z}_{t,(i)})\boldsymbol{z}_{(i)}^* \mid S] + \tilde{\kappa}_{T,j}^{(l)}
\end{aligned}
$$
(144)

where $\tilde{\kappa}_{T,j}^{(l)} = \mathbb{E}[\boldsymbol{z}_{(i)}^*\kappa_{T,j}^{(l)}]$. Similar to $\kappa_{T,j}^{(l)}$, $\tilde{\kappa}_{T,j}^{(l)}$ decay very fast as $T$ increases. Hence, we bound the other terms. We have

$$
\mathbb{E}[\gamma_j^{(l)}\boldsymbol{z}_{(i)}^* \mid S] \begin{cases} \leq \delta_l & \text{if } j \neq i \\ = 0 & \text{if } j = i \end{cases},
$$
(145)

$$
\mathbb{E}[E_{t-1,k}\mathrm{sign}(\boldsymbol{z}_{(k)}^* - \boldsymbol{z}_{t-1,(k)})\boldsymbol{z}_{(i)}^* \mid S] \begin{cases} \leq E_{t-1,k} & \text{if } k = i \\ = 0 & \text{if } k \neq i \end{cases},
$$
(146)

and

$$
\mathbb{E}[E_{t-1,k}\mathrm{sign}(\boldsymbol{z}_{(k)}^* - \boldsymbol{z}_{t-1,(k)})\mathrm{sign}(\boldsymbol{z}_{t,(k)})\boldsymbol{z}_{(i)}^* \mid S] \begin{cases} \leq E_{t-1,i} & \text{if } k = i \\ = 0 & \text{if } k \neq i \end{cases}
$$
(147)

Hence,

$$
\sum_{k \neq j}\mathbb{E}[E_{t-1,k}\mathrm{sign}(\boldsymbol{z}_{(k)}^* - \boldsymbol{z}_{t-1,(k)})\boldsymbol{z}_{(i)}^* \mid S] \begin{cases} \leq E_{t-1,i} & \text{if } j \neq i \\ = 0 & \text{if } j = i \end{cases}
$$
(148)

and

$$
\sum_{k \neq j}\mathbb{E}[E_{t-1,k}\mathrm{sign}(\boldsymbol{z}_{(k)}^* - \boldsymbol{z}_{t-1,(k)})\mathrm{sign}(\boldsymbol{z}_{t,(k)})\boldsymbol{z}_{(i)}^* \mid S] \leq \begin{cases} E_{t-1,i} & \text{if } j \neq i \\ 0 & \text{if } j = i \end{cases}
$$
(149)

Hence, for $j \neq i$, we can write

$$\mathbb{E}[\boldsymbol{z}^*_{(i)} \zeta^{(l)}_{T,j} \mid S]] \leq 2\delta_l + 2\frac{\mu_l}{\sqrt{m}} \sum_{t=1}^{T} \alpha(1-\alpha)^{T-t} E_{t-1,i} + \tilde{\kappa}^{(l)}_{T,j} \tag{150}$$

where from (103), we have

$$E_{t-1,i}(1-\alpha)^{T-t} \leq (1 + 2(s-1)\alpha\frac{\mu_l}{\sqrt{m}}(t-1))E_{0,\max}\delta_{\alpha,T-2}$$
$$+ (\beta^{(l)}_{\max}|\boldsymbol{z}^*_{\max}| + 2\gamma^{(l)}_{\max})\left(\sum_{k=1}^{t-1} \alpha(1-\alpha)^{T-k-1} + 2(s-1)\kappa_l(1-\alpha)^{T-t}\right) \tag{151}$$

Hence, given the sparsity level, the term below is bounded by $a_\gamma$ with probability of $1 - \epsilon^{(l)}_\gamma$.

$$\sum_{t=1}^{T} E_{t-1,i}(1-\alpha)^{T-t} \leq \sum_{t=1}^{T}(1 + 2(s-1)\alpha\frac{\mu_l}{\sqrt{m}}(t-1))E_{0,\max}\delta_{\alpha,T-2}$$
$$+ (\beta^{(l)}_{\max}|\boldsymbol{z}^*_{\max}| + 2\gamma^{(l)}_{\max})\left(\sum_{t=1}^{T}\sum_{k=1}^{t-1} \alpha(1-\alpha)^{T-k-1} + \sum_{t=1}^{T} 2(s-1)\kappa_l(1-\alpha)^{T-t}\right) \tag{152}$$
$$\leq (\beta^{(l)}_{\max}|\boldsymbol{z}^*_{\max}| + 2a^{(l)}_\gamma)(1 + s\kappa_l) = \mathcal{O}(a^{(l)}_\gamma)$$

Finally, we get

$$\mathbb{E}[\boldsymbol{z}^*_{(i)} \zeta^{(l)}_{T,j} \mid S]] \leq \begin{cases} 2\delta_l + \frac{\mu_l}{\sqrt{m}}\mathcal{O}(a^{(l)}_\gamma) + \tilde{\kappa}^{(l)}_{T,j} & \text{if } j \neq i \\ \tilde{\kappa}^{(l)}_{T,j} & \text{if } j = i \end{cases} \tag{153}$$

For appropriately large $T$, $\kappa^{(l)}_{T,j}$ can be make small; Hence, in this case, we get

$$\|U^{(l)}_{T,j}\|_2 \leq \mathcal{O}(\sqrt{p}p_{ij}\delta_l\|\boldsymbol{D}^{(l)}\|_2) \tag{154}$$

Now, we can re-write the gradient as

$$\boldsymbol{g}^{(l)}_{T,j} = p_j(1-\beta^{(l)}_j)(\boldsymbol{D}^{(l)}_j - \boldsymbol{D}^*_j) + p_j(-\beta^{(l)}_j \boldsymbol{D}^{(l)}_j + \frac{1}{p_j}U^{(l)}_{T,j} + \frac{1}{p_j}\gamma) = \tau(\boldsymbol{D}^{(l)}_j - \boldsymbol{D}^*_j) + \theta \tag{155}$$

where $\tau = p_j(1-\beta^{(l)}_j)$, and $\theta = p_j(-\beta^{(l)}_j \boldsymbol{D}^{(l)}_j + \frac{1}{p_j}U^{(l)}_{T,j} + \frac{1}{p_j}\gamma)$. We can bound the norm of $\theta$ as follows:

$$\|\theta\|_2 \leq p_j\beta^{(l)}_j\|\boldsymbol{D}^{(l)}_j\|_2 + \|U^{(l)}_{T,j}\|_2 + \gamma \tag{156}$$

Given $\|\boldsymbol{D}^{(l)}_j\|_2 = 1$, and $\beta^{(l)}_j = \langle \boldsymbol{D}^*_j - \boldsymbol{D}^{(l)}_j, \boldsymbol{D}^*_j \rangle = \frac{1}{2}\|\boldsymbol{D}^{(l)}_j - \boldsymbol{D}^*_j\|^2_2$, we modify the upper bound

$$\|\theta\|_2 \leq \frac{1}{2}p_j\|\boldsymbol{D}^{(l)}_j - \boldsymbol{D}^*_j\|^2_2 + \mathcal{O}(\sqrt{p}p_{ij}\delta_l\|\boldsymbol{D}^{(l)}\|_2) + \gamma \tag{157}$$

We assume a dictionary closeness during training, i.e., $\|\boldsymbol{D}^{(l)} - \boldsymbol{D}^*\|_2 \leq 2\|\boldsymbol{D}^*\|_2$, which we prove in Lemma A.7. Given this closeness, we have

$$\|\boldsymbol{D}^{(l)}\|_2 \leq \|\boldsymbol{D}^{(l)} - \boldsymbol{D}^*\|_2 + \|\boldsymbol{D}^*\|_2 = \mathcal{O}(\sqrt{\frac{p}{m}}) \tag{158}$$

Moreover, with $\gamma$ dropping with $\delta_l$, and for $s = \mathcal{O}(\sqrt{m})$, it is reduced to

$$\|\theta\|_2 \leq p_j\|\boldsymbol{D}^{(l)}_j - \boldsymbol{D}^*_j\|_2 \tag{159}$$

We get

$$\|\boldsymbol{g}^{(l)}_{T,j}\|_2 \leq p_j(1-\beta^{(l)}_j)\|\boldsymbol{D}^{(l)}_j - \boldsymbol{D}^*_j\|_2 + p_j\|\boldsymbol{D}^{(l)}_j - \boldsymbol{D}^*_j\|_2$$
$$\|\boldsymbol{g}^{(l)}_{T,j}\|^2_2 \leq p^2_j(2-\beta^{(l)}_j)^2\|\boldsymbol{D}^{(l)}_j - \boldsymbol{D}^*_j\|^2_2 \tag{160}$$

Using this bound, we can find a lower bound on the correlation between the gradient direction and the desired direction as follows

$$\|\boldsymbol{g}_{T,j}^{(l)}\|_2^2 = (p_j(1-\beta_j^{(l)}))^2\|\boldsymbol{D}_j^{(l)} - \boldsymbol{D}_j^*\|_2^2 + \|\theta\|_2^2 + 2p_j(1-\beta_j^{(l)})\langle\theta, \boldsymbol{D}_j^{(l)} - \boldsymbol{D}_j^*\rangle$$

$$2\langle\theta, \boldsymbol{D}_j^{(l)} - \boldsymbol{D}_j^*\rangle = -p_j(1-\beta_j^{(l)})\|\boldsymbol{D}_j^{(l)} - \boldsymbol{D}_j^*\|_2^2 + \frac{1}{p_j(1-\beta_j^{(l)})}\|\boldsymbol{g}_{T,j}^{(l)}\|_2^2 - \frac{1}{p_j(1-\beta_j^{(l)})}\|\theta\|_2^2$$

$$2\langle\boldsymbol{g}_{T,j}^{(l)}, \boldsymbol{D}_j^{(l)} - \boldsymbol{D}_j^*\rangle = p_j(1-\beta_j^{(l)})\|\boldsymbol{D}_j^{(l)} - \boldsymbol{D}_j^*\|_2^2 + \frac{1}{p_j(1-\beta_j^{(l)})}\|\boldsymbol{g}_{T,j}^{(l)}\|_2^2 - \frac{1}{p_j(1-\beta_j^{(l)})}\|\theta\|_2^2$$

$$\geq (p_j(1-\beta_j^{(l)}) - p_j\frac{1}{1-\beta_j^{(l)}})\|\boldsymbol{D}_j^{(l)} - \boldsymbol{D}_j^*\|_2^2 + \frac{1}{p_j(1-\beta_j^{(l)})}\|\boldsymbol{g}_{T,j}^{(l)}\|_2^2 \tag{161}$$

Hence, using the descent property of Theorem 6 from (Arora et al., 2015), setting the learning rate to $\eta = \max_j \frac{1}{p_j(1-\beta_j^{(l)})}$, and $\psi = \eta(p_j(1-\beta_j^{(l)}) - p_j\frac{1}{1-\beta_j^{(l)}}) \leq 1 - \frac{1}{(1-\beta_j^{(l)})^2}$

$$\|\boldsymbol{D}_j^{(l+1)} - \boldsymbol{D}_j^*\|_2^2 \leq (1-\psi)\|\boldsymbol{D}_j^{(l)} - \boldsymbol{D}_j^*\|_2^2 \tag{162}$$

∎

**Theorem 4.12** (Dictionary learning with fixed $\lambda_t$). *Given Assumptions 3.3 and 3.4, suppose $\boldsymbol{D}^{(l)}$ is $\mu_l$-incoherent and $(\delta_l, 2)$-close to $\boldsymbol{D}^*$ with $\delta_l = \mathcal{O}^*(1/\log p)$. If $s = \mathcal{O}(\sqrt{m})$, $\mu = \mathcal{O}(\log m)$, learning rate is $\eta = \mathcal{O}(\frac{p}{s(1-\delta_l^2/2)})$, and the regularizer $\lambda^{fixed}$ and step size are set according to Theorem 4.5, then for any dictionary update $l$ using $\boldsymbol{g}_T^{dec}$, with probability of at least $1 - \epsilon_{supp\text{-}pres}^{(l)}$,*

$$\|\boldsymbol{D}_j^{(l+1)} - \boldsymbol{D}_j^*\|_2^2 \leq (1-\psi)\|\boldsymbol{D}_j^{(l)} - \boldsymbol{D}_j^*\|_2^2 + \epsilon_\lambda^{(l)} \tag{16}$$

*where $\epsilon_\lambda^{(l)} := \eta\frac{2p}{s(1-\beta_j^{(l)})}\lambda^{fixed2}$, $\epsilon_{supp\text{-}pres}^{(l)} := \epsilon_{supp\text{-}rec}^{(l)} + \epsilon_\gamma^{(l)} = 2p\exp\left(\frac{-C_{\min}^2}{\mathcal{O}^*(\delta_l^2)}\right) + 2s\exp\left(\frac{-1}{\mathcal{O}(\delta_l)}\right)$.*

*Proof.* Following steps similar to Theorem 4.11, we write the gradient as

$$\boldsymbol{g}_{T,j}^{(l)} = p_j\boldsymbol{D}_j^{(l)}(1-\beta_j^{(l)})^2 - p_j\boldsymbol{D}_j^*(1-\beta_j^{(l)}) + U_{T,j}^{(l)} + \gamma = p_j(1-\beta_j^{(l)})\left((1-\beta_j^{(l)})\boldsymbol{D}_j^{(l)} - \boldsymbol{D}_j^*\right) + U_{T,j}^{(l)} + \gamma \tag{163}$$

where $U_{T,j}^{(l)} = \mathbb{E}[\boldsymbol{D}_S^{(l)}(\boldsymbol{I} - B_S^{(l)})E[\boldsymbol{z}_{(S)}^*\zeta_{T,j}^{(l)} \mid S]] - \mathbb{E}[\boldsymbol{D}_S^*E[\boldsymbol{z}_{(S)}^*\zeta_{T,j}^{(l)} \mid S]] + \mathbb{E}[\boldsymbol{D}_S^{(l)}\mathbb{E}[(\boldsymbol{z}_{(j)}^*(1-\beta_j^{(l)}) + \zeta_{T,j}^{(l)})\zeta_{T,S}^{(l)} \mid S]]$. Given this gradient, we now find a bound on $U_{T,j}^{(l)}$. First, we bound $\mathbb{E}[\boldsymbol{z}_{(i)}^*\zeta_{T,j}^{(l)} \mid S]]$ as following.

$$\mathbb{E}[\boldsymbol{z}_{(i)}^*\zeta_{T,j}^{(l)} \mid S]] \leq \sum_{t=1}^T \alpha(1-\alpha)^{T-t}\mathbb{E}[\gamma_j^{(l)}\boldsymbol{z}_{(i)}^* \mid S] + \sum_{t=1}^T \alpha(1-\alpha)^{T-t}\mathbb{E}[\lambda^{fixed}\text{sign}(\boldsymbol{z}_{t-1,(j)})\boldsymbol{z}_{(i)}^* \mid S]$$

$$+ \frac{\mu_l}{\sqrt{m}}\sum_{t=1}^T \alpha(1-\alpha)^{T-t}\sum_{k\neq j}\mathbb{E}[E_{t-1,k}\text{sign}(\boldsymbol{z}_{(k)}^* - \boldsymbol{z}_{t-1,(k)})\boldsymbol{z}_{(i)}^* \mid S] + \tilde{\kappa}_{T,j}^{(l)} \tag{164}$$

where we set all $\lambda_{t-1,j}^{(l)} = \lambda^{fixed}$ and $\tilde{\kappa}_{T,j}^{(l)} = \mathbb{E}[\boldsymbol{z}_{(i)}^*\kappa_{T,j}^{(l)}]$. Similar to $\kappa_{T,j}^{(l)}$, $\tilde{\kappa}_{T,j}^{(l)}$ decay very fast as $T$ increases. Hence, we bound the other terms. We have

$$\mathbb{E}[\gamma_j^{(l)}\boldsymbol{z}_{(i)}^* \mid S]\begin{cases}\leq \delta_l & \text{if } j \neq i \\ = 0 & \text{if } j = i\end{cases}, \tag{165}$$

$$\mathbb{E}[\lambda^{fixed}\text{sign}(\boldsymbol{z}_{t-1,(j)})\boldsymbol{z}_{(i)}^* \mid S]\begin{cases}= \lambda^{fixed} & \text{if } j = i \\ = 0 & \text{if } j \neq i\end{cases}, \tag{166}$$

$$\mathbb{E}[E_{t-1,k}\text{sign}(\boldsymbol{z}_{(k)}^* - \boldsymbol{z}_{t-1,(k)})\boldsymbol{z}_{(i)}^* \mid S]\begin{cases}\leq E_{t-1,k} & \text{if } k = i \\ = 0 & \text{if } k \neq i\end{cases} \tag{167}$$

Hence,

$$\sum_{k \neq j} \mathbb{E}[E_{t-1,k}\text{sign}(\boldsymbol{z}^*_{(k)} - \boldsymbol{z}_{t-1,(k)})\boldsymbol{z}^*_{(i)} \mid S] \begin{cases} \leq E_{t-1,i} & \text{if } j \neq i \\ = 0 & \text{if } j = i \end{cases} \tag{168}$$

Hence, for $j \neq i$, we can write

$$\mathbb{E}[\boldsymbol{z}^*_{(i)}\zeta^{(l)}_{T,j} \mid S]] \leq \delta_l + \frac{\mu_l}{\sqrt{m}}\sum_{t=1}^{T}\alpha(1-\alpha)^{T-t}(E_{0,i} + E_{t-1,i}) + \tilde{\kappa}^{(l)}_{T,j} \tag{169}$$

We have

$$E_{t-1,i}(1-\alpha)^{T-t} \leq (1 + 2(s-1)\alpha\frac{\mu_l}{\sqrt{m}}(t-1))E_{0,\max}\delta_{\alpha,T-2}$$
$$+ (\beta^{(l)}_{\max}|\boldsymbol{z}^*_{\max}| + 2\gamma^{(l)}_{\max})\left(\sum_{k=1}^{t-1}\alpha(1-\alpha)^{T-k-1} + 2(s-1)\kappa_l(1-\alpha)^{T-t}\right) \tag{170}$$

Hence, given the sparsity level, the term below is bounded by $a_\gamma$ with probability of $1 - \epsilon^{(l)}_\gamma$.

$$\sum_{t=1}^{T}E_{t-1,i}(1-\alpha)^{T-t} \leq \sum_{t=1}^{T}(1 + 2(s-1)\alpha\frac{\mu_l}{\sqrt{m}}(t-1))E_{0,\max}\delta_{\alpha,T-2}$$
$$+ (\beta^{(l)}_{\max}|\boldsymbol{z}^*_{\max}| + 2\gamma^{(l)}_{\max})\left(\sum_{t=1}^{T}\sum_{k=1}^{t-1}\alpha(1-\alpha)^{T-k-1} + \sum_{t=1}^{T}2(s-1)\kappa_l(1-\alpha)^{T-t}\right) \tag{171}$$
$$\leq (\beta^{(l)}_{\max}|\boldsymbol{z}^*_{\max}| + 2a^{(l)}_\gamma)(1 + s\kappa_l) = \mathcal{O}(a^{(l)}_\gamma)$$

Finally, we get

$$\mathbb{E}[\boldsymbol{z}^*_{(i)}\zeta^{(l)}_{T,j} \mid S]] \leq \begin{cases} \delta_l + \frac{\mu_l}{\sqrt{m}}\mathcal{O}(a^{(l)}_\gamma) + \tilde{\kappa}^{(l)}_{T,j} & \text{if } j \neq i \\ \lambda^{\text{fixed}} + \tilde{\kappa}^{(l)}_{T,j} & \text{if } j = i \end{cases} \tag{172}$$

For appropriately large $T$, $\tilde{\kappa}^{(l)}_{T,j}$ can be make small; Hence, in this case, we get

$$\|U^{(l)}_{T,j}\|_2 \leq \mathcal{O}(\sqrt{p}p_{ij}\delta_l\|\boldsymbol{D}^{(l)}\|_2 + p_j\lambda^{\text{fixed}}) \tag{173}$$

Now, we can re-write the gradient as

$$\boldsymbol{g}^{(l)}_{T,j} = p_j(1-\beta^{(l)}_j)(\boldsymbol{D}^{(l)}_j - \boldsymbol{D}^*_j) + p_j(-\beta^{(l)}_j\boldsymbol{D}^{(l)}_j + \frac{1}{p_j}U^{(l)}_{T,j} + \frac{1}{p_j}\gamma) = \tau(\boldsymbol{D}^{(l)}_j - \boldsymbol{D}^*_j) + \theta \tag{174}$$

where $\tau = p_j(1-\beta^{(l)}_j)$, and $\theta = p_j(-\beta^{(l)}_j\boldsymbol{D}^{(l)}_j + \frac{1}{p_j}U^{(l)}_{T,j} + \frac{1}{p_j}\gamma)$. We can bound the norm of $\theta$ as follows:

$$\|\theta\|_2 \leq p_j\beta^{(l)}_j\|\boldsymbol{D}^{(l)}_j\|_2 + \|U^{(l)}_{T,j}\|_2 + \gamma \tag{175}$$

Given $\|\boldsymbol{D}^{(l)}_j\|_2 = 1$, and $\beta^{(l)}_j = \langle \boldsymbol{D}^*_j - \boldsymbol{D}^{(l)}_j, \boldsymbol{D}^*_j \rangle = \frac{1}{2}\|\boldsymbol{D}^{(l)}_j - \boldsymbol{D}^*_j\|^2_2$, we modify the upper bound

$$\|\theta\|_2 \leq \frac{1}{2}p_j\|\boldsymbol{D}^{(l)}_j - \boldsymbol{D}^*_j\|^2_2 + \mathcal{O}(\sqrt{p}p_{ij}\delta_l\|\boldsymbol{D}^{(l)}\|_2 + p_j\lambda^{\text{fixed}}) + \gamma \tag{176}$$

We assume a dictionary closeness during training, i.e., $\|\boldsymbol{D}^{(l)} - \boldsymbol{D}^*\|_2 \leq 2\|\boldsymbol{D}^*\|_2$, which we prove in Lemma A.7. Given this closeness, we have

$$\|\boldsymbol{D}^{(l)}\|_2 \leq \|\boldsymbol{D}^{(l)} - \boldsymbol{D}^*\|_2 + \|\boldsymbol{D}^*\|_2 = \mathcal{O}(\sqrt{\frac{p}{m}}) \tag{177}$$

Moreover, with $\gamma$ dropping with $\delta_l$, and for $s = \mathcal{O}(\sqrt{m})$, it is reduced to

$$\|\theta\|_2 \leq p_j(\|\boldsymbol{D}^{(l)}_j - \boldsymbol{D}^*_j\|_2 + \lambda^{\text{fixed}}) \tag{178}$$

We get

$$
\begin{aligned}
\|\boldsymbol{g}_{T,j}^{(l)}\|_2 &\le p_j(1-\beta_j^{(l)})\|\boldsymbol{D}_j^{(l)}-\boldsymbol{D}_j^*\|_2 + p_j(\|\boldsymbol{D}_j^{(l)}-\boldsymbol{D}_j^*\|_2 + \lambda^{\text{fixed}}) \\
\|\boldsymbol{g}_{T,j}^{(l)}\|_2^2 &\le 2p_j^2(2-\beta_j^{(l)})^2\|\boldsymbol{D}_j^{(l)}-\boldsymbol{D}_j^*\|_2^2 + 2p_j^2\lambda^{\text{fixed2}}
\end{aligned}
\tag{179}
$$

Using this bound, we can find a lower bound on the correlation between the gradient direction and the desired direction as follows

$$
\|\boldsymbol{g}_{T,j}^{(l)}\|_2^2 = (p_j(1-\beta_j^{(l)}))^2\|\boldsymbol{D}_j^{(l)}-\boldsymbol{D}_j^*\|_2^2 + \|\theta\|_2^2 + 2p_j(1-\beta_j^{(l)})\langle\theta,\boldsymbol{D}_j^{(l)}-\boldsymbol{D}_j^*\rangle
$$

$$
2\langle\theta,\boldsymbol{D}_j^{(l)}-\boldsymbol{D}_j^*\rangle = -p_j(1-\beta_j^{(l)})\|\boldsymbol{D}_j^{(l)}-\boldsymbol{D}_j^*\|_2^2 + \frac{1}{p_j(1-\beta_j^{(l)})}\|\boldsymbol{g}_{T,j}^{(l)}\|_2^2 - \frac{1}{p_j(1-\beta_j^{(l)})}\|\theta\|_2^2
$$

$$
2\langle\boldsymbol{g}_{T,j}^{(l)},\boldsymbol{D}_j^{(l)}-\boldsymbol{D}_j^*\rangle = p_j(1-\beta_j^{(l)})\|\boldsymbol{D}_j^{(l)}-\boldsymbol{D}_j^*\|_2^2 + \frac{1}{p_j(1-\beta_j^{(l)})}\|\boldsymbol{g}_{T,j}^{(l)}\|_2^2 - \frac{1}{p_j(1-\beta_j^{(l)})}\|\theta\|_2^2
$$

$$
\ge (p_j(1-\beta_j^{(l)}) - 2p_j\frac{1}{1-\beta_j^{(l)}})\|\boldsymbol{D}_j^{(l)}-\boldsymbol{D}_j^*\|_2^2 + \frac{1}{p_j(1-\beta_j^{(l)})}\|\boldsymbol{g}_{T,j}^{(l)}\|_2^2 - \frac{2p_j}{(1-\beta_j^{(l)})}\lambda^{\text{fixed2}}
\tag{180}
$$

Hence, using the descent property of Theorem 6 from (Arora et al., 2015), setting the learning rate to $\eta = \max_j \frac{1}{p_j(1-\beta_j^{(l)})}$, and $\psi = \eta(p_j(1-\beta_j^{(l)}) - 2p_j\frac{1}{1-\beta_j^{(l)}})$

$$
\|\boldsymbol{D}_j^{(l+1)}-\boldsymbol{D}_j^*\|_2^2 \le (1-\psi)\|\boldsymbol{D}_j^{(l)}-\boldsymbol{D}_j^*\|_2^2 + \epsilon_\lambda
\tag{181}
$$

where $\epsilon_\lambda := \eta\frac{2p_j}{(1-\beta_j^{(l)})}\lambda^{\text{fixed2}}$ ∎

**Lemma A.7** (Dictionary maintains closeness). *Suppose $\boldsymbol{D}^{(l)}$ has $(\delta_l, 2)$-closeness to $\boldsymbol{D}^*$ where $\delta_l = \mathcal{O}^*(1/\log m)$, then with probability of $1 - \epsilon_{supp\text{-}pres}^{(l)}$, we have $\|\boldsymbol{D}^{(l+1)}-\boldsymbol{D}^*\|_2 \le 2\|\boldsymbol{D}^*\|_2$ when using $\boldsymbol{g}_T^{dec}$ and the network parameters set by Theorem 4.4.*

*Proof.* Given the dictionary update

$$
\boldsymbol{D}_j^{(l+1)}-\boldsymbol{D}_j^* = \boldsymbol{D}_j^{(l)}-\boldsymbol{D}_j^* - \eta\boldsymbol{g}_{T,j}^{(l)}
\tag{182}
$$

Then, with probability at least $1 - \epsilon_{\text{supp-pres}}^{(l)}$, we have the gradient

$$
\boldsymbol{g}_{T,j}^{(l)} = p_j(1-\beta_j^{(l)})(\boldsymbol{D}_j^{(l)}-\boldsymbol{D}_j^*) + p_j(-\beta_j^{(l)}\boldsymbol{D}_j^{(l)} + \frac{1}{p_j}U_{T,j}^{(l)} + \frac{1}{p_j}\gamma)
\tag{183}
$$

which we substitute in the dictionary update as below

$$
\boldsymbol{D}_j^{(l+1)}-\boldsymbol{D}_j^* = (1-\eta(p_j(1-\beta_j^{(l)})))(\boldsymbol{D}_j^{(l)}-\boldsymbol{D}_j^*) + \eta p_j\beta_j^{(l)}\boldsymbol{D}_j^{(l)} - \eta U_{T,j}^{(l)} - \eta\gamma)
\tag{184}
$$

writing the update in matrix form

$$
\boldsymbol{D}^{(l+1)}-\boldsymbol{D}^* = (\boldsymbol{D}^{(l)}-\boldsymbol{D}^*)\text{diag}(1-\eta(p_j(1-\beta_j^{(l)}))) + \eta\boldsymbol{D}^{(l)}\text{diag}(p_j\beta_j^{(l)}) - \eta\boldsymbol{D}^{(l)}F + \eta\boldsymbol{D}^*H - \eta\gamma)
\tag{185}
$$

where $F_{(ij)} = p_{ij}\mathbb{E}[(1-\beta_i^{(l)})\boldsymbol{z}_{(i)}^*\zeta_{T,j}^{(l)} \mid S] + p_{ij}\mathbb{E}[(\boldsymbol{z}_{(j)}^*(1-\beta_j^{(l)}) + \zeta_{T,j}^{(l)})\zeta_{T,i}^{(l)} \mid S]$, and $H_{(ij)} = p_{ij}\mathbb{E}[\boldsymbol{z}_{(i)}^*\zeta_{T,j}^{(l)} \mid S]$. Given, the bound $\|U_{T,j}^{(l)}\|_2 \le \mathcal{O}(\sqrt{p}p_{ij}\delta_l\|\boldsymbol{D}^{(l)}\|_2)$ from before, we get $\|F\|_F \le \mathcal{O}(pp_{ij}\delta_l)$ and $\|H\|_F \le \mathcal{O}(pp_{ij}\delta_l)$. Hence,

$$
\|\boldsymbol{D}^{(l)}F + \boldsymbol{D}^*H\|_2 \le \|\boldsymbol{D}^{(l)}\|_2\|F\|_F + \|\boldsymbol{D}^*\|_2\|H\|_F = \mathcal{O}(pp_{ij}\delta_l\|\boldsymbol{D}^*\|_2) = \mathcal{O}(\frac{s^2}{p\log m})\|\boldsymbol{D}^*\|_2
\tag{186}
$$

Using the maintained closeness at update $l$, we bound the terms in the dictionary update one by one below

$$
\|(\boldsymbol{D}^{(l)}-\boldsymbol{D}^*)\text{diag}(1-\eta(p_j(1-\beta_j^{(l)})))\| \le (1-\min_j\eta p_j(1-\beta_j^{(l)}))\|\boldsymbol{D}^{(l)}-\boldsymbol{D}^*\|_2 \le 2(1-\Omega(\eta s/p))\|\boldsymbol{D}^*\|_2
\tag{187}
$$

$$\|\boldsymbol{D}^{(l)}\mathrm{diag}(p_j\beta_j^{(l)})\|_2 \le \max_j p_j \frac{\delta_l^2}{2}\|\boldsymbol{D}^{(l)} - \boldsymbol{D}^* + \boldsymbol{D}^*\|_2 \le o(s/p)\|\boldsymbol{D}^*\|_2 \tag{188}$$

Given the bounds above, the dictionary update can be bounded as following

$$\|\boldsymbol{D}^{(l+1)} - \boldsymbol{D}^*\|_2 \le 2(1 - \Omega(\eta s/p))\|\boldsymbol{D}^*\|_2 + o(\eta s/p)\|\boldsymbol{D}^*\|_2 + \mathcal{O}(\frac{\eta s^2}{p\log m})\|\boldsymbol{D}^*\|_2 + \eta\gamma \le 2\|\boldsymbol{D}^*\|_2 \tag{189}$$

∎

### A.6 Interpretability

**Theorem 5.1** (Interpretable unrolled network)**.** *Consider the dictionary learning optimization of the form* $\min_{\boldsymbol{Z},\boldsymbol{D}} \frac{1}{2}\|\boldsymbol{X} - \boldsymbol{D}\boldsymbol{Z}\|_F^2 + \lambda\|\boldsymbol{Z}\|_1 + \omega/2\|\boldsymbol{D}\|_F^2$, *where* $\boldsymbol{X} = [\boldsymbol{x}^1, \boldsymbol{x}^2, \dots, \boldsymbol{x}^n] \in \mathbb{R}^{m\times n}$ *and* $\boldsymbol{Z} = [\boldsymbol{z}^1, \boldsymbol{z}^2, \dots, \boldsymbol{z}^n] \in \mathbb{R}^{p\times n}$. *Let* $\tilde{\boldsymbol{Z}}$ *be the given converged sparse codes, then stationary points of the problem w.r.t the network weights (dictionary) follows* $\tilde{\boldsymbol{D}} = \boldsymbol{X}\boldsymbol{G}^{-1}\tilde{\boldsymbol{Z}}^T$, *where we denote* $\boldsymbol{G} \coloneqq (\tilde{\boldsymbol{Z}}^T\tilde{\boldsymbol{Z}} + \omega\boldsymbol{I})la.$

*Proof.* For all stationary points, the objective gradient is $\boldsymbol{0}$ with respect to the dictionary, i.e.,

$$\boldsymbol{0} = (\boldsymbol{X} - \tilde{\boldsymbol{D}}\tilde{\boldsymbol{Z}})\tilde{\boldsymbol{Z}}^{\mathrm{T}} + \omega\tilde{\boldsymbol{D}} \tag{190}$$

where $\tilde{\boldsymbol{D}}$ is the learned dictionary at convergence. Re-aranging the terms, we get

$$\tilde{\boldsymbol{D}} = \boldsymbol{X}\tilde{\boldsymbol{Z}}^{\mathrm{T}}(\tilde{\boldsymbol{Z}}\tilde{\boldsymbol{Z}}^{\mathrm{T}} + \omega\boldsymbol{I})^{-1} \tag{191}$$

Using the relation $\boldsymbol{A}^{\mathrm{T}}(\boldsymbol{A}\boldsymbol{A}^{\mathrm{T}} + \omega\boldsymbol{I})^{-1} = (\boldsymbol{A}^{\mathrm{T}}\boldsymbol{A} + \omega\boldsymbol{I})^{-1}\boldsymbol{A}^{\mathrm{T}}$, we can re-write the solution as

$$\tilde{\boldsymbol{D}} = \boldsymbol{X}\boldsymbol{G}^{-1}\tilde{\boldsymbol{Z}}^{\mathrm{T}} \tag{192}$$

where we denote $\boldsymbol{G} \coloneqq (\tilde{\boldsymbol{Z}}^{\mathrm{T}}\tilde{\boldsymbol{Z}} + \omega\boldsymbol{I})$. ∎

## B Appendix - future works and limitations

**Beyond dictionary learning** Our results are founded on three main properties: Lipschitz differentiability of the loss, proximal gradient descent, and strong convexity in finite-iteration. The findings can be applied to other min-min optimization problems, e.g., ridge regression and logistic regression, following such properties. For example, our analysis generalizes to the unrolled network in (Tolooshams et al., 2020) for learning dictionaries using data from the natural exponential family. In this case, the least-squares loss is replaced with negative log-likelihood, and the dictionary models the data expectation.

**Limitations** Finite-iteration support selection (Proposition 4.1) (Hale et al., 2007) and strong convexity may seem stringent going beyond dictionary learning. Ablin et al. (2020) discuss generalization of local gradient convergence by relaxing strong convexity to the $p$-Łojasiewicz property (Ablin et al., 2020; Attouch & Bolte, 2009). We considered the noiseless setting and conjecture that the relative comparison of the gradients in the presence of noise still holds, where the upper bounds will involve an additional noise term. We focused on infinite sample convergence to highlight the relative differences between the gradients. We leave for future work the derivation of finite-sample bounds, a step similar to (Chatterji & Bartlett, 2017; Arora et al., 2015).

## C Appendix - details of experiments

PUDLE is developed using PyTorch (Paszke et al., 2017). We used one GeForce GTX 1080 Ti GPU.

### C.1 Numerical experiments for theories

**Dataset**  We generated $n = 10,000$ samples following (2). We sampled $\boldsymbol{D}^* \in \mathbb{R}^{50 \times 100}$ from zero-mean Gaussian distribution, and normalized the columns. The codes are 5-sparse with their support uniformly chosen at random and their amplitudes are sampled from Uniform$(1, 2)$.

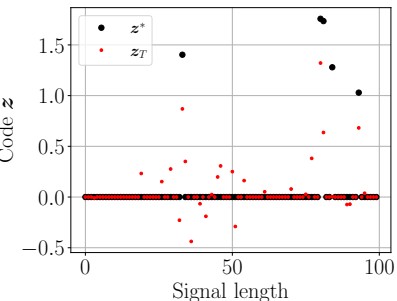

**Training**  We let $T = 200$, $\lambda = 0.2$, and $\alpha = 0.2$. The dictionary is initialized to $\boldsymbol{D} = \boldsymbol{D}^* + \tau_B \boldsymbol{B}$ with $\boldsymbol{B} \sim \mathcal{N}(\boldsymbol{0}, \frac{1}{m}\boldsymbol{I})$. For Figures 2, 3 and 4a, we set $\tau_B \approx {}^{0.55}/_{\log m}$. Figure 11 shows the sparse code estimates from one example given this initialized dictionary; this is to highlight that a) the initial dictionary is not very close to the ground-truth dictionary, and b) our algorithm is able to successfully perform dictionary learning and recover the support by the end of training in spite of a failed exact recovery of the support.

Figure 11: Example of code estimates with the initialized dictionary.

For Figures 4b and 4c, we chose much larger noise level, $\tau_B \approx {}^{2.8}/_{\log m}$. The network is trained for 600 epochs with full-batch gradient descent using Adam optimizer (Kingma & Ba, 2014) with learning rate of $10^{-3}$ and $\epsilon = 10^{-8}$. The learned dictionary is evaluated based on the error $\|\boldsymbol{D} - \boldsymbol{D}^*\|_2 / \|\boldsymbol{D}^*\|_2$. The results and conclusion were consistent across various realizations of the dataset and across various optimizers. Hence, in the main paper, the figures visualize results of one realization.

**Noisy measurements**  We repeated the dictionary learning experiments shown in Figures 4b and 4c where the measurements $\boldsymbol{x}$ are corrupted by zero-mean Gaussian noise such that the SNR is approximately 12 dB. Accordingly, we set $\lambda = 0.3$. Figure 12 shows the results for both noisy and noiseless scenarios.

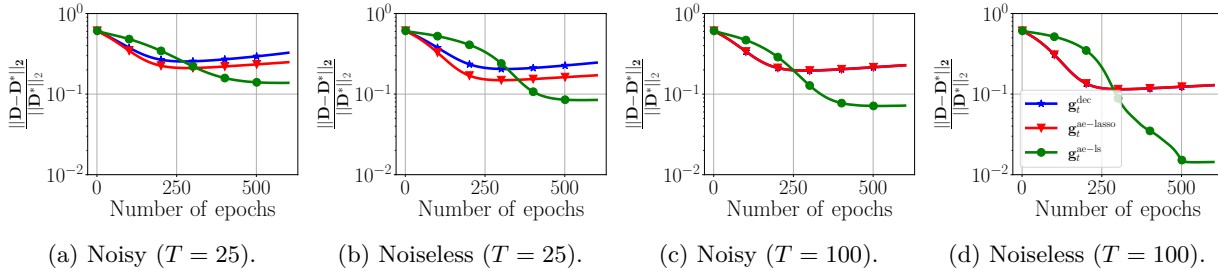

(a) Noisy ($T = 25$).  (b) Noiseless ($T = 25$).  (c) Noisy ($T = 100$).  (d) Noiseless ($T = 100$).

Figure 12: Dictionary learning in noisy and noiseless scenarios.

**Stochastic Dictionary Learning**  In addition to the full-batch gradient descent results in the main paper, we repeated the experiments in Figure 4c using batch size of $4, 16$ and $64$. We observed (Figure 13) that in all scenarios PUDLE is able to learn a good estimate of the ground-truth dictionary, and $\boldsymbol{g}_t^{\text{ae-ls}}$ is superior to the other two. We note that for lower batch-size, there will be more gradient updates in one epoch, hence, the algorithm converges in lower number of epochs.

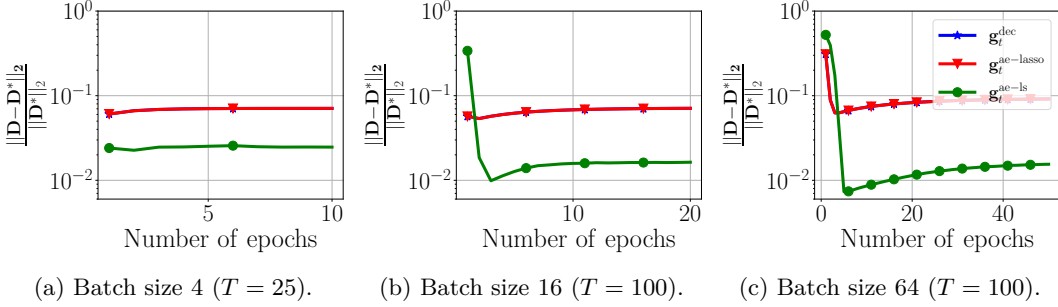

(a) Batch size 4 ($T = 25$).  (b) Batch size 16 ($T = 100$).  (c) Batch size 64 ($T = 100$).

Figure 13: Dictionary learning using various batch sizes.

**Effect of learning rate**  We performed the experiments in Figure 4c for various learning rate of $10^{-4}, 10^{-3}$, and $10^{-2}$. This shows the robustness of the gradient-based dictionary learning against learning rate. Figure 14

demonstrates such results where PUDLE successfully converges to the neighbourhood of the ground-truth dictionary; Regardless of the learning rate, $\boldsymbol{g}_t^{\text{ae-ls}}$ converges to a closer neighbourhood than the other two gradients. Overall, smaller the learning rate, more epochs is needed to reach convergence.

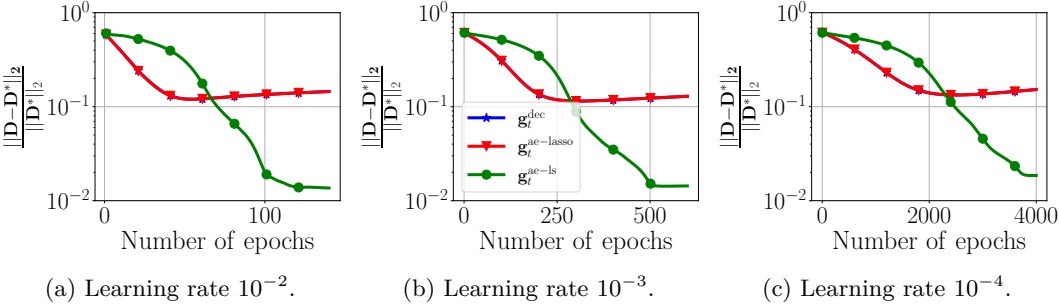

Figure 14: Dictionary learning for various learning rates when $T = 100$.

**Dictionary initialization** We conducted similar experiments to Figures 4b and 4c. We let $n = 10{,}000$, $m = 100$, and $p = 100$. We generated an orthogonal $\boldsymbol{D}^*$. The sparse codes $\boldsymbol{z}^*$ are 5-sparse and their amplitudes are drawn from sub-Gaussian $\mathcal{N}(0,1)$. We set $\lambda = 0.05$, and $\alpha = 0.2$. We used the pairwise method proposed by Arora et al. (2015) to initialize the dictionary. This close initialization resulted in a dictionary that provides support recovery prior training. Figure 15 shows successful dictionary learning where $\boldsymbol{g}_t^{\text{ae-ls}}$ converges to a closer neighbourhood of $\boldsymbol{D}^*$ than the other two gradients. We used linear sum assignment optimization (i.e., `scipy.optimize.linear_sum_assignment`) to find the correct column permutations before computing the dictionary distance error.

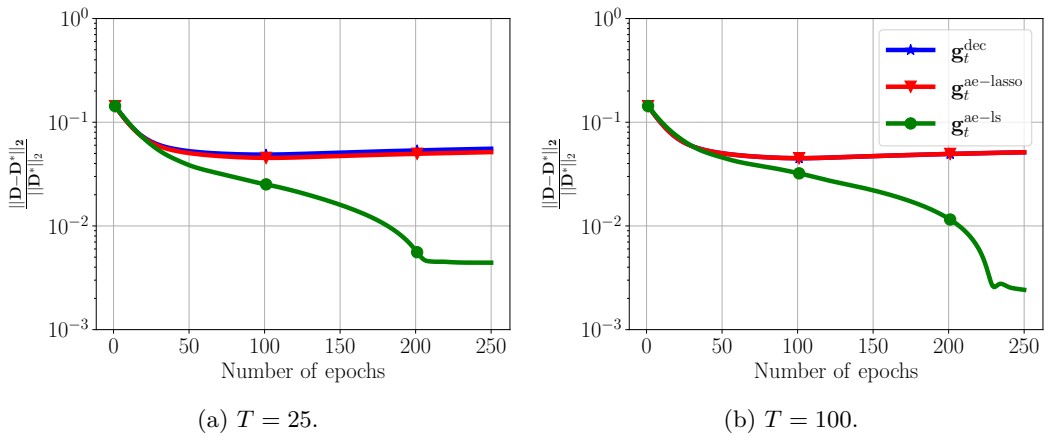

Figure 15: Dictionary learning when $\boldsymbol{D}$ is initialized using the pairwise method Arora et al. (2015).

## C.2 Dictionary learning

**Dataset** We generated $n = 50{,}000$ samples following (2). We let $m = 1000$ and $p = 1500$, and sample $\boldsymbol{D}^*$ from zero-mean Gaussian distribution, and then normalized the columns. The sparse codes $\boldsymbol{z}^i$ are 10, 20, 40-sparse, where their supports are chosen uniformly at random and amplitudes are sampled from Uniform$(1, 2)$.

**Training** The dictionary is initialized to $\boldsymbol{D} = \boldsymbol{D}^* + \tau_B \boldsymbol{B}$ with $\boldsymbol{B} \sim \mathcal{N}(\boldsymbol{0}, \frac{1}{m}\boldsymbol{I})$ where $\tau_B \approx 1/\log m$. We let $\lambda = 0.2$, and $\alpha = 0.2$, and $T = 100$. The network is trained for 1,000 iterative updates with batch-size of 50 using Adam (Kingma & Ba, 2014) with learning rate of $10^{-3}$ and $\epsilon = 10^{-3}$. For decay method, $\nu$ is decreased in value by 0.005 every 100 update iterations. Each filter is normalized after every update. The learned dictionary is evaluated based on the relative error $\|\boldsymbol{D} - \boldsymbol{D}^*\|_2 / \|\boldsymbol{D}^*\|_2$.

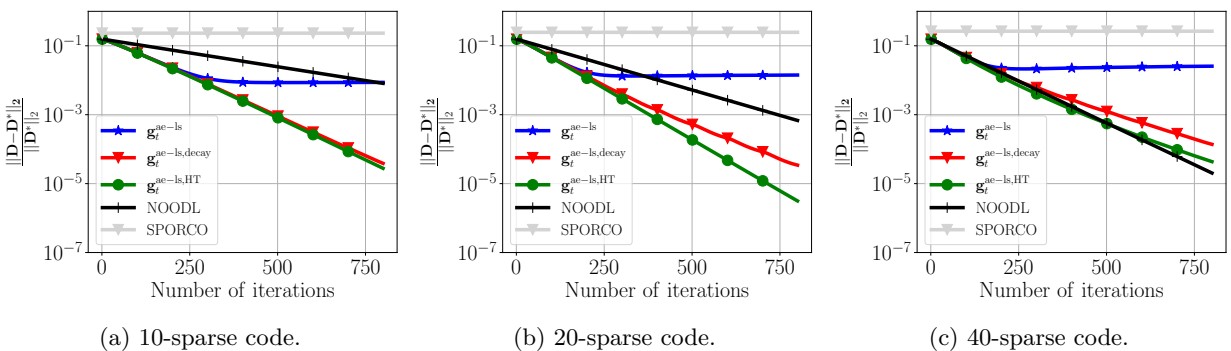

Figure 16: Dictionary learning convergence using $\boldsymbol{g}_t^{\text{ae-ls}}$ compared to NOODL and SPORCO.

## C.3  Image denoising

**Training**  We trained PUDLE where the dictionary is convolutional with 64 filters of size $9 \times 9$ and strides of 4. The encoder unrolls for $T = 15$, and the step size is set to $\alpha = 0.1$. Unlike the theoretical analysis where full-batch gradient descent is studied, the network is trained stochastically with Adam optimizer (Kingma & Ba, 2014) with a learning rate of $10^{-4}$ and $\epsilon = 10^{-3}$ for 250 epochs. At every training iteration, a random $129 \times 129$ patch is cropped and a zero-mean Gaussian noise with a standard deviation of 25 is added. We utilize random horizontal and vertical flip for augmentation. We report results in terms of the peak signal-to-noise ratio (PSNR). The standard deviation of the test PSNR across multiple noise realizations was lower than 0.02 dB for all the methods. Hence, we only reported the mean PSNR of the test set.

## C.4  Interpretable sparse coding and dictionary learning

We focused on digits of $\{0, 1, 2, 3, 4\}$ of MNIST. We set $T = 15$, $\lambda = 0.7$, and $\alpha = 1$. The dictionary dimensions are $m = 784$ and $p = 500$. We trained the network for 200 epochs using Adam optimizer with a learning rate of $10^{-4}$ and batch size of 32. For construction of $\boldsymbol{G}$, $\omega$ is set to 0.001. For Figure 8, we computed the image contributions using 6,000 randomly chosen training images. The Gram matrix used in Figure 9, is constructed by 6,000 training examples, and the reconstruction is from the 200 most contributed training images.

