# OpenReview forum: "Stable and Interpretable Unrolled Dictionary Learning"
_TMLR — Accepted by TMLR_

### Review · Reviewer_YjJw · 2022-06-03

**Summary Of Contributions:**

This manuscript is focused on applying unrolled sparse coding to dictionary learning and investigates the bias of the sparse coding step and the gradient stability when using auto differentiation (back-propagation) to calculate the dictionary gradient. The motivation of this work is to address two limitations in the previous work: (1) the bias introduced by the sparse coding step with constant sparsity regularization; (2) the unstable gradient issue during back-propagation when updating the dictionary.

**Requested Changes:**

In terms of issues, I think the major issue of this manuscript is the lack of discussions of many important assumptions for the theorems, especially on if and how they are (approximately, at least) satisfied in real-world problems.
- The dictionary closeness assumption (Assumption 3.5) is an important one for the theorems proposed later. But without access to information about D^*, it could be hard to have such good initialization. In Appendix C.1 and C.2, the initialization of D is done by adding noises to the ground truth D^*. In the image denoising experiments (Appendix C.3), the detail of D initialization is dismissed. This needs to be discussed.
- The decaying scheme of lambda_t in Theorem 4.2 also depends on the access to the ground-truth code z^*, which is not practical in real optimization. I understand that in practice maybe we can just decay lambda_t with a constant decaying ratio, as the authors have done in the experiments. But some discussions are needed.

Overall, this work has done a great job providing clear and confusion-free definitions and notations. However, there are still some places that confuse me:
- The biggest confusion to me is the definition of Code Jacobian (Def 4.1), where the Jacobian of z_t, J_t, is defined as \partial{z_t} over \partial{D}. It is natural to understand in terms of z_t because the calculation of z_6 depends on D according to the forward pass in eqn (6). But how should I understand \hat{J} and J^* in Theorem 4.6, when \hat{z} and z^* are just two objects that could be totally independent of D? At least, it does not make sense to me what they denote without specifying an algorithm to obtain them.
- In the formulation of gradients during the backward pass in eqn (7), why the gradient of D depends on the lowercase t but not the uppercase T?


Minor issues:
- Page 1, 5th line to the last, "... is ubiquitous is many other fields...", the second "is" --> "in "
- Page 7, last line of Theorem 4.1 about the sign consistency. Why a ReLU is used there? With ReLU, the signs on the left side of the equation are either 0 or +1, but on the other side, the signs of the ground-truth code z^* can definitely take -1, according to the code distribution assumption 3.3.
- Page 10, Fig.5, the lines of NOODL and SPORCO are not quite distinguishable. I would suggest add markers to the lines for better visualization.

**Strengths And Weaknesses:**

The authors proposed the so-called PUDLE framework (a Provable Unrolled Dictionary LEarning). Under this framework, they provide conditions on network initialization and data distribution sufficient to recover and preserve the support of the latent code. They also provide theoretical analysis on the forward pass to characterize the behaviors of the code Jacobian, which are later used for the backward analysis. In the backward analysis, the authors analyzed the convergence of two alternative gradients using back-propagation along with the standard decoder gradient used in classic dictionary learning methods in the context of the unrolled dictionary, and provide results of convergence and biases towards the local gradient and global gradient.

The authors conduct experiments on synthesized data and real-image denoising data to verify their results and corroborated the effectiveness of their proposed PUDLE method with properly modified loss. The proposed method also enables them to interpret how each atom in the learned dictionary is related to samples in the training set and how to relate new testing samples with training data.

The math proof looks like no problem to me.

Overall, I think this is a very nice piece of work that contains complete and deep analyses of the forward and backward passes in an unrolled dictionary learning framework, and that provides how the choices on the loss function, unrolling, and back-propagation influence the convergence of the learned dictionary. However, there are still a few issues that need to be fixed before acceptance.

---

> ### Author Response · Authors · 2022-06-08
> **Initial Authors' Response**
>
> The authors thank the reviewer for their insightful comments and enthusiastic assessment. We will shortly provide our response, in the form of comments, addressing the reviewer's comments. Additionally, we will soon upload a revision of our manuscript addressing the comments.

---

> ### Author Response · Authors · 2022-06-25
> **Initial Response from Authors Regarding Dictionary Closeness, Decaying of the Regularization Parameter, and Minor Comments**
>
> We would like to thank the reviewer for their insightful comments and enthusiastic assessment. We address the reviewer's comments both below and in the manuscript. We will soon provide our response regarding Jacobian and upload a revision of our manuscript where we have highlighted our modifications in blue.
>
> $\textbf{Dictionary closeness}$:
> 1. Arora et al. (2015) proposed a dictionary initialization method offering $(\delta, 2)$-close to ${\bf D}^{\ast}$ for $\delta = O^{\ast}(1 / \log m)$ (See Theorem 4 and Section 5 for its proof in (Arora et al., 2015). The method is based on pairwise reweighting of samples {${\bf x}^i$} for $i=1, ..., n$ from the generative model (2), and does not require access to ${\bf D}^{\ast}$. In addition, Ramabhatla et al. (2018) utilize dictionary closeness assumptions and such dictionary initialization for their theoretical analysis. Moreover, Agarwal et al. (2017) proposed a clustering approach to find a close initial estimate of the dictionary. We have added the aforementioned information after Assumption 3.5.
> 2. We have conducted additional experiments similar to Figures 4b and 4c, where the dictionary is initialized using the aforementioned pairwise method (Arora et al., 2015). We have included this in Appendix C.1 paragraph ``Dictionary initialization''.
> 3. We chose to use random initialization of the dictionary in our image denoising experiment to follow the initialization norm in the deep learning literature. This was to emphasize the practicality and usefulness of PUDLE in the absence of an initialization method. The dictionary is initialized using a standard Normal distribution. We have included this information to Section 4.3.
>
> $\textbf{Decaying scheme of $\lambda_t$:}$
> 1. The utilization of knowledge of the code error (in $\ell_1$ or $\ell_2$ norm) to set the proper thresholding/bias/regularization parameters ($\lambda$), both in optimization problems and neural network architectures, constitutes a fairly standard practice, as we argue below. We include the following discussion after Theorem 4.2 and explain that, indeed, the bounds on the $\lambda$ can further break down into terms related to the dictionary closeness.
>    - For the preservation of correct signed-support in a sparse coding network, Ramabhatla et al. (2018) in Lemma 2 provided a proper thresholding value at every iteration as a function of the $\ell_1$ norm of the code error with respect to a ground-truth code. In addition, Ramabhatla et al. (2018) in  Lemma 3 demonstrate that we can find an upper bound on the estimate of the code coefficients as a function of dictionary closeness. Our framework is closest to theirs.
>    - Nguyen et al. (2019) use information on the range of ground-truth code to choose proper biases in their neural network to guarantee support recovery.
>    - Given the dictionary closeness assumption, and code information such as its sparsity, it is possible to find error bounds between a code estimate and the ground-truth code. For example, Chatterji \& Bartlett (2017) use a different sparse coding algorithm but provide $\ell_{\infty}$-norm upper bounds as a function of terms such as code sparsity, data dimensionality, code range, and dictionary error (see Section 2.1 in their paper).
>    - Chen et al. (2018) similarly provide an upper bound on the bias of their unrolled sparse coding network at every layer as a function of $\ell_2$-norm error between the code estimate at the layer and the ground-truth code (see (25) page 2 of their supplementary material).
>
> 2. We agree with the reviewer that setting the $\lambda_t$ in practice is a challenging task, but it is a possible task given the results discussed by Ramabhatla et al. (2018). We explain this in the new version of the manuscript.
>
> $T$ vs. $t$: Given $T$ iterations in the forward pass, we should use ${\bf g}_T$ to update the dictionary. We have resolved this confusion by replacing small $t$ with a capital $T$ in (7).
>
> $\textbf{Minor}$: Such issues are addressed in the updated manuscript.
>
> A. Agarwal, A. Anandkumar and P. Netrapalli, "A Clustering Approach to Learning Sparsely Used Overcomplete Dictionaries,'' in IEEE Transactions on Information Theory, vol. 63, no. 1, pp. 575-592, Jan. 2017.

---

> ### Author Response · Authors · 2022-06-25
> **Authors' Response Concerning Jacobian.**
>
> $\textbf{Jacobian:}$ We thank the reviewer for their comment. Indeed, elaboration is needed here and we include this in the new version of the manuscript. Let $\mathcal{L}({\bf z}, {\bf D}) = \frac{1}{2} \\| {\bf x} - {\bf D} {\bf z}\\|_2^2$, $h({\bf z}) = \lambda \\| {\bf z} \\|_1$, and $f({\bf D}, {\bf z}) = \mathcal{L}({\bf z}, {\bf D}) + h({\bf z})$.
>
> - We define $\hat {\bf J} $ as follows: considering the function ${\bf z} \rightarrow \mathcal{L}\_{{\bf x}}({\bf z}, {\bf D} ) + h({\bf z})$, $\hat {\bf z}({\bf D} )$ is its minimizer and $\hat {\bf J}  = \frac{\partial \hat {\bf z}({\bf D} )}{\partial {\bf D} }$. Now, consider the minimizer $\hat {\bf z}({\bf D} )$, we have ${\bf 0} \in \nabla\_1 f(\hat {\bf z}, {\bf D})$, hence, we get ${\bf 0} \in \hat {{\bf J}}^{+}({{\bf D} }) \nabla\_{11}^2 f(\hat {\bf z}, {\bf D} ) + \nabla\_{21}^2 f(\hat {\bf z}, {\bf D} )$. Given the support recovery, the relation $\hat {\\bf J}^{+} ({\\bf D} ) (\\nabla_{11}^{2} \\mathcal{L}\_{{\\bf x}}(\\hat {\\bf z}, {\\bf D} ) \\odot \mathbf{1}\_{S^{\\ast}}) + \nabla\_{21}^2 \mathcal{L}\_{{\bf x}}(\hat {\bf z}, {\bf D} ) \odot \mathbf{1}\_{S^{\ast}} = {\bf 0}$  also holds which is equivalent to $Q(\hat {\bf z}, \hat {\bf J} )$ under the support. To show this, given the recursion ${\bf z}\_{t+1} = \Phi({\bf z}\_t, {\bf D} )$, we differentiate it and get ${\bf J}\ _{t+1} = \nabla\_1 \Phi({\bf z}\_t, {\bf D} ) {\bf J}\ _t + \nabla\_2 \Phi({\bf z}_t, {\bf D} )$. Given the support recovery and fixed-point property, we can write
>
> $$
> \hat {\bf J} = \mathbf{1}\_{S} \odot (\hat {\bf J} - \alpha \nabla\_{11}^2 \mathcal{L}\_{{\bf x}}({\bf z}\_t, {\bf D})^{\text{T}} \hat {\bf J}) + \mathbf{1}\_{S} \odot (- \alpha \nabla\_{21}^2 \mathcal{L}\_{{\bf x}}({\bf z}\_t, {\bf D}))^{\text{T}}
> $$
> $$
> \hat {\bf J} - \mathbf{1}\_{S} \odot \hat {\bf J} =  - \hat {\bf J} \alpha \nabla\_{11}^2 \mathcal{L}\_{{\bf x}}({\bf z}\_t, {\bf D})^{\text{T}} \odot \mathbf{1}\_{S}  - \alpha \nabla_{21}^2 \mathcal{L}\_{{\bf x}}({\bf z}\_t, {\bf D})^{\text{T}} \odot \mathbf{1}\_{S}\\
> $$
> $$
> {\bf 0} = \hat {\bf J}^{+} (\nabla_{11}^2 \mathcal{L}\_{{\bf x}}(\hat {\bf z}, {\bf D}) \odot \mathbf{1}\_{S}) +  \nabla_{21}^2 \mathcal{L}\_{{\bf x}}\hat {\bf z}, {\bf D}) \odot \mathbf{1}\_{S}
> $$
>
> If the term $(\nabla\_{11}^2 \mathcal{L}({\bf z}\_t, {\bf D}) \odot \mathbf{1}\_{S})$ is invertible, then we can write
> $$
> \hat {\bf J}^{+} = - (\nabla\_{11}^2 \mathcal{L}\_{{\bf x}}(\hat {\bf z}, {\bf D}) \odot \mathbf{1}\_{S})^{-1} \nabla\_{21}^2 \mathcal{L}\_{{\bf x}}(\hat {\bf z}, {\bf D}) \odot \mathbf{1}\_{S}
> $$
>
> For the Jacobian corresponding to row $i$ of the dictionary, we get
> $$
> \hat {\bf J}\_{(i,:)} = - ({\bf D}\_S^{\text{T}} {\bf D}\_S)^{-1} ({\bf D}\_{i, :} {\hat {\bf z}}^{\text{T}} + ({\bf D}\_{i, :}^{\text{T}} \hat {\bf z} - {\bf x}\_i) {\bf I}\_p)\_S
> $$
>
> on the support. Outside of the support $S$, the Jacobian is zero. This proof is similarly provided by Malézieux et al. (2022) (see Page 18, (41)). We have revised our manuscript accordingly to increase clarity.
>
> - For ${\bf z}^{\ast}$, we define the mapping function ${\bf z} \rightarrow \mathcal{L}\_{{\bf x}}({\bf z}, {\bf D} )$, where ${\bf z}^{\ast}({\bf D} )$ is its minimizer evaluated at ${\bf D}^{\ast}$, i.e., $\nabla_{1} \mathcal{L}\_{{\bf x}}({\bf z}^{\ast}, {\bf D}^{\ast} ) = {\bf D}^{\ast\text{T}} ({\bf D}^{\ast} {\bf z}^{\ast} - {\bf x}) = 0$ given the generative model (${\bf x} = {\bf D}^{\ast} {\bf z}^{\ast}$). Hence, we define the Jacobian ${\bf J}^{\ast} = \frac{\partial {\bf z}^{\ast}({\bf D} )}{\partial {\bf D} }\rvert\_{{\bf D} = {\bf D}^{\ast}}$. From implicit function theorem, we get
> $$
> {\bf J}^{\ast+}  \nabla\_{11}^2 \mathcal{L}\_{{\bf x}}({\bf z}^{\ast}, {\bf D}^{\ast}) + \nabla\_{21}^2 \mathcal{L}\_{{\bf x}}({\bf z}^{\ast}, {\bf D}^{\ast}) = {\bf 0}
> $$
>
> which is used in the global backward pass analysis. Alternatively, if $\nabla\_{11}^2 \mathcal{L}\_{\bf x}({\bf z}^{\ast}, {\bf D})$ is invertible, then we can compute ${\bf J}^{\ast+}$ as follows:
> $${\bf J}^{\ast+} = - \nabla\_{21}^2 \mathcal{L}\_{\bf x}({\bf z}^{\ast}, {\bf D}^{\ast}) \nabla\_{11}^2 \mathcal{L}\_{{\bf x}}({\bf z}^{\ast}, {\bf D}^{\ast})^{-1}$$
>
> The Jacobian w.r.t row $i$ of the dictionary is
> $$
> {\bf J}^{\ast}\_{(i,:)} = - ({\bf D}^{\ast\text{T}}\_{S^{\ast}} {\bf D}^{\ast}\_{S^{\ast}})^{-1} ({\bf D}^{\ast}\_{i, :} {\bf z}^{\ast\text{T}} + ({\bf D}^{\ast\text{T}}\_{i, :} {\bf z}^{\ast} - {\bf x}\_i) {\bf I}_p)\_{S^{\ast}}
> $$
>
> on the support $S^{\ast}$ of ${\bf z}^{\ast}$. Outside of the support, it is zero. We have added the above explanations to Theorem 4.6.

---

### Review · Reviewer_JMW1 · 2022-06-16

**Summary Of Contributions:**

This paper seeks to establish theoretical guarantees for unrolled dictionary learning. In unrolled dictionary learning, the gradients are back-propagated from the loss of the decoded sparse code through the algorithm that generated the sparse code. The task is global estimation, i.e. recovery of a ground truth dictionary D* and corresponding sparse code z*.
This paper has a few contributions. First, they establish sufficient conditions on the data distributions and the sparse coding parameters so that the “forward pass” i.e. sparse coding recovers the true code support.  Understandably success in recovering true sparse code relies on the proximity of the dictionary initialization to the optimum and a “bias” stemming from l-1 regularization
Second, they study three dictionary gradient variants using the final sparse code z_T-
1) g-dec : a gradient computed using only the loss incurred by decoding z_T,
2) g-ae-lasso : a gradient computed by back-propagating the lasso loss through the unrolled layers.
3) g-ae-ls : a gradient computed by back-propagating the least squares through the unrolled layers.
This paper analyzes the effectiveness of each of these gradients under a strong assumption - the existence of infinite sample data and the sparse code computed from infinite unrolled iterations. They conclude that the gradient g-ae-ls (in this infinite limit) is closer to the optimal gradient pointing directly to D*.



**Broader Impact Concerns:**

The paper does not have a broader impact statement. However, I don’t foresee any negative impact.


**Requested Changes:**

1) According to Definition A1. \mu-incoherence implies inner-products are bounded by \mu/\sqrt{m}.
2)  In Lemma 3.1, the inner-products of D^{l} are directly bounded by \mu_l rather than \mu_l/\sqrt{m}. Please correct and adjust subsequent calculations.
3) In equations 17, the index 1 on the LHS is in the superscript for z_1 and sub-script for z*. Please modify the notation to be consistent.
4) Please differentiate the notation of the three gradients for the following two cases 1) finite sample 2) infinite samples.
5) In equation 7, the indices on the gradient variants on the LHS should be T rather than t.
6) Typo in Proposition 4.1 “supp(z_B)”
7) Is C_min strictly required to be greater than 0 ?
8) Assumption 3.6 - for a unique minimizer the singleton set is exactly the argmin ?

**Strengths And Weaknesses:**

Strengths:
They attempt to find the right recipe for unrolled dictionary learning. Their final inference on the gradient g-ae-ls  indicates that it is better to relegate the need for sparse regularization to the sparse coding steps rather and that upon the computation of a sparse code z_T, the quality of a current dictionary iterate is better captured by the least squares objective.

Weaknesses:
1) Although this is a theoretical paper, the requirement of infinite data and infinite unrolled iterations feels quite strong. Infinite data in essence removes the stochasticity in the learning task, and it is unclear what a back-propagated gradient is under infinite unrolled iterations…
2) Theorem 4.1 makes high probability statements on the recover of the true code z*’s support. But the failure probability as stated is a fixed constant once the parameters - C_min , delta_l , p are fixed from the assumptions. A high probability bound requires a free parameter that can be varied such that the failure probability tends to 0 and in this case it is not so. Theorem 4.2 makes a similar probabilistic argument but here to set lambda_t appropriately requires knowledge of the true sparse code! Dependence of step-size or regularization hyperparameters on Lipschitz constants or diameters is relatively common but a direct dependence on the optimal sparse code is quite strong. This raises the following thought - does the procedure only give me convergence in high probability when the optimal sparse code is already known or should I treat this as an existence of unknowable optimal value for the parameter lambda_t ?
3) The analysis is tailored to the lasso problem for sparse coding and they infer that l-1 regularization biases the recovery of the true sparse code. But couldn’t this be rectified by simply considering the l-0 regularized problem and IHT for sparse coding?
4) Section 5 switches track to study the interpretability of learnt dictionaries. As a reader, this section appears to be completely independent of the remaining 90% of the paper. Theorem 5 for eg. is a straightforward inference from the stationary points. It would be great if the authors can clarify which part of Theorem 5 or their discussion thereafter is different for standard vs unrolled dictionary learning.
5) Additionally for the statement (on page 11, left of Figure 6) - “this proves the dictionary span the training set”, did the authors perhaps mean to say “this proves the dictionary *lives in* the span of the training set” as indicated by Equation 11? Further, in Equation 12, is x^j an actual test image? Or is the reconstructed version of the test image using a learnt dictionary?  The latter is more understandable and in essence just says that dictionary representation of an image is a linear combination of dictionary atoms each of which depend on the actual data. In this case, it is unclear what the interpretation is on direct test images. For eg - Even for a test image from a different subspace, I can produce reconstructed image using the sparse codes and quantify the contribution of different training images towards this reconstruction.

---

> ### Author Response · Authors · 2022-06-25
> **Initial Authors' Response to Infinite Data Assumption and Unrolling Layers**
>
> We would like to thank the reviewer for their insightful and helpful comments. Below, we provide our response to their comments.
>
> 1. We clarify on infinite data assumption and unrolling layers.
>    - Asymptotic unrolling results: In this paper, we provided asymptotic results, i.e., as the network unrolls, how fast the gradient estimates converge. Such analysis is similarly used by Ablin et al. (2020) and  Malézieux et al. (2022) for analysis of strongly bi-convex optimization objective and dictionary learning, respectively. For the local gradient estimates (Section 4.2, page 8), we focus on such asymptotic results. The local upper bounds as a function of $t$ also applies to the global upper bounds on the gradient errors with $t$ unfolding (see (53) to (55)). We clarify that we show the global gradient errors in the infinite unfolding in (9) to highlight the difference between the two gradients; both ${\bf g}^{\text{ae-lasso}}$ and ${\bf g}^{\text{ae-ls}}$ have same rate of convergence as a function of $t$. Similarly, the usage of ${\bf D}^{\ast}$ in (10) is to highlight the difference between the two gradients given the best possible dictionary estimate. Overall, our experiments uses unrolling of $T=25$ (Figure 4b), $T=100$ (Figure 4c), and $T=100$ (Figure 5) which highlights the difference between the gradients in the presence of finite number of unrolling.
>    - Amount of data: We clarify that the infinite amount of data is not a requirement to the gradient comparisons we have provided. We provide two arguments for the comparison of the gradients: 1) ${\bf g}^{\text{ae-lasso}}$ and ${\bf g}^{\text{ae-ls}}$ gradients converges faster than ${\bf g}^{\text{dec}}$ as the network unfolds, and 2) ${\bf g}^{\text{ae-ls}}$ has less bias than ${\bf g}^{\text{ae-lasso}}$ in computation of the desired gradient direction. We argue that such comparative results still hold in the finite-data regime and it is similar to the analysis in (Malézieux et al. (2022)). Moreover, to highlight the effect stochasticity, we have included additional experiments in Appendix C.1 paragraph ``Stochastic dictionary learning'' where instead of full-batch gradient descent, we use batch-size of $4$, $16$, and $64$ to learn the dictionary; in all scenarios, the networks successfully find a good estimate of the ground-truth dictionary, and ${\bf g}^{\text{ae-ls}}$ converges to a closer neighbourhood that ${\bf g}^{\text{ae-lasso}}$ and ${\bf g}^{\text{dec}}$.

---

> ### Author Response · Authors · 2022-06-25
> **Authors' Response Concerning Theorems 4.1 and 4.2**
>
> 2. Theorems 4.1 and 4.2:
>
> High probability statement: We note that as the dictionary learning continues and we get a better estimate of the dictionary, $\delta_l$ decreases. Hence, both support recovery and presentation probability increase. Specifically, their probability reaches one the as $\delta_l$ goes to zero (i.e., ${\bf D} \rightarrow {\bf D}^{\ast}$). Moreover, such high probability statements exist in the literature (Arora et al., 2015; Ramabhatla et al., 2018).
>
> Knowledge of code error:The utilization of knowledge of the code error to set the proper thresholding/bias/regularization parameters ($\lambda$), both in optimization problems and neural network architectures, constitutes a fairly standard practice, as we argue below. We include the following discussion after Theorem 4.2 and that indeed the bounds on the $\lambda$ can further break down into terms related to the dictionary closeness.
>
> -  For the preservation of correct signed-support in a sparse coding network, Ramabhatla et al. (2018) in Lemma 2 provided a proper thresholding value at every iteration as a function of the $\ell_1$ norm of the code error with respect to a ground-truth code. In addition, Ramabhatla et al. (2018) in  Lemma 3 demonstrate that we can find an upper bound on the estimate of the code coefficients as a function of dictionary closeness. Our framework is closest to theirs.
> - Nguyen et al. (2019) use information on the range of ground-truth code to choose proper biases in their neural network to guarantee support recovery.
> - Given the dictionary closeness assumption, and code information such as its sparsity, it is possible to find error bounds between a code estimate and the ground-truth code. For example, Chatterji \& Bartlett (2017) use a different sparse coding algorithm but provide upper bounds $\ell_{\infty}$-norm error as a function of terms such as code sparsity, data dimensionality, code range, and dictionary error (see Section 2.1 in their paper).
> - Chen et al. (2018) similarly provides an upper bound on the bias of their unrolled sparse coding network at every layer as a function of $\ell_2$-norm error between the code estimate at the layer and the ground-truth code (see (25) page 2 of their supplementary material).
>
> Overall, in response to ``does the procedure only give me convergence in high probability when the optimal sparse code is already known or should I treat this as an existence of unknowable optimal value for the parameter $\lambda_t$ ?'': We do not require the knowledge of the exact ground-truth code in practice, the code error in the the lower bound on the regularization parameter is upper bounded by $\mathcal{O}(s)$. Hence, as long as we have the knowledge of sparsity and dictionary closeness and incoherence, the regularization parameter can be set to be lower bounded on the order of ${\bf \Omega}(\frac{s\log{m}}{\sqrt{m}})$.
>
>
> 3. $\ell_0$ vs. $\ell_1$: We agree with the reviewer that using iterative hard-thresholding (IHT) based on $\ell_0$ resolves the bias introduced by $\ell_1$ for sparse coding. IHT is indeed studied by Ramabhatla et al. (2018). We have chosen to study an unrolled encoder architecture that arises from $\ell_1$ for the following two reasons:
>
> - The goal of our work is not to provide an unbiased trainable sparse coding network, but to show how to decrease the bias of an $\ell_1$-norm based encoder, variations on which have become increasingly popular in the literature (Gregor & LeCun, 2010; Moreau & Bruna, 2017; Chen et al., 2018; Simon & Elad, 2019; ; Liu & Chen, 2019; Tolooshams et al., 2021a; Malézieux et al., 2022).
> - Although both NOODL ($\ell_0$-based encoder) and ours ($\ell_1$-based with decaying lambda) can recover the dictionary (Figure 5), our image denoising analysis highlights that $\ell_1$-based encoder outperforms $\ell_0$ (Table 1).

---

> ### Author Response · Authors · 2022-06-25
> **Authors' Response on Interpretability and Minor Comments.**
>
> 4. Interpretability: We agree with the reviewer that the stationary point analysis can similarly be applied to dictionary learning solved through alternating-minimization optimization-based methods. One of the motivations behind using algorithm unrolling (Monga et al., 2019) as inference networks as opposed to generic deep neural networks is their interpretability. However, the literature lacks a systematic approach to support that unrolled networks are interpretable. In this section of the paper, we show that we can interpret the neural network weights and inference given the fact that the inference network is constructed based on the dictionary learning generative model. Otherwise, the interpretability of a deep neural network is not that simple. Additionally, we note that such mathematical relation and interpretability results is missing in the dictionary-learning literature, irrespective of whether one use an unrolling network. We have added this last sentence to the manuscript for clarification before Theorem 5.1.
>
> 5.
> - Span of the dataset: We agree with the reviewer on their statement about the span. We have resolved this confusion by modifying our statement to ``the dictionary lives in the span of the training set''.
> - Equation (12) should read $\hat {\bf x}^j = \tilde {\bf D} \hat {\bf z}^j$, where $\hat {\bf x}^j$ is reconstruction of the test image ${\bf x}^j$. We have accordingly modified the manuscript. In our analysis shown in Figures 7 and 8, we focused on the relative contribution of the training images. We have added an explanation to Figure 8 that the reported vector ${\bf \beta}_k^j$ is normalized. Aside from the relative comparison, we can look into the absolute contributions (no normalization). We can evaluate the overall quality of the reconstruction by looking into ${\bf \beta}_k^j$ in (12). For example, we observed that for test MNIST, unnormalized ${\bf \beta}_k^j$ corresponding to high contributing training images is above $1$. However, for resized-CIFAR, unnormalized ${\bf \beta}_k^j$ of high contributing training images are often half or an order of magnitude lower than the MNIST case. This informs us of a bad representation/reconstruction of CIFAR image by the trained network. We have added the above explanation to the manuscript.
>
> For requested changes,
>
> 1-2. $\mu$-incoherence: We thank the reviewer for pointing this out. We have corrected the typo in Lemma 3.1 as follows: $\mu_l = \mu + 2 \sqrt{m} \delta_l$.
>
> 3. We use subscript t to denote unfolding layers, and subscript (i) to denote $i$th entry of the vector. We have updated our manuscript accordingly.
>
> 5. We have fixed Equation (7).
>
> 6. We have fixed the typo on Proposition 4.1.
>
> 7. Strictly positive $C_min$ is not required. Same assumption is used by Arora et al. (2015) and Ramabhatla et al. (2018).
>
> 8. For better clarify, we have changed $\in$ to $=$.

---

> ### Comment · Reviewer_JMW1 · 2022-07-25
> **Some Minor Comments**
>
> I would like to first thank the authors for the clarification provided in their comments. The changes made to the draft appear to address the concerns. I have a few minor comments -
>
> 1) Your abstract says "Finally, we demonstrate PUDLE’s interpretability, a driving factor
> in designing deep networks based on iterative optimizations, by building a mathematical
> relation between network weights, its output, and the training set."
> - Based on the discussions I'm still unconvinced on the specific link between - PUDLE/unrolled networks and interpretability. The statements and experiments on interpretability would appear to hold for any dictionary learning algorithm with the only difference being the converged sparse codes. Hence I would request that you modify the statement to reflect this.
>
> 2) "Strictly positive C_{min} is not required" - in the case where C_{min} < 0, there are no high-probability statements as the LHS in inequality in 27 is 1, thus the failure probability is .. 1?  I might be missing something trivial here and if so, let me know.
>
> 3) Regarding "high probability bounds"
> - I agree that as the dictionary learning algorithm progresses, \delta_l decreases, but this is still not a high probability bound right? At any given fixed l, delta_l is a fixed constant, so the probability of code support recovery is strictly less than 1 and does not approach 1 by modifying the parameters (such as lambda_0).
> - Further when l=1, to ensure that the code-support recovery in first iteration happens with high probability - say a high fixed value of 0.95 - one needs to ensure that the initialization is strong enough to guarantee this. But this shifts the burden from the actual learning algorithm to the initialization algorithm (say of Arora(2015)) which in turn requires a much larger number of samples to achieve this.
> - Certainly these theorems hold with certain failure probabilities, the only thing I'm contending is that these are failure probabilities can be larger than desired.
> - One immediate fix that might work is to simply set lambda_0 to be C_free * C_{min}/2  where C_free is a free parameter and have the probability statement in (27) be Prob ( |v_i| > C_{free} * C_{min}/4 ) \leq 2 exp( -  (C_free * C_{min})^2/ O( \delta_l) ).

---

> > ### Author Response · Authors · 2022-08-02
> > **Authors' comments for camera ready**
> >
> > We thank the reviewer for their feedback. We have addressed the comments in the camera-ready version as follows:
> >
> > 1. We have clarified the interpretability before Theorem 5.1. "We note that such mathematical relation and interpretability results also hold for dictionary learning. However, it is missing in the literature, irrespective of whether one uses an unrolling network."
> > 2. $0 < C_{\text{min}}$ is addressed.
> > 3. We acknowledge that the probability of successful support recovery depends on the initial dictionary estimate. We have rephrased some of the "with high probability" statements to "with probability of $1 - \epsilon^{(l)}\_{\text{supp-pres}}$". This is to clarify that the success probability depends on $\epsilon^{(l)}\_{\text{supp-pres}}$ which might be large depending on the dictionary closeness and other factors.

---

### Review · Reviewer_i5t1 · 2022-06-16

**Summary Of Contributions:**

This paper introduces puddle, which is a new way of updating the dictionary in dictionary learning. After the sparse coding phase, the gradient of the model w.r.t. the dictionary is estimated by doing backprop through the least-squares loss (this gradient is called $g^{ae-ls}$), and not through the standard lasso cost function of dictionary learning (the gradient is called $g^{ae-lasso}$), or the standard gradient of the loss with respect to the dictionary (called $g^{dec}$).

The authors also provide a theoretical framework around dictionary learning, where it is shown that under sufficient conditions, one step of ISTA with appropriate choice of soft thresholding level gives perfect support recovery, and then go on to show linear convergence of the method. The authors then analyse the error made using the different gradient estimators compared to the gradient at the optimum of the lasso $\hat{z}$ and the gradient obtained at $z^*$, the true sparse code. The authors show that the proposed gradient $g^{ae-ls}$ is, under suitable hypothesis, a good estimator of the gradient obtained at $z^*$, while the other gradients are not. This justifies the approach of the paper.

These findings are demonstrated on some synthetic experiments, and on an image denoising task. They also provide a study of the learned atoms on MNIST.

**Broader Impact Concerns:**

In my view, there is no broader impact concern.

**Requested Changes:**

- In my view, a proposition that shows that assumption 3.5 is somewhat realistic would greatly strengthen the paper : is it true that if $D$ is close to $D^*$ at initialization, then it stays close to $D^*$ during training?

- Section 5 seems very disconnected from the rest of the paper, I think the paper would be strengthened if it were removed and instead replaced by more convincing and extensive experiments demonstrating the merits of the proposed method.

**Strengths And Weaknesses:**

Strengths
-------------

This paper proposes a new gradient estimator for the dictionary update. This gradient update seems promising when the observed data are actually a noiseless sparse combination of atoms.

Weaknesses
-----------------

**Theoretical analysis is lacking**
-  Theorem 4.1 and 4.2 show that under very restrictive conditions (among which, the fact that the current estimate of the dictionary $D$ and the true dictionary $D^*$ are close), then one iteration of ISTA is enough for support recovery. This is in starck contrast with what is usually observed in practice, and the well documented slow convergence and slow support identification of ISTA, see e.g. [1]. Also, these theorems are really local convergence theorems, since in practice one cannot get the guarantee that at initialisation, $D$ and $D^*$ are close
- Theorem 4.6 is very confusing: what is the Jacobian of $z^*$ w.r.t. $D$? This is critical, and should be defined. I don't understand how $z^*$, the ground truth code, depends on the dictionary $D$. Also, *this theorem is never proven* in appendix !
- The authors mention that their method solves some stability issues observed in [Malezieux et al. 2022], but in fact the authors just provide some very restrictive conditions upon which ISTA leads to support recovery and stable linear convergence, bypassing the problems raised by [Malezieux et al. 2022]. To me, the phrasing in most of the paper made it sound as if the authors proposed a new method to prevent this problem, not a simple setting in which these problems do not arise.

- Assumption 3.5 assumes that after each dictionary update, the dictionary $D$ stays close enough from $D^*$. This is a strong assumption; one could imagine that drifting happens and that even though $D$ starts close from  $D^*$, it goes away from it during training. Such case is not investigated.

- Lemma 3.1 looks suspicious to me: in the best scenario where $D^l = D^*$, then $D^l$ is only $\mu$-incoherent since $D^*$ is $\mu$-incoherent by definition. However, Lemma 3.1 would show that $D^l$ is rather $\mu / \sqrt{m}$-incoherent, which is stronger than $\mu$-incoherence since $m \geq 1$.

**I found the paper quite confusing and hard to read**

 As examples:
- the used gradients $g^{dec}, g^{ae-lasso}$ and $g^{ae- ls}$ are mentioned many times in the introduction but only properly defined on page 6.
- In assumptions 3.4  / 3.5, it would be better to write the meaning of incoherence and closeness in the main text, since these definitions are short and easy to understand.
- In assumption 3.5, $\delta_0$ and the sequence $\delta_l$ are not defined... They will be defined in hindsight in the next theorems.
- P.5, the loss function was already defined
- In algorithms 1 adn 2, there seem to be some projection step missing
- In eq.7, $\partial h(z)$ is a set, so how is it chosen when $z$ is 0?
- In prop 4.1, thm 4.3 and thm 4.5, why even introduce $B$ if in the end the conditions in thm 4.1/4.2 allow to take $B=1$?
- In eq.9, there are many higher order terms that can be safely removed, for instance $O(\hat{\delta}^2  + \hat{\delta}) = O(\hat{\delta})$.


**Experiments are not conclusive**

- The code, as it is shipped now, is impossible to run, since the datasets are not provided. Running e.g. ``python train.py with cfg.dictionarylearning_g_aelasso_T25`` gives  the error ``FileNotFoundError: [Errno 2] No such file or directory: '../data/simulated_data.pt'`` .
- In the experiments, the dictionary $D$ is initialized as a small perturbation of the true dictionary $D^*$: what happens when the initialization is random (or when we don't know in advance $D^*$? -- which is the case on real data). Also, the authors only choose one learning rate for the experiments, while a grid-search would have been much more convincing to highlight the merits of the proposed method.
- It would be preferable to have more experiments where the benefits of the proposed method are displayed.
- In the denoising experiment, the authors mention "$g^{ae-ls}$ uses full backpropagation and stays stable. This
highlights stability of the gradient computation as opposed to $g^{ae-lasso}$.", but this instability is never demonstrated or shown here.
- The denoising experiment is done only on a small dataset of 432 images. It would be more convincing if it where done on several datasets. Also, the authors should indicate the error bars in table 1, using different initializations for the dictionaries.
- The authors only compare to two other DL methods, while there are numerous methods readily available (e.g. online DL in scikit-learn).
- Section 5 is independent from the rest of the paper, and is a standard analysis of the learned atoms on the MNIST dataset. It is not connected to the results in the previous sections.

[1] : Liang, Jingwei, Jalal Fadili, and Gabriel Peyré. "Local linear convergence of Forward--Backward under partial smoothness." Advances in neural information processing systems 27 (2014).

---

> ### Author Response · Authors · 2022-06-26
> **Authors' Response on Dictionary Closeness, Jacobian, and Stability**
>
> We would like to thank the reviewer for assessment of our paper and their helpful comments. We address each comment one-by-one below. We will soon upload a revision of our manuscript where we have highlighted our modifications in blue.
>
> $\textbf{Dictionary closeness:}$ We agree with the reviewer that some assumptions may seem unnatural to practitioners. Taking close initialization as an example, while empirically we find dictionary learning succeeds without close initialization (e.g., the image denoising example), to our knowledge, there do not exist theoretical analysies of dictionary with random initialization. Indeed, all existing analyses (e.g., Arora et al., 2015; Chatterji \& Bartlett, 2017; Agarwal et al., 2017; Ramabhatla et al., 2018) assume some form of close initialization. Prior works exists to obtain such close initialization in practice.
> - We discuss two methods below. Arora et al. (2015)  proposed a dictionary initialization method offering $(\delta, 2)$-close to ${\bf D}^{\ast}$ for $\delta = O^{\ast}(1 / \log m)$ (See Theorem 4 and Section 5 for its proof in Arora et al., 2015). The method is based on pairwise reweighting of samples {${\bf x}^i$} for $i=1,...,n$ from the generative model (2), and does not require access to ${\bf D}^{\ast}$. In addition, Ramabhatla et al. (2018) utilize dictionary closeness assumptions and such dictionary initialization for their theoretical analysis. Moreover, Agarwal et al. (2017) proposed a clustering approach to find a close initial estimate of the dictionary. We have added the aforementioned information after Assumption 3.5.
> - We have conducted additional experiments similar to Figures 4b and 4c, where the dictionary is initialized using the pairwise method (Arora et al., 2015). We have included this in Appendix C.1 paragraph ``Dictionary initialization'' (Figure 15).
> - We agree with the reviewer that ISTA has slow convergence. However, support recovery after one iteration in unrolled networks is studied in several prior works in the literature (Arora et al., 2015; Ramabhatla et al., 2018; Chen et al., 2018; Nguyen et al., 2019). We have added this explanation before Theorem 4.1 where we also cite the paper suggested by the reviewer.
> - For Figures 4b and 4c, we have added an additional figure (Figure 11 in the Appendix) to highlight the sparse coding performance given the initialized dictionary. Figure 11 shows the sparse code estimates from one example given this initialized dictionary; this is to highlight that a) the initial dictionary is not very close to the ground-truth dictionary, and b) our algorithm is able to successfully perform dictionary learning and recover the support by the end of training in spite of a failed exact recovery of the support.
>
> $\textbf{Jacobian:}$ See Jacobian response to Reviewer YjJw. Additionally, we have included the proof and our detailed modifications for Theorem 4.6 in the Appendix.
>
> $\textbf{Stability:}$ We clarify our statement on stability as below. Our solution to the instability of backpropagation is two-fold. One, as stated by the reviewer, we provide a sufficient condition to avoid such instability. Second, we propose ${\bf g}^{\text{ae-ls}}$ which resolves the instability in the absence of support identification. This is indeed supported by the fact that ${\bf g}^{\text{ae-ls}}$ does not suffer from gradient explosion in the image denoising experiment. We have included additional figure to highlight this instability (see Figure 6). We have accordingly added the following explanation. First, we highlight the stability of ${\bf g}_t^{\text{ae-ls}}$ against ${\bf g}_t^{\text{ae-lasso}}$; Figure 6a shows the network dynamics in terms of test PSNR as a function of epochs when $\lambda = 0.16$ for ${\bf g}_t^{\text{ae-ls}}$, ${\bf g}_t^{\text{ae-lasso}}$, ${\bf g}_t^{\text{dec}}$ and $b=0.05$ for ${\bf g}_t^{\text{ae-ls, HT}}$.  We observed that ${\bf g}_t^{\text{ae-ls}}$ uses full backpropagation and stays stable. However, the training with ${\bf g}_t^{\text{ae-lasso}}$ is not stable and unstable to perform denoising where the noisy PSNR is approximately $20$ dB (Malézieux et al., 2022). Second, Figure 6b shows that compared to ${\bf g}_t^{\text{dec}}$, the backpropagated gradients result in a smoother improvement during training. We have modified our manuscript (specifically, the contribution section) accordingly and willing to clarify our language around our contribution to resolving the instability problem.
>
> $\textbf{Incoherence:}$ We thank the reviewer for pointing out the typo regarding $\mu_l$. We have corrected the typo in Lemma 3.1 as follows: $\mu_l = \mu + 2 \sqrt{m} \delta_l$.

---

> ### Author Response · Authors · 2022-06-26
> **Authors' Response Addressing Experiments**
>
> First, we thank the reviewer for sharing comments on clarity. We have incorporated such comments as found proper. For example, we have simplified the higher order terms for clarity. We have also highlighted in color blue our statement regarding projection step in Preliminaries.
>
> Second, we provide the following clarifications regarding our experiments.
>
> $\textbf{Code:}$ We have our data generating functions in "utils/datasets.py". Hence, the user should be able to generate similar data given this script and the parameters provided in "conf.py". We will try to upload the simulated data we have generated.
>
> $\textbf{Initialization:}$ See above for our explanation on the additional experiment we included for dictionary initialization and evaluation of the closeness of the perturbed dictionary (Figure 15 and Figure 6). We would like to note that for real data, the common practice in deep learning is to initialize the weights (dictionary) randomly following a certain distribution. In addition, for the dictionary learning literature, the common practice is to initialize the atoms by extracting random images from the dataset or performing PCA.
>
> $\textbf{Learning rate:}$ We have provided additional experiments where in addition to $10^{-3}$, the learning rate of $10^{-2}$, $10^{-4}$ is used. We have added additional explanation as below. We performed the experiments in Figure 3c for various learning rate of $10^{-4}, 10^{-3}$, and $10^{-2}$. This shows the robustness of the gradient-based dictionary learning against learning rate. Figure 14 demonstrates such results where PUDLE successfully converges to the neighbourhood of the ground-truth dictionary; Regardless of the learning rate, ${\bf g}_t^{\text{ae-ls}}$ converges to a closer neighbourhood than the other two gradients. Overall, smaller the learning rate, more epochs is needed to reach convergence.
>
> $\textbf{BSD data:}$ BSD dataset is a popular training dataset for denoising. This dataset has been used in literature for training of various state-of-the-arts deep learning frameworks (Zhang et al., 2017; Simon & Elad, 2019; Mohan et al., 2019). In our process of adding noise to the data, we use a difference noise realizations at every batch; hence, no noisy image is going to be seen twice. Moreover, we are cropping random patches of size $129 \times 129$ from an image that is of size $481 \times 321$. Then, we utilize random horizontal and vertical flip for augmentation. Overall, the literature finds this amount of data enough for training networks for image denoising. We have included additional explanation about the BSD dataset and our training procedure to the manuscript.
>
> $\textbf{Error bars:}$ We have repeated our image denoising experiment for three independent trials. We have added the mean and std of the test PSNR to Table 1.
>
> $\textbf{Interpretability:}$ Please see our response to Reviewer JMW1 regarding interpretability. Given the motivation we discussed, we choose to keep this section in our manuscript.
>
> Mohan, Sreyas, et al. "Robust and interpretable blind image denoising via bias-free convolutional neural networks." preprint arXiv:1906.05478 (2019).
>
> Zhang, Kai, et al. "Beyond a gaussian denoiser: Residual learning of deep cnn for image denoising." IEEE transactions on image processing 26.7 (2017): 3142-3155.

---

> ### Author Response · Authors · 2022-06-26
> **Authors' Comments for Dictionary Staying Close**
>
> We thank the reviewer for their comment of "is it true that if ${\bf D}$ is close to ${\bf D}^{\ast}$ at initialization, then it stays close to ${\bf D}^{\ast}$ during training?"
>
> We have begun theoretical analysis to provide such results which require additional work. We will post an update within the time frame of this interactive review session.

---

> ### Author Response · Authors · 2022-07-05
> **Adding New Results for Contractive Dictionary Update to Address Dictionary Closeness Comment**
>
> We thank the reviewer for their comment related to dictionary closeness. We found it helpful to make our manuscript stronger. In this regard, we have added series of analyses (four new theorems, one lemma, and their corresponding explanations) to our manuscript. Below, we describe each and refer the reader to the manuscript for detailed technical description of the theorems. The current version of the manuscript contains the theorems in the main paper, and we will include their proof soon in the appendices.
>
> $\textbf{Theorem 4.4}$ (Global forward pass code error with variable $\lambda_t$): This theorem considers the case where the network parameters (step size and regularization) are set by Theorem 4.2. First, we provide an error between the code entries at every unrolled iteration with respect to the ground-truth code. The upper bound is a function of sparsity, dictionary error, amount of unrolling, dictionary incoherence, etc. In this expression $e_{t,j}^{(l)\text{unroll}}$ vanishes as $t$ increases.
> $$
> | {\\bf z}\_{t,(j)}^{(l)} - {\\bf z}\_{(j)}^{\ast} | \\leq \\mathcal{O}(\\sqrt{s \\|{\\bf D}\_j^{(l)} - {\\bf D}\_j^{\ast} \\|\_2} + e\_{t,j}^{(l)\text{unroll}})
> $$
>
> Second, we provide the code estimation at layer $T$ as a function of ground-truth code, dictionary error, etc. In this expression, $\beta\_j^{(l)}$ and $\zeta\_{T,j}^{(l)}$ go to zero as dictionary error $\delta_l$ decays.
> $$
> {\\bf z}\_{T,(j)} = {\\bf z}^{\ast}\_{(j)} (1 - \\beta\_j^{(l)}) + \\zeta\_{T,j}^{(l)}
> $$
>
> $\textbf{Theorem 4.5}$ (Global forward pass code error with fixed $\lambda_t$): This theorem considers the case where the regularization is fixed and not changed across unrolled iterations. In this scenario, compared to Theorem 4.4, the code error is upper bounded by an additional term containing $\lambda$; this highlights the amplitude bias estimation of $\ell_1$-norm based sparse coding.
> $$
> | {\bf z}\_{t,(j)}^{(l)} - {\bf z}\_{(j)}^{\ast} | \leq \mathcal{O}(\sqrt{s \\|{\bf D}\_j^{(l)} - {\bf D}\_j^{\ast} \\|\_2} + e_{t,j}^{(l)\text{unroll, fixed}} + \lambda^{\text{fixed}})
> $$
>
> Regarding the code estimation at layer $T$, $\beta\_j^{(l)}$ goes to zero as dictionary error $\delta_l$ decays. However, $| \zeta\_{T,j}^{(l)}|$ is upper bounded by $\lambda^{\text{fixed}}$.
> $$
> {\bf z}\_{T,(j)} = {\bf z}^{\ast}\_{(j)} (1 - \beta\_j^{(l)}) + \zeta\_{T,j}^{(l)}
> $$
>
> The above forward pass results are used in dictionary learning theorems which we describe next.
>
> $\textbf{Theorem 4.11}$ (Dictionary learning with variable $\lambda_t$): This theorem consider the network parameters set by Theorem 4.4 in the forward pass. It proves that using ${\bf g}\_{T}^{\text{dec}}$, PUDLE recovers the dictionary; the dictionary error contracts at every update with probability of at least $1 - \epsilon\_{\text{supp-pres}}^{(l)}$.
> $$
> \\| {\bf D}\_j^{(l+1)} - {\bf D}\_j^{\ast} \\|_2^2 \leq (1 - \psi) \\| {\bf D}\_j^{(l)} - {\bf D}\_j^{\ast} \\|\_2^2
> $$
>
> $\textbf{Theorem 4.12}$ (Dictionary learning with fixed $\lambda_t$): This theorem consider the case where $\lambda$ is fixed (Theorem 4.5), and proves that ${\bf g}\_{T}^{\text{dec}}$ provides a contractive dictionary updates up to a ${\bf D}^{\ast}$-neighbourhood characterized by $\lambda^{\text{fixed}}$.
> $$
> \\| {\bf D}\_j^{(l+1)} - {\bf D}\_j^{\ast} \\|_2^2 \leq (1 - \psi) \\| {\bf D}\_j^{(l)} - {\bf D}\_j^{\ast} \\|\_2^2 + \epsilon\_{\lambda}
> $$
>
> The above analyses requires for ${\bf D}^{(l)}$ to maintain a closeness to ${\bf D}^{\ast}$ which we provide a proof in Lemma A.7. Hence, Assumption 3.5 stays valid for the duration of training.

---

### Author Response · Authors · 2022-06-27
**Updated Manuscript following Initial Reviewers' Comments**

We would like to thank the reviewers for assessment of our paper and their comments. We have uploaded an updated manuscript. It includes modifications following authors' initial response to reviewers' comments. The changes are made in blue. Main changes are concerning dictionary closeness, Jacobian definitions, stability, Theorems 4.1-2. Moreover, we have conducted additional experiments and added figures as follows:

- Reported mean and std for image denoising results (Table 1).
- Stability of backpropagation for image denoising (Figure 6).
- Sparse coding performance given the initial dictionary estimate (Figure 11).
- Dictionary learning in the presence of measurement noise (Figure 12).
- Stochastic dictionary learning as batch size varies (Figure 13).
- Effect of learning rate on dictionary learning (Figure 14).
- Successful dictionary learning given an initial dictionary estimate computed by a pairwise method proposed by Arora et al. (2015) (Figure 15).

---

### Author Response · Authors · 2022-07-08
**All Reviewers' Comments are Addressed in the Updated Manuscript**

The authors would like to thank the reviewers for their comments. We have tried our best in addressing all comments (modifications are made in blue). We look forward to hearing from the reviewers.

---

### Decision · Action_Editors · 2022-07-27

**Recommendation:** Accept with minor revision

**Comment:**

This paper focuses on applying unrolled sparse coding to dictionary learning and studies theoretical guarantees for unrolled dictionary learning, where the gradient with respect to the dictionary is estimated by doing back-propagation. The authors establish sufficient conditions on data distribution and dictionary so that the unrolled sparse coding recovers the true code support, and then shows dictionary error contracts at every back-propagation update. The work also proposes approaches to address (1) the bias introduced by the vanilla unrolled sparse coding and (2) the unstable gradient issue during back-propagation when updating the dictionary.

All the reviewers acknowledged the contribution of the analyses of the forward and backward passes in an unrolled dictionary learning framework. The reviewers pointed out comments on theoretical analysis (such as dictionary closeness and data assumption) as well as experiments. The authors submitted a revised version that addresses these comments and derives new theoretical results. All the reviews either lean toward or recommend accepting the paper. I agree with the reviewers to recommend accepting.

There are additional minor comments recently posted by one reviewer, which can be addressed in the final version. I may agree with the reviewer that it makes more sense to require C_{min} nonnegative. Aside from the reason for making failure probability small, at the end of the proof of Theorem 4.1, the regularizer parameter \lambda_0 is chosen as C_{min}/2 which also implies that nonnegative C_{min}.